# The January 2022 Hunga eruption cooled the southern hemisphere in 2022 and 2023

Ashok Kumar Gupta [1] ✉, Tushar Mittal [2], Kristen E. Fauria [3], Ralf Bennartz[3] & Jasper F. Kok [1]

The 2022 Hunga volcanic eruption injected a significant quantity of water vapor into the stratosphere while releasing only limited sulfur dioxide. It has been proposed that this excess water vapor could have contributed to global warming, potentially pushing temperatures beyond the 1.5 °C threshold of the Paris Climate Accord. However, given the cooling effects of sulfate aerosols and the contrasting impacts of ozone loss (cooling) versus gain (warming), assessing the eruption's net radiative effect is essential. Here, we quantify the Hunga-induced perturbations in stratospheric water vapor, sulfate aerosols, and ozone using satellite observations and radiative transfer simulations. Our analysis shows that these components induce clear-sky instantaneous net radiative energy losses at both the top of the atmosphere and near the tropopause. In 2022, the Southern Hemisphere experienced a radiative forcing of $-0.55 \pm 0.05$ W m$^{-2}$ at the top of the atmosphere and $-0.52 \pm 0.05$ W m$^{-2}$ near the tropopause. By 2023, these values decreased to $-0.26 \pm 0.04$ W m$^{-2}$ and $-0.25 \pm 0.04$ W m$^{-2}$, respectively. Employing a two-layer energy balance model, we estimate that these losses resulted in cooling of about $-0.10 \pm 0.02$ K in the Southern Hemisphere by the end of 2022 and 2023. Thus, we conclude that the Hunga eruption cooled rather than warmed the Southern Hemisphere during this period.

On 15 January 2022, the Hunga Tonga volcano erupted from a shallow (~200 m deep) submarine vent, injecting unprecedented amounts of water vapor[1–5] (~120–150 Tg) into the stratosphere. In addition, in situ and space-borne observations show that the eruption released a moderate amount of sulfur dioxide (~0.4–0.7 Tg) into the stratosphere, which was subsequently converted to sulfate aerosols over several weeks[2,3,5–16]. These two components, sulfate aerosol and water vapor, have opposite climate effects. Sulfate aerosols cool by scattering solar radiation, while water vapor, as a greenhouse gas, leads to stratospheric cooling and surface warming[9]. Its net radiative effect at the top of the atmosphere (TOA) depends on altitude, potentially causing either warming or cooling[9–12]. Given the unprecedented stratospheric water vapor injection, several studies have hypothesized that the Hunga eruption resulted in net climate warming[4,8,9]. Assessing this possibility is crucial, as volcanic-induced warming could increase the likelihood of temporarily exceeding the 1.5 °C threshold set by the Paris Agreement[4] in the coming years. In contrast, past large subaerial (continental arc) eruptions—such as the 1991 Pinatubo, 1883 Krakatau, and 1815 Tambora events—led to prolonged global cooling due to the dominant influence of sulfate aerosols[15–21].

The efficacy of sulfate aerosol-induced cooling is influenced by aerosol properties such as number concentration, mass concentration, particle size distribution, and residence time[5,12,14,22], all of which substantially impact mass extinction efficiency (MEE)—a measure of an aerosol's ability to attenuate radiation per unit mass[23]. MEE depends on particle radius through two primary factors: (1) Extinction efficiency: This parameter, for monodisperse particle, exhibits a non-monotonic dependence on particle size, with its first peak approximately 25% greater than its second peak[23]. For a broader, lognormal size distribution, these distinct peaks can merge into a single, smoother MEE maximum as the distribution width and effective particle size increase. (2) Total geometric cross-section: For a fixed aerosol mass, the total geometric cross-section is inversely proportional to particle radius[23]. Consequently, smaller particles have a larger total cross-sectional area per unit mass, which enhances scattering and increases MEE.

Explosive volcanic eruptions can also potentially impact stratospheric ozone concentrations, particularly when sulfate aerosols interact with a moist stratosphere[23–30]. Stratospheric ozone depletion leads to net cooling at Earth's TOA due to reduced shortwave absorption by ozone[19,24–35]. The climatic effects of shallow submarine eruptions can differ significantly

[1]Department of Atmospheric and Oceanic Sciences, University of California, Los Angeles, CA, USA. [2]Department of Geosciences, Pennsylvania State University, University Park, PA, USA. [3]Department of Earth and Environmental Sciences, Vanderbilt University, Nashville, TN, USA. ✉e-mail: ashokgupta@atmos.ucla.edu

from those of subaerial eruptions due to variations in water vapor, $SO_2$, and halogen gas emissions, which in turn influence ozone chemistry[36–38]. Thus, accurately determining the net impact of the Hunga eruption on stratospheric composition and radiative forcing is crucial. Most submarine eruptions occur in deep water and release minimal volcanic material into the atmosphere. However, the Hunga eruption—the largest shallow submarine eruption of the satellite era—offers a rare opportunity to assess the climate impacts of such events[39–41]. Thus, this requires assessing the perturbation of three key radiatively active species: water vapor, ozone, and sulfate aerosols.

Previous studies have either focused on a single atmospheric constituent or relied on simplified scaling analyses to estimate radiative forcing in the first and second years after the eruption[11,12]. This has likely contributed to conflicting conclusions, with some studies suggesting that the Hunga eruption caused warming[4], while others found it led to cooling[11,12]. For instance, Schoeberl[11] estimated a net negative (cooling) radiative forcing by combining a scaling analysis based on past subaerial eruptions (for aerosols) with radiative calculations for water vapor[38,42–44]. In a separate study, Schoeberl[12] applied this simplified aerosol radiative forcing scaling method to demonstrate a global reduction in downward radiative flux at the tropopause over two years, attributing this decline to changes in water vapor, aerosols, and ozone. In contrast, Jenkins[4] concluded that the Hunga eruption could contribute to global warming. However, this study only considered the impact of stratospheric water vapor ($SH_2O$) perturbation while neglecting the effects of aerosols, which are critical to the overall radiative balance.

Here, we conduct a comprehensive analysis of the instantaneous net radiative response to the Hunga eruption in 2022 and 2023 by: (a) Utilizing satellite remote sensing observations to quantify the spatiotemporal distribution of $SH_2O$, ozone, and sulfate aerosols (including their size distributions). (b) Using these observations as inputs for idealized 1D radiative transfer model simulations. Here, the model is run independently for each grid cell across the near-global domain, producing a three-dimensional output that captures both spatial and vertical variability within the stratosphere.

## Results

We used space-borne remote sensing data from SAGE III[45,46] aboard the International Space Station (ISS) (solar occultation, v5.3; available since June 2017) to quantify the three-dimensional distribution of key radiatively active stratospheric species affected by the Hunga eruption: $SH_2O$, aerosols, and ozone (see Methods and Supplementary Fig. 1). SAGE III/ISS employs a solar occultation technique, capturing up to 31 measurements daily during sunrise and sunset to generate dawn and dusk stratospheric profiles. This method requires approximately one month to cover latitudes from ~60°N to 60°S[45,46], providing detailed monthly coverage of the tropics and mid-latitudes, with seasonal variations influencing the sampling pattern.

Additionally, we use multi-wavelength SAGE III/ISS observations to retrieve stratospheric aerosol particle size before and after the Hunga eruption.

For our radiative forcing analysis, we used SAGE III/ISS data due to its exceptional vertical resolution of 0.5 km. This high-resolution dataset on radiatively active species—water vapor, ozone, and aerosols—is critical for accurately estimating atmospheric heating rates and radiative forcings at the TOA and tropopause[47]. However, SAGE III/ISS data have known limitations, including a dry bias in water vapor measurements[48]. Earlier versions of SAGE III showed a ~ 10% dry bias in the stratosphere, which improved to ~5% in Version 5.2 and 5.3 for the mid-stratosphere[49,50]. Additionally, water vapor data below ~20 km are noisier due to aerosol and cloud interference, contributing to ~10–20% uncertainty in radiative forcing estimates, particularly in the lower stratosphere[48]. Furthermore, the infrequent sampling of aerosol extinction profiles at multiple wavelengths introduces biases in stratospheric aerosol optical depth (SAOD) estimates. To address this, we compare SAGE III-derived SAOD values with those from multiple instruments, including OMPS-NASA[51–54] (Ozone Mapping and Profiler Suite; NASA aerosol retrieval algorithms), OMPS-SASK[51–54] (Ozone Mapping and

Profiler Suite; University of Saskatchewan aerosol retrieval algorithms), OSIRIS[54] (Optical Spectrograph and Infrared Imaging System), and GloSSAC[54,55] (Global Space-based Stratospheric Aerosol Climatology) (see Supplementary Fig. 2).

We calculated the changes in $SH_2O$, sulfate aerosols, and ozone during the first and second years following the Hunga eruption ("Hunga-2022" and "Hunga-2023") relative to their variations in the pre-eruption period (7 June 2017–9 December 2021; "CLIM") using SAGE-III/ISS observations. Perturbed stratospheric profiles for water vapor and aerosols from the eruption are identified as those with values exceeding two standard deviations above the background climatology. Similarly, ozone perturbations from the eruption are identified using a similar filtering criterion as described by Wilmouth[33] (See Methods). To compare the SAGE-III based perturbations in $SH_2O$ and ozone following the eruption, we used long-term (2005–2023) data from the Aura Microwave Limb Sounder (MLS)[56]. Here, the absolute changes in water vapor and ozone are compared with SAGE-III observations (Supplementary Fig. 3a–d for $SH_2O$ and ozone from MLS[56]; 3e–h for $SH_2O$ and ozone from SAGE-III).

We computed the instantaneous net (shortwave and longwave) atmospheric radiative heating rates and perturbations to the TOA and near the tropopause energy budgets using the LibRadtran[57] radiative transfer model (see Supplementary Fig. 1 and Methods). These energy budget perturbations serve as proxies for the overall atmospheric energy balance, accounting for the contributions of stratospheric water vapor, ozone, and sulfate aerosols. In the following sections, we first examine the individual radiative effects of each perturbed species before analyzing their combined net impact.

## Changes in stratospheric water vapor after Hunga eruption

We show the absolute changes in the zonal-mean latitude-altitude variations of stratospheric $H_2O$ mixing ratio between the (Fig. 1a) "Hunga-2022" and "CLIM" periods and (Fig. 1b) "Hunga-2023" and "CLIM" periods based on the SAGE III/ISS observations.

We observe that the $SH_2O$ injected by the Hunga eruption spread throughout the middle stratosphere from the southern mid-latitudes to the northern tropics during the first year (Fig. 1a). Around the eruption latitude (20°S–0°S latitude), we observed a 2 km thick layer, extending from 27 to 29 km, with a 90% enhancement in $SH_2O$ mixing ratio relative to the background climatology (Fig. 1a and Supplementary Fig. 4a). Further south, from 20°S to 50°S, the increase in stratospheric water vapor mixing ratio was more modest, indicating lower concentrations of water vapor due to its latitudinal transport by stratospheric wind patterns. The near-global $SH_2O$ content map, integrated from the lower to the upper stratosphere (up to an altitude of 45 km), shows a noticeable absolute increase of approximately 0.5–1 g m$^{-2}$ in the SH (Fig. 1c).

In the second year (2023) following the eruption, the injected $SH_2O$ has spread from the southern to the northern hemisphere, covering the middle to the upper stratosphere. The substantial dispersion of $SH_2O$ during the second year has led to a decrease in its mixing ratio and concentration (Fig. 1b, d; see relative changes in $SH_2O$ in Supplementary Fig. 4a, b). This accumulated $SH_2O$ has reduced by around 10% relative to the first year, as observed from SAGE-III (Fig. 1d; Table 1). The widespread vertical distribution of $SH_2O$ in 2023, in contrast to 2022, is attributed to the stratospheric Brewer-Dobson circulation in combination with some perturbations in stratospheric circulation patterns due to stratospheric temperature perturbations following the Hunga eruption[31,32,39].

We also quantify the impact of the 2022 Hunga eruption on the seasonal timeline of changes in $SH_2O$ concentration from 2022 to 2023, compared to its seasonal 5-year climatology value between 2017 and 2021 (Fig. 2a). We observe a distinct increase in $SH_2O$ concentration from January 2022 onwards from SAGE-III observations. The peak in absolute changes in $SH_2O$ concentration value is ~0.5 g m$^{-2}$ (30% higher relative to the background value) in the SH during spring season (MAM) of 2022 (Fig. 2a). The increase in $SH_2O$ gradually decreases in the summer (JJA) and autumn (SON) of 2022 and other seasons of 2023. Note that the decrease in

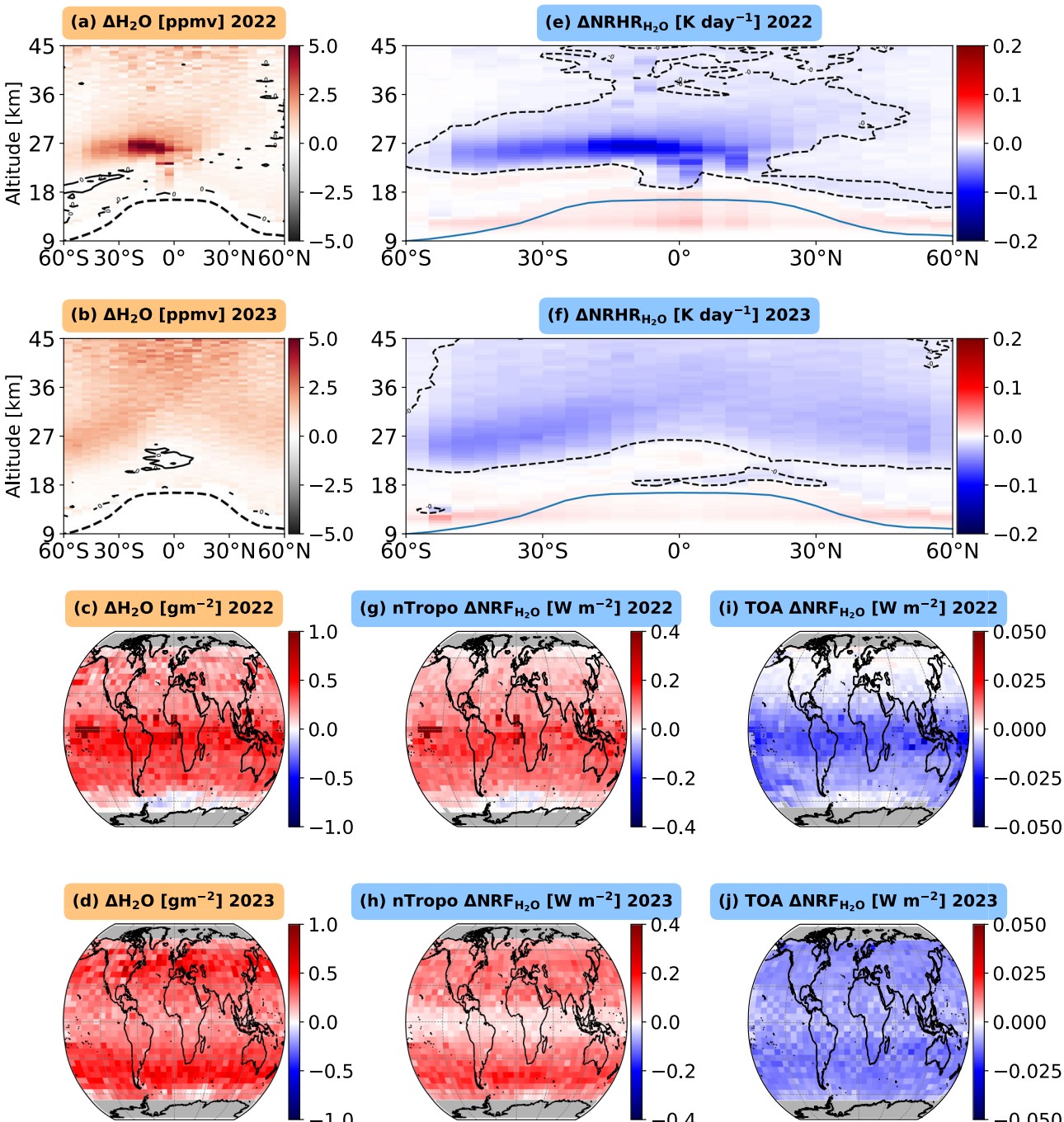

**Fig. 1 | Observed enhancement of stratospheric water vapor and the associated simulated instantaneous radiative forcing during 2022 and 2023 following the Hunga eruption. a, b** show the zonal- and annual-mean latitude–altitude variation in the change of the stratospheric water vapor ($H_2O$) mixing ratio [ppmv] for 2022 and 2023, respectively, relative to the reference climatology (CLIM) period from 2017 to 2021. Contours denote near-zero changes, and the dashed black line represents the tropopause height. **c, d** show the corresponding vertically integrated changes from the lower stratosphere to the upper stratosphere. **e, f** show the latitude–altitude distribution of the difference in net (longwave + shortwave) radiative heating rates (ΔNRHR; K day$^{-1}$) of water vapor within the stratosphere between each post-eruption year and the CLIM period. **g, h** depict changes in near-tropopause net radiative forcing (ΔNRF$_{H_2O}$; W m$^{-2}$) for 2022 and 2023, highlighting perturbations primarily observed near the lowest levels of the lower stratosphere (above the tropopause height). **i, j** show the changes at the top-of-atmosphere (TOA) net radiative forcing (ΔNRF$_{H_2O}$; W m$^{-2}$) for 2022 and 2023, respectively.

SH$_2$O concentration in the SH from 2022 to 2023 is minimal with a decline of around 10–15% (Fig. 2a). The NH and near-global domain also experience a substantial increase in the magnitude of SH$_2$O in 2022 and 2023. The strong vertical transport of tropical middle stratosphere SH$_2$O into the upper stratosphere and towards mid- and high-latitudes regions in 2023 in the SH and NH are attributed to Brewer-Dobson circulation[31,39,44], as evident from the vertical and spatial map of SH$_2$O in Fig. 1a–d.

## Net radiative heating rate and radiative forcing of the Hunga-perturbed SH$_2$O: 2022 versus 2023

We used idealized radiative transfer model simulations (see Methods) to assess the stratospheric radiative heating rates (Fig. 1e, f) and radiative forcing of the perturbed SH$_2$O near the tropopause (Fig. 1g, h) and at the TOA (Fig. 1i, j). In 2022, in the middle stratosphere (24–26 km) around the eruption latitude, the additional SH$_2$O caused a cooling of −0.20 K day$^{-1}$ due

**Table 1 | Properties and hemispherical mean net radiative forcing of radiatively important stratospheric species perturbed by the 2022 Hunga eruption**

| Hunga eruption-perturbed species | SH | | NH | | nGL | |
|---|---|---|---|---|---|---|
| | **2022** | **2023** | **2022** | **2023** | **2022** | **2023** |
| $r_{eff_{HTHH}} [\mu m]$ | 0.30 | 0.29 | 0.26 | 0.27 | 0.28 | 0.28 |
| $MEE_{HTHH} [m^2 g^{-1}]$ | 4.21 | 4.24 | 4.22 | 4.17 | 4.21 | 4.21 |
| $M_{SO_{2HTHH}} [Tg]$ | 0.46 | 0.31 | 0.26 | 0.19 | 0.71 | 0.50 |
| $SAOD_{HTHH} [Unitless]$ | 0.017 | 0.013 | 0.010 | 0.007 | 0.013 | 0.010 |
| | | | | | | |
| $\Delta M_{SO_{2HTHH}} [Tg]$ | 0.35 | 0.18 | 0.13 | 0.06 | 0.48 | 0.24 |
| $\Delta SAOD_{HTHH} [Unitless]$ | 0.014 | 0.007 | 0.005 | 0.002 | 0.008 | 0.005 |
| $\Delta NRF_{SAOD} [Wm^{-2}]$ *(near tropopause)* | **-0.66** | **-0.31** | **-0.18** | **-0.06** | **-0.41** | **-0.19** |
| $\Delta NRF_{SAOD} [Wm^{-2}]$ *(TOA)* | -0.53 | -0.26 | -0.14 | -0.05 | -0.34 | -0.15 |
| | | | | | | |
| $\Delta H_2O_{Total} [Tg]$ | 86 | 72 | 59 | 66 | 145 | 138 |
| $\Delta H_2O [gm^{-2}]$ | 0.37 | 0.32 | 0.24 | 0.30 | 0.32 | 0.31 |
| $\Delta NRF_{H_2O} [Wm^{-2}]$ *(near tropopause)* | **0.12** | **0.09** | **0.08** | **0.08** | **0.10** | **0.08** |
| $\Delta NRF_{H_2O} [Wm^{-2}]$ *(TOA)* | -0.010 | -0.01 | -0.005 | -0.009 | -0.008 | -0.009 |
| | | | | | | |
| $\Delta O_3 [DU]$ (2DF with $H_2O$) | -0.06 | 0.45 | 1.38 | -0.36 | 0.66 | 0.05 |
| $\Delta NRF_{O_3} [Wm^{-2}]$ (2DF; near tropopause) | **0.018** | **-0.028** | **-0.05** | **0.02** | **-0.018** | **-0.004** |
| $\Delta NRF_{O_3} [Wm^{-2}]$ (2DF; TOA) | -0.009 | 0.01 | 0.02 | -0.006 | 0.007 | 0.002 |
| | | | | | | |
| Near tropopause $\Delta NRF_{H_2O+SAOD+O_{3,2DF}}$ | -0.52 | -0.25 | -0.15 | +0.04 | -0.33 | -0.11 |
| At TOA $\Delta NRF_{H_2O+SAOD+O_{3,2DF}}$ | -0.55 | -0.26 | -0.13 | -0.07 | -0.34 | -0.16 |

The blue rows list the mean effective radius ($r_{eff}$; in μm) of retrieved sulfate aerosols, the retrieved vertically integrated SO₂ mass (MSO₂; in Tg), the retrieved mass extinction efficiency (MEE; in m²g⁻¹) per gram of SO₂, and the SAGE-III/ISS observed SAOD for Hunga for the Southern Hemisphere (SH), Northern Hemisphere (NH), and near-global (n-GL) domains for 2022 and 2023. The light orange rows present the retrieved absolute (Hunga–CLIM) change in sulfate aerosols ($\Delta$MSO₂ₕ₂ₙ₉ₐ; Tg), the SAGE-III/ISS observed absolute (Hunga–CLIM) changes in the vertically integrated SAOD ($\Delta NRF_{SAOD}$; W m⁻²) near the tropopause and at the TOA. The dark orange rows show the SAGE-III/ISS observed total injected H₂O mass ($\Delta H_2O$; Tg) and its corresponding simulated mean net radiative forcing ($\Delta NRF_{H_2O}$; W m⁻²) near the tropopause and at the TOA. The light magenta rows display the absolute changes in ozone derived from the 2D-filtered technique ($\Delta O_3$; DU; 2DF with water vapor) and their simulated mean net radiative forcing ($\Delta NRF_{O_3}$; W m⁻²; 2DF) near the tropopause and at the TOA. The final gray rows show the combined net radiative forcing of the three species ($NRF_{H_2O+SAOD+O_{3,2DF}}$; W m⁻²) near the tropopause and at the TOA.

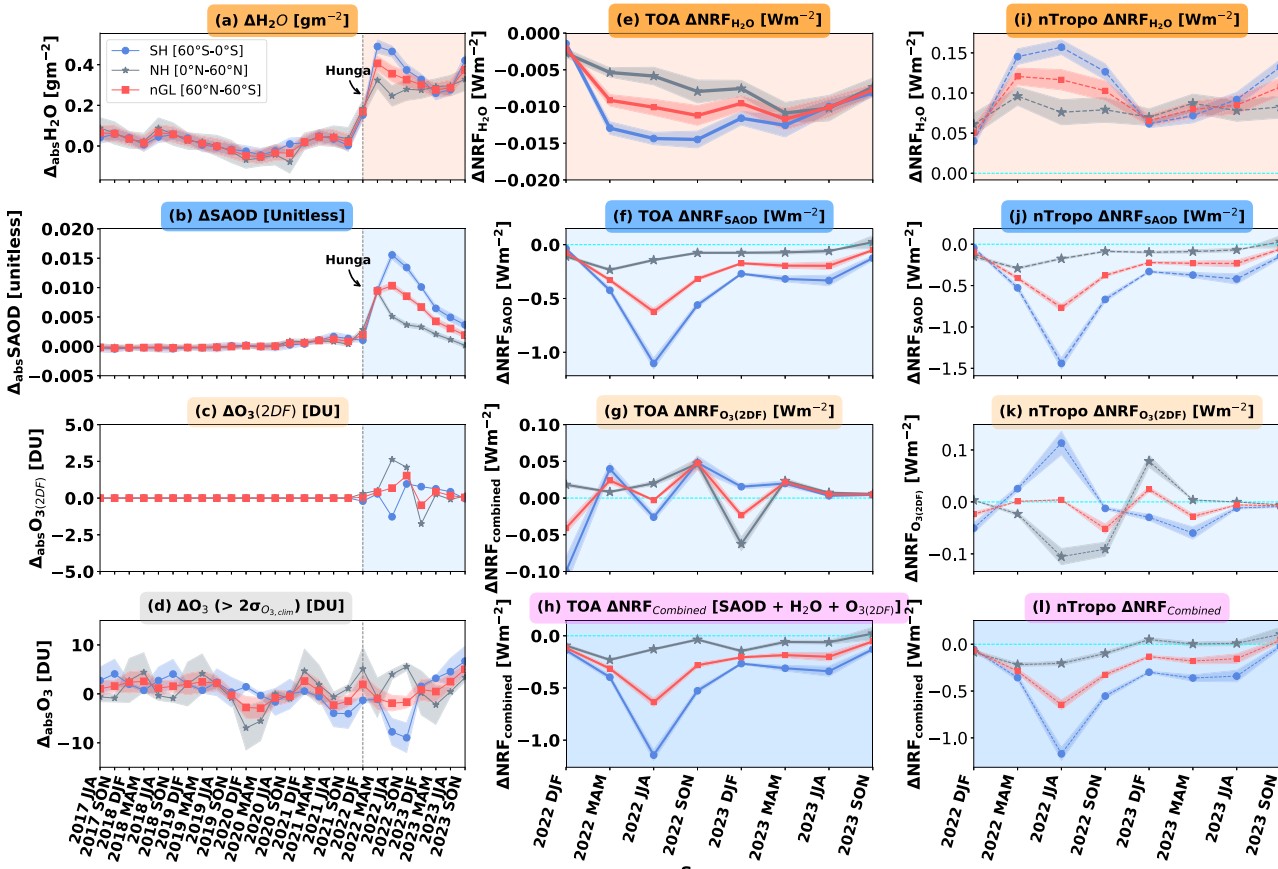

**Fig. 2 | Evolution of the absolute changes and corresponding top-of-atmosphere (TOA) and near-tropopause instantaneous net radiative forcing (ΔNRF) of three radiatively important stratospheric species perturbed by the Hunga eruption.** Show the evolution of the seasonal mean absolute changes (relative to the reference climatology [CLIM] period) from 2017 until 2023 for: **a** stratospheric water vapor (SH₂O) concentration, **b** stratospheric aerosol optical depth (SAOD), **c** ozone concentration (with 2D-filtered condition), and **d** ozone concentration (unfiltered, showing values exceeding 2σ above background climatology, UNF). These values are presented for the Southern Hemisphere (SH; blue line), the Northern Hemisphere (NH; gray line), and near-global (60°S–60°N; nGL; red line). Show the clear-sky net (longwave plus shortwave) radiative forcing (ΔNRF) at TOA due to post-eruption perturbations in: **e** SH₂O, **f** SAOD, **g** ozone (2D-filtered; 2DF), and **h** the combined effect of SH₂O + SAOD + ozone (2D-filtered). **i–l** show the corresponding dashed blue, gray, and red lines indicate the clear-sky ΔNRF near the tropopause. Dashed cyan lines indicate the horizonal bar near zero in (**e–h**) and (**i–l**). Shaded regions represent the interannual standard deviation (1σ) of the background values (except for 2D-filtered ozone, which is extracted above 2σ; see Methods). The perturbations in SH₂O, SAOD, and ozone (2D-filtered) following the eruption exceed their interannual variability, resulting in the combined instantaneous radiative forcing shown in (**l**).

to enhanced emission of longwave radiation (Supplementary Fig. 5a). This cooling was partially offset by heating of up to +0.05 K day⁻¹ from the additional absorption of SW radiation by the SH₂O (Supplementary Fig. 5g). These effects combined to produce a net radiative cooling rate of up to −0.15 K day⁻¹ (dark blue color in Fig. 1e). During 2023, the vertical spreading of the SH₂O layer in both hemispheres exhibited a net cooling effect below −0.1 K day⁻¹ (Fig. 1f).

During 2022 and 2023, the SH₂O perturbation in the middle and upper stratosphere following the eruption substantially affected the clear-sky net instantaneous radiative forcing near the tropopause (Fig. 1g, h) and at the TOA (Fig. 1i, j) relative to background climatology. Enhanced SH₂O, a potent greenhouse gas, substantially increases downward longwave radiation emission from the water vapor layer, contributing to a higher net radiative forcing within Earth's atmosphere while having minimal impact on shortwave radiation (Supplementary Fig. 5g–l). We find that the Hunga eruption's impact on SH₂O in 2022 increased the net radiative flux by +0.12 ± 0.01 Wm⁻² near the tropopause in the SH and +0.10 ± 0.01 Wm⁻² in the near-global domain (Table 1). In contrast, at Earth's TOA, the enhanced SH₂O in 2022 decreased the net radiative flux by ∼ −0.010 ± 0.001 Wm⁻² in the SH and ∼ −0.008 ± 0.001 Wm⁻² in the near-global domain (Table 1). This is due to the enhanced longwave radiation emission from water vapor in the upper stratosphere.

During the summer of 2022, we estimated a peak positive instantaneous net radiative flux of +0.15 ± 0.02 Wm⁻² near the tropopause and a negative instantaneous net radiative flux of −0.014 ± 0.001 Wm⁻² at TOA in the SH, driven by enhanced SH₂O (Fig. 2e, i). In the SH, the increased positive net instantaneous radiative flux of +0.15 ± 0.02 Wm⁻² near the tropopause contributed to a surface warming of +0.015 K ± 0.004 K by the end of 2022, as estimated using the FaIR[58,59] simple two-layer energy balance climate emulator (see Methods; light cyan color in Supplementary Fig. 6e). The increased net instantaneous radiative forcing and its associated uncertainty near the tropopause and at the TOA are calculated solely based on changes in SH₂O, assuming a constant background temperature profile for both pre- and post-eruption periods. Notably, these calculations do not explicitly account for the influence of clouds and temperature adjustments, which may lead to an underestimation of uncertainties.

In 2023, Hunga-perturbed SH₂O increased the net radiative flux near the tropopause by +0.09 ± 0.01 Wm⁻² in the SH and +0.08 ± 0.01 Wm⁻² in the NH. These changes in radiative flux led to a mean surface warming of +0.022 ± 0.003 K in the SH and +0.017 ± 0.002 K in the NH (see blue and cyan colors in Supplementary Fig. 6e). At Earth's TOA, Hunga-perturbed SH₂O decreased the net instantaneous radiative flux by −0.01 ± 0.001 Wm⁻² in the SH and −0.009 ± 0.001 Wm⁻² in the near-global domain.

In aggregate, the $SH_2O$ in the first year (2022) and second year (2023) following the eruption contributed to increased radiative forcing near the tropopause and decreased radiative forcing at Earth's TOA, primarily in the SH, but without a strong longitudinal pattern (Fig. 1c, g, i). In 2023, the widespread dispersion of $SH_2O$ led to tropospheric warming near the tropopause in both hemispheres (Fig. 1d, h).

The spread of stratospheric water vapor is driven by the Brewer–Dobson circulation, a large-scale atmospheric circulation pattern that transports tropical tropospheric air into the stratosphere, followed by its poleward movement and descent at higher latitudes[5,44]. Furthermore, the perturbed water vapor levels may influence this background circulation. A more detailed investigation of these interactions would require the application of advanced climate modeling, which is beyond the scope of this study.

### Changes in stratospheric aerosols after the Hunga eruption

The Hunga eruption injected not only a substantial amount of water vapor but also a moderate quantity of sulfur dioxide ($SO_2$) into the stratosphere, estimated at ~0.4–0.7 Tg[11,12,60]. The total $SO_2$ transported from the troposphere to the stratosphere is estimated to be around 1 Tg[60]. However, compared to previous volcanic eruptions such as Raikoke in 2019 (1.5 Tg[61,62]) and the Pinatubo eruption in 1991 (~17 Tg[63]), the amount of $SO_2$ injected into the stratosphere by the Hunga eruption was considerably smaller.

To quantify the impact of the Hunga eruption on the stratosphere, we identified perturbed aerosol extinction in 2022 and 2023 by detecting deviations exceeding above two standard deviations ($\sigma$) from the background climatology. This method is based on the expectation that, under normal conditions, stratospheric aerosol extinction remains relatively stable; therefore, any significant deviation likely reflects the additional aerosol load injected by the eruption, which in turn influences radiative forcing and stratospheric dynamics[62]. This approach allowed us to assess the three-dimensional distribution of perturbed stratospheric aerosol extinction and the associated SAOD. We observed an enhanced aerosol extinction of solar radiation over 50°S–20°N in the stratosphere at 22–24 km altitude in 2022 following the eruption (Fig. 3a), with a maximum relative increase in aerosol extinction reaching 800% (Supplementary Fig. 4c). The near-global SAOD map, which represents integrated aerosol extinction from the lower to the upper stratosphere (Fig. 3c), reveals a substantial absolute increase of approximately +0.014 ± 0.001 in the SH in 2022 (Fig. 3c; Fig. 2b; Table 1). Similar to $SH_2O$, stratospheric aerosols are predominantly concentrated in the SH, exhibiting no strong longitudinal pattern (Fig. 3c). During the second-year post-eruption (2023), we observe a decrease in 521 nm sulfate aerosol extinction from 0.004 km$^{-1}$ to 0.002 km$^{-1}$ between 50°S and 20°N, at altitudes ranging from 10 km in the southern midlatitudes to 25 km in the tropics (Fig. 3b).

Unlike water vapor, the spread of SAOD is predominantly confined to the SH (Fig. 3c, d). In 2023, perturbed SAOD decreased by around 50% compared to the first-year aerosol perturbation (Fig. 2b; Fig. 3c, g). With minimal aerosol perturbation in the NH, the cooling impact of SAOD in the SH contributes to a hemispheric asymmetry under idealized conditions. The reduction in SAOD observed in 2023 could be attributed to the gravitational settling of stratospheric sulfate aerosols[64,65].

During 2022, following the eruption, we retrieved a mean effective radius of sulfate aerosols of approximately 0.3 μm (with a median of 0.27 μm) in the mid-stratosphere (Table 1 and Supplementary Fig. 7a, e), which is consistent with the findings of Knepp[66]. In 2023, the mean effective radius of sulfate aerosols in the SH was 0.29 μm (Table 1 and Supplementary Fig. 7i,m), nearly identical to that observed in 2022. Boichu[14] and Khaykin[67] showed that during the first two months after the eruption, sulfate aerosol particles were larger; over time, these particles decreased in size, transitioning to finer particles.

We compare the annual mean percentage change in near-global ΔSAOD for 2022, derived from SAGE III, with OMPS-NASA[54], OMPS-SASK[54], OSIRIS[54], and GloSSAC[54] to assess biases (Supplementary Fig. 2; Table 1). Among datasets, OMPS-NASA has the largest ΔSAOD

discrepancy (−66.5%), followed by OMPS-SASK (−37.8%), OSIRIS (−12.6%), and SAGE-III/ISS (−15.5%), while GloSSAC shows a slight increase (+1.8%). Despite inter-instrument biases, GloSSAC estimates align well with our SAOD approach based on >2σ perturbations. These differences highlight the need for multi-instrument comparisons to refine SAOD estimates and radiative forcing assessments post-eruption. OMPS-NASA shows a > 50% high bias due to its fixed aerosol size assumption, while OMPS-SASK mitigates this by accounting for size variations[52].

### Net radiative heating rate and radiative forcing of the Hunga-perturbed stratospheric sulfate aerosol: 2022 versus 2023

We present the near-global distribution of changes in net radiative flux near the tropopause (Fig. 3g) and at the TOA (Fig. 3i) resulting from the perturbation of sulfate-dominated stratospheric aerosols within the lower-to-upper stratosphere. During 2022, when averaged over the SH, our results indicate that the mean net radiative flux at Earth's TOA decreased by 0.53 ± 0.04 W m$^{-2}$, leading to a surface cooling of 0.07 ± 0.01 K, as estimated from the FaIR[58,59] model (Supplementary Fig. 6b). This reduction is attributed to an increase in SAOD (i.e., ΔSAOD) of approximately 0.014 ± 0.001 (Fig. 2b; Fig. 3c; Table 1). Similarly, the mean net radiative flux near the tropopause in the SH decreased by 0.66 ± 0.05 W m$^{-2}$ (Fig. 2b; Fig. 3g; Table 1), resulting in a surface cooling of 0.09 ± 0.01 K (Supplementary Fig. 6f). On a near-global scale, the mean radiative flux changes at the TOA and near the tropopause are estimated to be −0.34 ± 0.03 W m$^{-2}$ and −0.41 ± 0.04 W m$^{-2}$, respectively, associated with a ΔSAOD of around 0.008 ± 0.001 (Fig. 2b, f, j; Fig. 5a; Table 1).

During 2023, the increase in SAOD associated with the Hunga eruption amounted to 0.007 ± 0.001 in the SH, leading to a reduction in net radiative flux at the TOA and near the tropopause by 0.26 ± 0.03 W m$^{-2}$ and 0.31 ± 0.04 W m$^{-2}$, respectively. Consequently, surface temperatures cooled by around 0.11 ± 0.01 K by the end of the year. The system's thermal inertia helped sustain this cooling, similar to the radiative forcing changes observed in 2022.

On a near-global scale, during 2023, an increase in SAOD of 0.005 ± 0.001 resulted in a reduction in net radiative flux of 0.15 ± 0.03 W m$^{-2}$ at the TOA and 0.19 ± 0.04 W m$^{-2}$ near the tropopause (Fig. 2b, f, j; Fig. 5a, b; Table 1).

Using the above results at TOA, we estimate a radiative forcing efficiency (near-global mean radiative forcing at TOA per unit SAOD) of ~25–45 Wm$^{-2}$ during 2022 and 2023. This estimate is relatively higher than those reported by Marshall[68] and Schmidt[69] for other subaerial volcanic eruptions.

We retrieved sulfate aerosol effective radii of ~0.3 μm (with a median radius of 0.27 μm) using Mie theory (see Methods), which indicates that these aerosols scatter incoming solar radiation most efficiently per unit mass[65]. This enhanced scattering efficiency, associated with a particle median radius of 0.27 μm, may have contributed to the higher radiative forcing efficiency observed for sulfate aerosols (Supplementary Fig. 8a, b), highlighting the need for further chemistry-based sensitivity analyses of radiative efficiency in future studies.

Our estimated effective (median) radius of sulfate aerosols has a mean value of 0.3 μm (0.27 μm). These particle sizes result in a high sulfate MEE (Table 1; Supplementary Fig. 7d, h, l, p and Supplementary Fig. 8), thereby enhancing sulfate aerosol forcing. The calculated MEE and its associated standard deviation are 4.21 ± 0.13 m$^2$ g$^{-1}$ (4.24 ± 0.13 m$^2$ g$^{-1}$) in the SH, 4.22 ± 0.17 m$^2$ g$^{-1}$ (4.17 ± 0.17 m$^2$ g$^{-1}$) in the NH, and 4.21 ± 0.25 m$^2$ g$^{-1}$ (4.21 ± 0.25 m$^2$ g$^{-1}$) for the near-global domain during 2022 (2023), respectively (Table 1).

### Changes in stratospheric ozone after the Hunga eruption: 2D-Filtered technique

Following the methodologies of Wilmouth[33] and Santee[35], we assessed changes in the ozone mixing ratio in 2022 and 2023 relative to background mean values using both SAGE III[45] and MLS[56] datasets. To better quantify

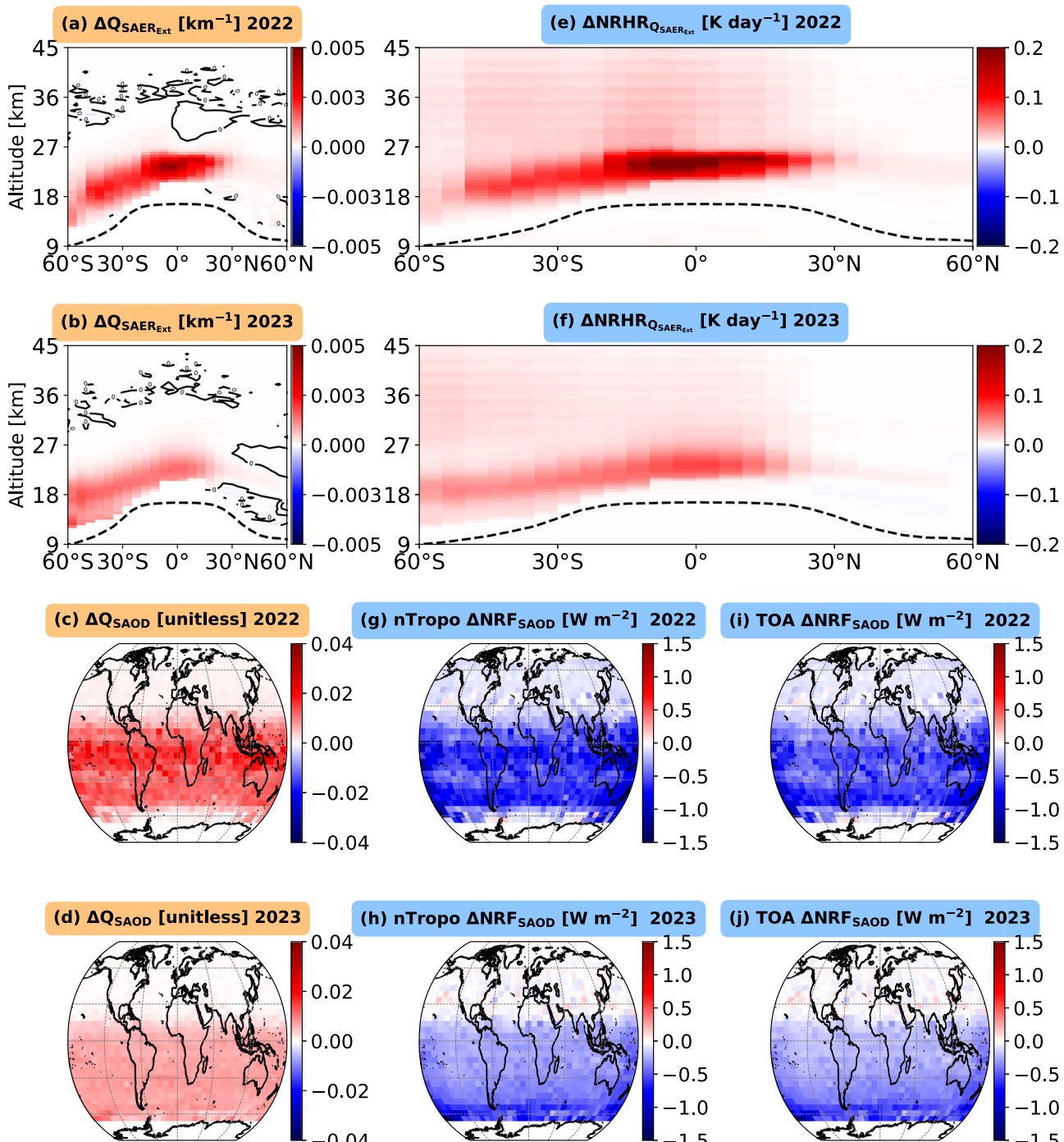

**Fig. 3 | Observed enhancement of stratospheric aerosol and the associated simulated instantaneous radiative effects during 2022 and 2023 following the Hunga eruption. a, b** show the zonal- and annual-mean latitude–altitude variation in the change of the aerosol extinction coefficient at 521 nm ($Q_{SAERext}$; $km^{-1}$) for 2022 and 2023, respectively, relative to the reference climatology (CLIM) period from 07 June 2017 to 09 Dec 2021. Contours denote near-zero changes, and the dashed black line represents the tropopause height. **c, d** display the vertically integrated aerosol extinction coefficient (SAOD) for the perturbed lower to upper stratosphere.

**e, f** show the latitude–altitude distribution of the difference in net (longwave + shortwave) radiative heating rates ($\Delta NRHR$; K $day^{-1}$) of stratospheric aerosol between each post-eruption year and the CLIM period. **g, h** depict the changes in near-tropopause net radiative flux ($\Delta NRF_{SAOD}$; W $m^{-2}$) for 2022 and 2023, respectively, highlighting perturbations primarily observed in the lowest levels of the lower stratosphere (above the tropopause height). **i, j** show the corresponding changes in the top-of-atmosphere (TOA) net radiative flux ($\Delta NRF_{SAOD}$; W $m^{-2}$) for 2022 and 2023, respectively.

the impact of the Hunga eruption, we applied a 2D-filtered (2DF) technique (see Methods) to distinguish changes in the zonal-mean mixing ratio. Using the 2DF technique, the extent of the Hunga eruption's influence on ozone fluctuations becomes discernible, as the observed changes following the eruption exceed the typical year-to-year variability seen before the event (Fig. 2c).

We find a slightly stronger negative ozone mixing ratio anomaly, with values of around −0.5 ppmv at 27 km and −1 ppmv at 32 km in the SH during the summer of 2022 (Fig. 4a; Fig. 2c; Supplementary Fig. 9c). In 2023, we also observe an increase in the ozone mixing ratio anomaly, with values exceeding 0.5 ppmv between 22 km and 36 km altitude near equatorial regions and in the SH (Fig. 4b; Fig. 2c; Supplementary Fig. 9i, j).

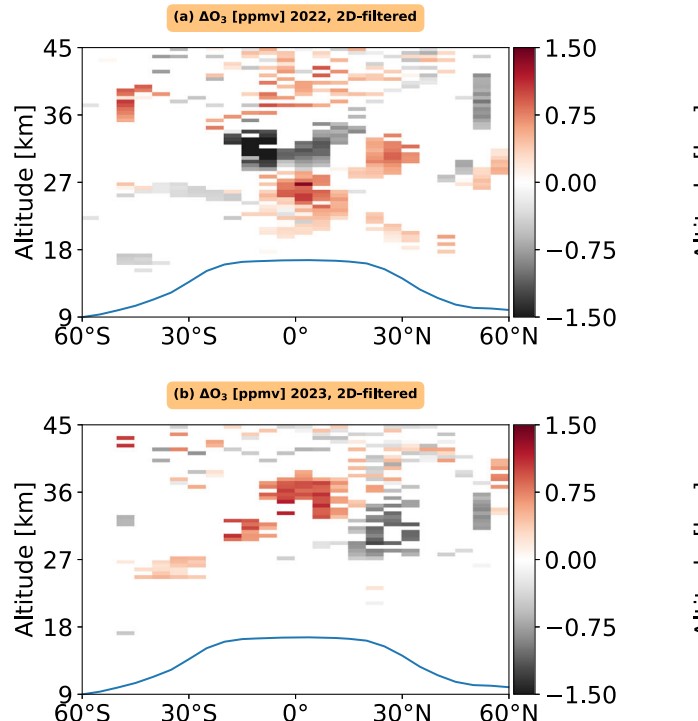

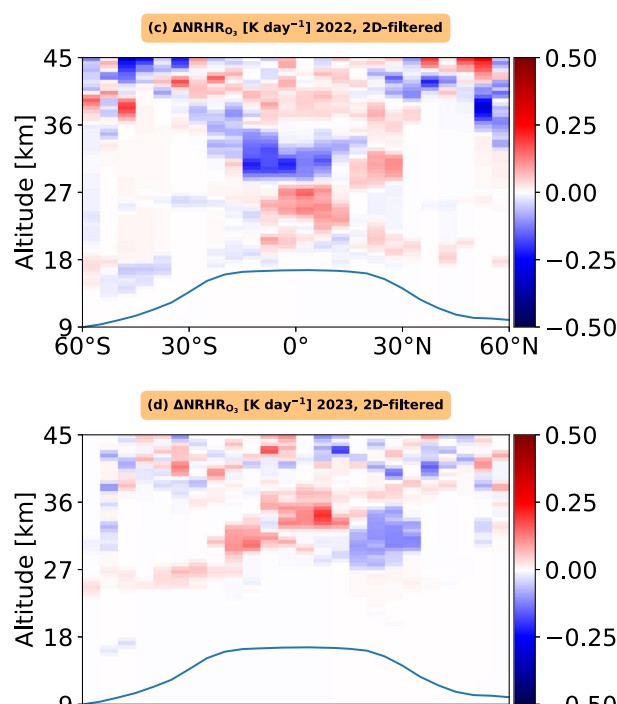

**Fig. 4 | Observed 2D-filtered changes in stratospheric ozone mixing ratios and the associated simulated instantaneous radiative forcing during 2022 and 2023 following the Hunga eruption. a**, **b** show the zonal- and annual-mean latitude–altitude variation of the changes in stratospheric ozone mixing ratios for 2022 and 2023, respectively. **c**, **d** present the latitude–altitude distribution of the differences in net (longwave + shortwave) radiative heating rates (ΔNRHR) for stratospheric ozone for 2022 and 2023, respectively.

Wilmouth[33], Wang[34], and Santee[35] conducted a thorough analysis of various trace gas species and also used global climate model simulations to determine the mechanism behind the potential ozone loss in 2022. The results of their study demonstrate that the eruption impacted ozone by influencing both stratospheric dynamics and ozone chemistry. We note that to exactly pinpoint the physical mechanism related to the effects of the Hunga eruption on ozone gain in 2023, detailed modeling studies coupling stratospheric dynamics and chemistry are needed which is out of the scope of this study. Therefore, our primary results are focused on assessing the influence of radiative forcing of water vapor, SAOD and ozone (2D-Filtered technique).

## Net radiative heating rate and radiative forcing of the Hunga-perturbed stratospheric ozone (2D-Filtered): 2022 versus 2023

Because ozone molecules absorb a substantial amount of solar radiation, a decrease in their concentration would lead to reduced warming within the stratosphere. This, in turn, would decrease the upwelling solar radiation from the uppermost layer of the stratosphere, resulting in a relative reduction in the net radiative flux at Earth's TOA[31]. The opposite effect occurs in the case of ozone gain.

We find that the ozone mixing ratio, derived from the 2D-filtered technique, decreased in the stratosphere over 20°S–40°S between 24–27 km altitude and over 10°S–30°S between 30 and 35 km altitude during the first year after the eruption (Fig. 4a). This ozone loss contributed to stratospheric cooling of more than −0.15 K day$^{-1}$ (Fig. 4c). Additionally, we observed an increase in ozone in the equatorial regions (15°S–15°N) in the lower stratosphere below 27 km (Fig. 4a), leading to stratospheric warming of more than +0.1 K day$^{-1}$ (Fig. 4c). We observed a strong increase in ozone mixing ratio from the middle stratosphere to lower stratosphere in 2023, possibly influenced by dynamical processes associated with the easterly phase of the Quasi-Biennial Oscillation and secondary stratospheric circulation[12,30–34],

contributing to a stratospheric heating rate of approximately +0.15 to +0.2 K day$^{-1}$.

During 2022, we found that the reduction in O$_3$ concentration was primarily located in the SH (Table 1). This decline in stratospheric ozone concentration by −0.06 DU in the SH contributed to a small negative net radiative flux of −0.009 W m$^{-2}$ at the TOA and a positive net radiative flux of +0.018 W m$^{-2}$ near the tropopause (Table 1). During 2023, an increase in stratospheric ozone concentration by +0.45 DU in the SH contributed to a small positive net radiative flux of +0.01 W m$^{-2}$ at the TOA and a negative net radiative flux of −0.028 W m$^{-2}$ near the tropopause (Table 1). During 2022, an increase in ozone concentration by 1.38 DU in the NH resulted in a net positive radiative flux change of +0.02 ± 0.01 W m$^{-2}$ at the TOA and −0.05 ± 0.01 W m$^{-2}$ near the tropopause (Table 1).

## The combined net radiative effects of the three perturbed stratospheric species

After determining the domain-averaged net instantaneous radiative effects of enhanced stratospheric water vapor, stratospheric sulfate aerosol, and ozone using the 2D-filtered technique, we combined these three radiative perturbations to estimate the net TOA radiative forcing and the associated surface temperature changes due to the eruption (Fig. 5; Table 1). Using a simple climate emulator (FaIR)[58,59], we find that the Hunga eruption induced a mean surface cooling effect of approximately −0.10 ± 0.01 K by the end of 2022 (Fig. 5e). This cooling resulted from a combined instantaneous radiative forcing of ∼ −0.55 ± 0.05 W m$^{-2}$ at the TOA, concentrated almost entirely in the SH (Fig. 4c; Table 1). This cooling effect was primarily driven by sulfate aerosols. During 2023, in the SH, the combined effects of SH$_2$O, SAOD, and ozone (O$_3$-2DF) resulted in a TOA radiative forcing of −0.26 ± 0.04 W m$^{-2}$, leading to a cooling of −0.10 ± 0.01 K by the end of 2023 (Fig. 5d). Thus, the Hunga-associated cooling (and any potential future warming) is likely very small compared to natural climate variability and will be difficult to detect observationally.

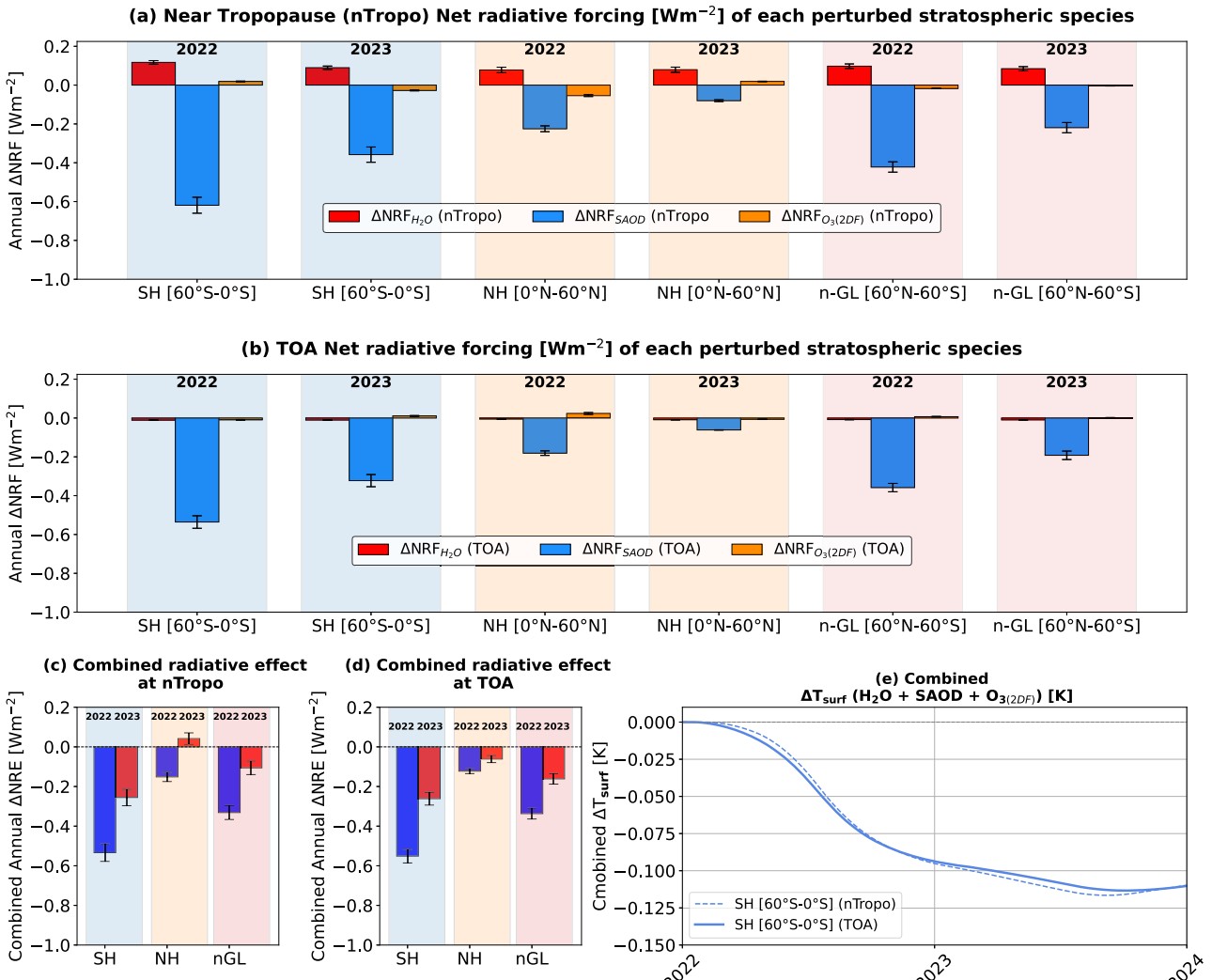

**Fig. 5 | Net radiative forcing during 2022 and 2023 due to three radiatively important stratospheric species perturbed by the Hunga eruption. a** Shows the clear-sky net (longwave + shortwave) instantaneous radiative forcing near the tropopause due to perturbations in stratospheric $H_2O$ (red bars), aerosols (blue bars), and $O_3$ (orange bars) for the Southern Hemisphere (SH), Northern Hemisphere (NH), and near-global (60°S–60°N) regions. **b** Presents the clear-sky net instantaneous radiative forcing at the top-of-the-atmosphere (TOA) for the same species and regions. **c, d** Show the combined $\Delta$NRF near the tropopause and at the TOA, respectively, averaged for the SH, NH, and near-global regions during 2022 and 2023. **e** Shows the surface temperature changes in the SH, estimated using a two-layer energy balance FaIR model based on the perturbed instantaneous energy balance at the TOA and near the tropopause. These estimates represent only a first-order approximation of the actual surface temperature changes. Uncertainty bars (black) in (**a–c**) indicate the background interannual variations in the radiative forcing for $SH_2O$, SAOD, and $O_3$ (black caps).

The stronger surface cooling observed in both 2022 and 2023, despite variations in the magnitude of the combined net radiative flux at the TOA and near the tropopause altitude, suggests a cumulative effect of radiative forcing over time, influenced by thermal inertia and feedback processes. This implies that, although radiative forcing in 2023 was relatively small, the temperature change in 2023 also reflects the residual effects of the stronger forcing from 2022, leading to a similar overall temperature response.

Also, given that the Pinatubo eruption injected approximately 20 Tg of sulfur, resulting in a maximum cooling of about –0.5 K, a simple linear scaling suggests that the Hunga event, which injected less than 1 Tg of sulfur, would induce a cooling effect of approximately –0.025 K[12,19]. Such a small temperature change is likely to be masked by natural variability and may not be directly detectable. Given the modest magnitude of the expected cooling, further investigation using comprehensive climate model simulations is necessary to determine whether these gradients could trigger any measurable atmospheric responses.

In conclusion, our findings indicate that the Hunga eruption did not cause warming in the SH or globally between 2022 and 2023. Instead, it had a cooling effect in the SH. The efficient conversion of $SO_2$ into sulfate aerosols, with effective radii of ~0.3 μm, in a water-rich stratosphere in the SH likely contributed to the net cooling effect observed at Earth's TOA and near the tropopause in 2022 and 2023.

However, in 2023, we find a slight increase in net radiative flux near the tropopause in the NH, primarily due to increased $SH_2O$ and reduced ozone, with minimal perturbation of SAOD. Given that the stratospheric lifetime of $SH_2O$ is much longer than that of SAOD (around 2.5 years[64,65]), this warming trend in the NH may persist and spread over the years until the perturbed $SH_2O$ is fully depleted. Consequently, depending on ozone perturbations, this longer-lasting $SH_2O$ effect could eventually outweigh the cooling effect of SAOD in the SH.

Our findings show that the climate impact of shallow submarine eruptions (water-rich eruptions with a moderate amount of $SO_2$) differs from that of subaerial eruptions (e.g., Pinatubo[19]), depending on the injected altitude and composition of the volcanic plume.

## Methods

This study aimed to assess the radiative effects of the Hunga eruption. To achieve this, we analyzed observed changes in the three main radiatively important species in the stratosphere that were substantially perturbed by the eruption: water vapor, aerosols, and ozone (see Supplementary Information). We then used the observed changes in these three stratospheric species before and after the eruption to simulate the radiative effects of the Hunga eruption using the LibRadtran radiative transfer model (Supplementary Fig. 1).

### Reporting summary

Further information on research design is available in the Nature Portfolio Reporting Summary linked to this article.

## Descriptions of data sets
### SAGE-III/ISS observations

We analyzed SAGE III/ISS data from 7 June 2017 to 31 December 2023, covering latitudes from 60°N to 60°S, depending on variations in solar insolation, solar zenith angle, and the coverage range of the ISS instrument. Specifically, we used solar occultation data from SAGE III on the ISS (SAGE III/ISS; version 5.3, accessed on 28 October 2023; Algorithm Theoretical Basis Document (ATBD)[70,71]), which provided approximately 30 atmospheric profiles per day of $H_2O$, $O_3$, and aerosol extinction coefficient at sunrise and sunset across the globe.

Compared to estimates of stratospheric temperature and composition from passive limb sounding instruments (e.g., Aura Microwave Limb Sounder; MLS), the solar occultation technique of SAGE III/ISS provides much higher vertical resolution measurements. While MLS offers daily near-global observations (82°S–82°N; Supplementary Figs. 11–12), its lower vertical resolution makes it challenging to fully resolve vertical variations in $SH_2O$ and ozone, as well as the associated radiative forcings[47,48]. From the perspective of higher vertical resolution, accurate estimation of atmospheric heating rates and top-of-atmosphere radiative forcing requires detailed profiling of radiatively active species, such as water vapor, ozone, and aerosols[47]. Therefore, in this study, we primarily use multi-wavelength SAGE III/ISS data as input for most of our radiative forcing analysis. However, we also compare our results with other satellite products, including MLS for $H_2O$ and $O_3$ (Supplementary Figs. 3, 11–14) and SAOD measurements from multiple instruments, such as OMPS-SASK, OMPS-NASA, and GloSSAC, to ensure consistency and to assess and highlight uncertainties across different datasets (Supplementary Fig. 2; Table 1).

We analyzed approximately 70,000 profiles over our study period, conducting the analysis on a seasonal basis to ensure sufficient global observations while accounting for temporal changes in solar angle and other variables when calculating the radiative effects of the eruption (Supplementary Methods). We computed the observed changes in $SH_2O$, multi-wavelength aerosol extinction coefficient, and ozone as a function of altitude (binned at 0.5 km), latitude (binned at 5° to ensure robust sampling statistics per bin), and longitude (binned at 10°). Additionally, we estimated the seasonal variations in a near-global map, depicting the vertical integral of water vapor, aerosol extinction coefficient, and ozone concentration from the lower to upper stratosphere.

We divided the background period before the Hunga eruption (CLIM period) into four seasons: DJF (December–January–February), MAM (March–April–May), JJA (June–July–August), and SON (September–October–November), using data from 7 June 2017 to 9 December 2021 (data was unavailable between 10 and 18 December 2021). We included all available data from December 2021 before the initial phase of the Hunga eruption began on 19 December 2021. Similarly, we divided the Hunga eruption period into: Hunga-2022: JF (15 January–28 February 2022), MAM (1 March–31 May 2022), JJA (1 June–31 August 2022), and SON (1 September–30 November 2022). Hunga-2023: DJF (December 2022–28 February 2023), MAM (1 March–31 May 2023), JJA (1 June–31 August 2023), and SON (1 September–30 November 2023).

For our radiative calculations, we applied the nearest-neighbor interpolation technique (K-dimensional tree interpolation) when necessary to achieve complete latitude-longitude coverage between 60°S and 60°N. Finally, we computed the weighted mean, accounting for differences in the number of days across the four seasonal analysis periods during both the Hunga and CLIM periods. This approach enabled us to assess annual-mean changes in the radiation budget, thereby quantifying the annual radiative effect of Hunga-2022 and Hunga-2023.

*Ozone perturbations analysis following the Hunga Eruption: 2D-filtered anomaly* To assess ozone perturbations associated with the Hunga eruption during 2022 and 2023, we employed a structured methodology based on the approach adopted by Wilmouth[33]:

(a) *Evaluation of ozone mixing ratio anomalies exceeding 2σ*: We analyzed the zonal-mean and seasonal-mean ozone ($O_3$) mixing ratio in 2022 and 2023 to identify deviations exceeding ±2 standard deviations from the 2005 to 2021 MLS and 2017 to 2021 SAGE III seasonal means.

(b) *Detection of ozone anomalies exceeding 2σ, synchronized with > 2σ water vapor anomalies:* The Hunga eruption strongly perturbed water vapor, meaning that any ozone changes caused by the eruption should co-occur with water vapor anomalies (Wilmouth[33]). Therefore, regions where water vapor was perturbed served as a reference to determine when and where the eruption also affected ozone.

Following Wilmouth[33], we identified concurrent anomalies in ozone and water vapor. Specifically, we assessed ozone changes as a function of season, latitude bin (5°), and pressure level for MLS (and 0.5 km altitude bin for SAGE III), ensuring that they coincided with water vapor anomalies exceeding >2σ above the MLS 16-year mean and SAGE III 5-year mean.

(c) *Removal of pre-existing anomaly values from 2022 and 2023*: To quantify ozone anomalies caused by the Hunga eruption, we examined whether the 2022–2023 anomalies had occurred in previous years and whether they fell within background variability, independent of the eruption. Using abnormally high water vapor levels in 2022 and 2023 as a reference, we identified and excluded zonal-mean ozone anomaly values (classified by season, latitude, and altitude) that predated these exceptional conditions.

(d) *Application of the 2D-Filtered (2DF) technique:* The 2D-Filtered (2DF) technique was applied to ozone anomalies as a function of season, latitude, and altitude (or pressure level) if they met the criteria from Steps (a) to (c) (Fig. 2c; Fig. 4a, b; Supplementary Figs. 10, 13, 14). When the 2DF technique was applied without water vapor constraints, it was referred to as 2D-Filtered (2DF) without water vapor (see Supplementary Fig. 10). Ozone anomalies that did not meet these criteria were classified as unfiltered conditions (Supplementary Fig. 3e–h). Note that one limitation of this filtering approach is that it may underestimate ozone anomalies influenced by secondary circulation effects that extend beyond the core of the water vapor anomalies.

## Mass extinction efficiency of sulfate aerosol

Mass extinction efficiency (MEE; $m^2 g^{-1}$) of sulfate aerosols indicates how efficiently of aerosols scatter or absorb of solar radiation per unit mass. It can be expressed as:

$$MEE\left[m, 521nm, D_s\right] = \frac{\int_{D_{smin}}^{D_{smax}} \frac{3}{2\rho_s D_s} Q_{ext}\left(D_s, n, k, 521nm\right) n_M\left(D_s\right) dD_s}{\int_{D_{smin}}^{D_{smax}} n_M\left(D_s\right) dD_s}$$

$$(1)$$

The mass size distribution, $n_M\left(D_s\right)$, is given by:

$$n_M\left(D_s\right) = \frac{\pi}{6}\rho_s D_s^3 n\left(D_s\right)$$

where $n\left(D_s\right)$ represents the log-normal number size distribution at geometric standard deviation of 1.2. $\rho_s$ is the $SO_2$ density. Based on the retrieved effective radius ($r_{eff}$) of sulfate aerosol ($D_s = 2r_{eff}$) and the complex refractive index of sulfate aerosol at 521 nm (real part: $n = 1.431$; imaginary part:

$k = 1.0 \times 10^{-8}$), we determined the extinction efficiency factor ($Q_{ext}$ at 521 nm) using Mie theory (Supplementary Fig. 7, 8, 11). Details of the expression of $Q_{ext}(r_{eff}, n, k, 521nm)$ are provided in the Supplementary Methods. The refractive index of sulfate aerosol is obtained from GEISA database.

The hemispherical mean value of MEE is also estimated using:

$$MEE = \frac{SAOD * Surface\ area\left[m^2\right]}{mass\ of\ SO_2\left[g\right] * \left(\frac{\frac{Molar\ Mass\ of\ H_2SO_4}{Molar\ mass\ of\ SO_2} * 1}{wt}\right)} \quad (2)$$

Here, wt is the 75% $H_2SO_4$ weight.

## Relative and absolute changes of three stratospheric species: Hunga versus CLIM

We determined the relative and absolute changes in the three stratospheric species ($Y$) for the Hunga and CLIM periods using:

$$Absolute\ change = \triangle Y = Y[HTHH] - Y[CLIM] \quad (3)$$

$$\%\ Relative\ change = 100 \times \frac{\triangle Y}{Y[CLIM]} \quad (4)$$

## Retrieval of optical and microphysical properties of stratospheric sulfate aerosol: Hunga versus CLIM

We utilized near-global coverage of the spatial and temporal distribution of aerosol extinction [km$^{-1}$] (SAGE-III; ATBD, 2002) at nine wavelengths (384.224 nm, 448.511 nm, 520.513 nm, 601.583 nm, 676.037 nm, 755.979 nm, 869.178 nm, 1021.20 nm, and 1543.92 nm). It is important to note that uncertainty in aerosol extinction varies with wavelength[70]. For instance, at 407 nm and 1089 nm, the uncertainty can exceed 25%, whereas at 450 nm, it decreases to 10%, and at 521 nm, it is less than 5%. To maximize accuracy, we jointly utilized all nine wavelengths to retrieve stratospheric sulfate aerosol optical and microphysical properties using a pre-calculated look-up table (LUT) based on Mie theory[66,71–74].

## Mie-LUT based retrieval of stratospheric sulfate aerosol

An accurate assessment of the size distribution of stratospheric aerosols is crucial for accurately quantifying their radiative effects before and after the eruption. When analyzing background stratospheric aerosols, studies have shown that their size distribution can typically be characterized by a unimodal lognormal distribution[75]. These aerosols are predominantly spherical and consist of approximately 75% sulfuric acid and 25% water vapor[68–73].

Ground-based stratospheric aerosol observations from AERONET indicate that during the early stages of aerosol growth following the Hunga eruption, bi-modal and tri-modal log-normal size distributions better represented the aerosol characteristics, particularly the coarse mode of aerosol particles[13,14]. However, within a few months after the eruption, the data showed that fine-mode stratospheric aerosols became dominant in aerosol extinction[13,67]. Thus, we adopted a unimodal log-normal size distribution to represent the size distribution of sulfate aerosols resulting from the Hunga eruption, with a geometric standard deviation of 1.2. This is broadly consistent with the more comprehensive size distribution analysis by Knepp[66], Khaykin[67], and Duchamp[13]. This size distribution parameterization has previously been used to assess stratospheric sulfate aerosol growth one year after the Pinatubo eruption[19]. Notably, Knepp[66] introduced a novel method based on SAGE III data, indicating a spectral width of 1.2 to 1.3 post-Hunga eruption, which aligns with our choice of 1.2 for the geometric standard deviation in the Hunga case[66,67].

We used the Mie LUT[72,73] to retrieve the optical and microphysical properties of sulfate aerosols[76,77] from SAGE III/ISS stratospheric aerosol extinction profiles at nine wavelengths, covering the period from 7 June 2017 to 31 December 2023 (Hunga−2022 vs. CLIM and Hunga-2023 vs. CLIM periods). For this purpose, we followed these steps:

1. We calculate the aerosol extinction ratios at 521 nm as: 384.224/520.513 nm, 448.511/520.513 nm, 520.513/520.513 nm, 601.583/520.513 nm, 676.037/520.513 nm, 755.979/520.513 nm, 869.178/520.513 nm, 1021.20/520.513 nm, and 1543.92/520.513 nm) using SAGE-III/ISS observations.
2. We compare the SAGE-III aerosol extinction ratios (from point 1) with the Mie LUT aerosol extinction ratios and determine the closest matching extinction ratios at each given wavelength.
3. We calculate the squared difference between the eight aerosol extinction ratios from SAGE-III/ISS and the Mie LUT for each effective radius, then sum these differences to quantify the discrepancy between the two datasets.

Here, Eq. (5) summarizes above step 3:

$$p\left(r_{eff}\right) = \sum_{\lambda_i}\left[Q_{ratio_{SAGE}}\left(\lambda_i\right) - Q_{ratio_{LUT}}\left(\lambda_i, r_{eff}\right)\right]^2 \quad (5)$$

where $Q_{ratio,SAGE}$ and $Q_{ratio_{LUT}}$ represent the extinction ratio at 521 nm from SAGE-III observations and LUT, respectively. Here $\lambda_i$ sums over the eight wavelengths from 400 to 1450 nm (as listed above), determined for both SAGE-III and LUT table using interpolation techniques.

4. The minimum sum of squared differences represents the best solution for the effective radius, asymmetry factor, and single scattering albedo. This step identifies the closest match between the observed and modeled data.

## Simulations of radiative effects of three stratospheric species (Hunga versus CLIM periods) | Columnar- & Zonal-mean

We utilized observational data from multiple instruments (SAGE III/ISS and Aura MLS) to obtain atmospheric and aerosol extinction profiles. Specifically, we retrieved vertical profiles of temperature, pressure, density, water vapor ($H_2O$), ozone ($O_3$), and multi-wavelength aerosol extinction from SAGE III/ISS observations between 9 km and 100 km altitude. For atmospheric profiles below 9 km, we incorporated data from other source, such as ERA5[56,78].

Using the input data described above, we conducted idealized radiative transfer simulations to examine the 3D distribution of stratospheric radiative heating rates and instantaneous radiative fluxes between two periods: Hunga-2022 and Hunga-2023 versus the CLIM period, at the TOA and near the tropopause. A summary of these data sources and radiative transfer model simulations is provided in Supplementary Fig. 1. Using the LibRadtran model with the DISORT solver[57,79,80] and eight streams, we simulated radiative flux changes and associated heating rates by incorporating observed seasonal-mean vertical profiles of atmospheric constituents, aerosol properties, and seasonal-mean daily-insolation weighted solar zenith angle as a function of latitude and longitude. These simulations were conducted separately for the shortwave (0.28–4 μm) and longwave (4.0–100 μm) components of radiative flux changes and heating rates (see Supplementary Methods). We used the CLIM period to define the atmospheric temperature profiles (per season) and applied these fixed temperature profiles for the Hunga period analysis. Thus, our radiative calculations do not account for any temperature adjustments in the stratosphere in the year following the eruption due to net heating/cooling from different radiative species. This analysis design allows us to clearly isolate the instantaneous radiative response of the Hunga eruption from eruption-associated climate feedbacks (e.g., dynamical changes) that influence the observed stratospheric and tropospheric temperature profiles over the past year.

## Uncertainty analysis related to three stratospheric species

We estimated seasonal and yearly standard deviation values for three stratospheric species ($H_2O$, sulfate aerosol, and $O_3$) using SAGE III/ISS observations from 2017 to 2021, before the Hunga eruption. Based on these yearly standard deviation values of the perturbed stratospheric species, we calculated the corresponding standard deviation associated with their instantaneous radiative effects at the TOA and near the tropopause (Fig. 2). Additionally, we determined the propagated standard deviation for the combined radiative effects of the three stratospheric species (Fig. 2). It is important to note that the standard deviation associated with surface temperature changes, estimated using the FaIR model[58,59], is derived from the propagated standard deviation of the combined instantaneous radiative effects of the three stratospheric species[66]. We presented the standard deviation values rounded to two and three places. Other sources of uncertainty, such as dynamical variations, cloud effects, and biases across different satellite products, may also influence temperature changes. However, these factors are beyond the scope of this study and are therefore not included in our uncertainty estimate[66].

## Limitations of radiative transfer model

Even though we directly use the observed vertical profiles of aerosols, stratospheric water vapor, and ozone as inputs to the LibRadtran model, our results are subject to several important limitations.

First, the infrequent sampling of aerosol extinction profiles at multiple wavelengths by SAGE III introduces uncertainties in SAOD retrievals, particularly in the lower stratosphere following the Hunga eruption. Thus, cross-instrument comparisons are critical for improving confidence in SAOD estimates and reducing uncertainties in radiative forcing assessments following major volcanic events (Supplementary Fig. 2 and Supplementary Table 1).

Second, the high signal-to-noise ratio SAGE-III/ISS dataset we used provides only dawn-dusk solar stratospheric profiles. Consequently, we cannot capture the diurnal variations of radiative fluxes in our calculations, which are limited to daytime periods (i.e., no diurnal variability is considered). However, we anticipate that this limitation does not significantly affect our results, as the impact of diurnal temperature and water vapor variations in the stratosphere is relatively small[81–84].

Third, for computational simplicity, we assume a fixed, idealized clear-sky surface albedo of 0.15 between 60°N and 60°S. This assumption may influence the net magnitude of the cooling effect, which could be even stronger if surface albedo were higher. However, the overall sign of the radiative effect remains the same even if the clear-sky surface albedo is assumed to be 0.2 over the same latitude range.

Fourth, our idealized radiation-only simulations do not account for the dynamical effects of radiative components. Additionally, we do not incorporate measured temperature profiles from the Hunga eruption period to estimate radiative forcing under a relaxed stratospheric state (i.e., effective radiative forcing). Notably, Schoeberl[12] accounted for temperature anomalies and found that while they are not always as pronounced as those associated with water vapor or aerosols, they are non-negligible. To illustrate the potential impact of stratospheric temperature anomalies (cooling), we provide idealized radiative forcing column calculations in Supplementary Figs. 15–19, showing that these anomalies primarily affect longwave radiation. When accounting for stratospheric temperature changes, we find that the radiative forcing at both the TOA and near the tropopause altitude due to stratospheric water vapor remains positive. Furthermore, surface albedo should ideally be treated as a dynamic variable, varying with surface type, land use, and seasonal changes. Nonetheless, despite these simplifications, our primary conclusion—that sulfate aerosols dominate the stratospheric radiative cooling effect among the three species considered—remains robust.

## The Finite-amplitude Impulse-Response (FaIR) model

We use the Finite-Amplitude Impulse-Response (FaIR) model, an emulator of a two-layer energy balance system, to estimate the first-order temperature response to the Hunga eruption[58,59]. In this model, "finite-amplitude" refers to the response magnitude being determined by the forcing strength, while "impulse-response" describes the system's reaction to a sudden forcing change. FaIR is applied to assess the temperature response to Hunga-induced radiative forcing from stratospheric water vapor, sulfate aerosols, and ozone. The model provides an estimate of the resulting temperature change, particularly in the SH and NH. However, as FaIR primarily simulates global and hemispheric-scale surface temperature changes, it does not fully account for regional variability or all Earth system feedbacks.

## Data availability

The SAGE-III/ISS v053 solar datasets (filename: g3bssp_53) are freely available from (https://asdc.larc.nasa.gov/project/SAGE%20III-ISS/g3bssp_53). The Aura MLS dataset is also freely available and can be obtained using https://search.earthdata.nasa.gov/. The ECMWF ERA5 reanalysis pressure level datasets can be obtained from https://cds.climate.copernicus.eu/cdsapp#!/search?type=dataset. Underlying data (Supplementary Data 1-5) related to the manuscript is publicly available at Zenodo website (https://zenodo.org/records/14955808; Gupta[85]). Note that only the supplementary data within 60°N to 60°S (see Supplementary Data; https://zenodo.org/records/14955808; Gupta[85]) are used in this study. The data related to the refractive index of sulfate aerosols is taken from https://geisa.aeris-data.fr/litms/. The Mie table for sulfate aerosol is available here: https://github.com/matthew2e/easy-volcanic-aerosol/blob/master/eva_Mie_lookuptables.nc[72].

## Code availability

The LibRadtran model is available at http://www.libradtran.org/doku.php?id=download here. All the Figures (including Supplementary Figs.) were originally produced and plotted using various open-source Python libraries (e.g., https://matplotlib.org/stable/). The LibRadtran model-based processed data and input is also publicly available at the Zenodo (https://zenodo.org/records/14955808; Gupta[85]).

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

## Acknowledgements

We thank Ralph Kahn for his constructive feedback and comments during scientific meetings. We also thank John Rausch for his assistance in obtaining the dataset and for his valuable discussions. This work was supported by National Aeronautics and Space Administration (NASA) grant 80NSSC20K1450 awarded to the PIs K.E.F. and R.B. A.G. received partial support from this funding during his postdoctoral tenure at Vanderbilt, and additional support from NSF grant 2151093 awarded to J.F.K. at UCLA during his current tenure at AOS, UCLA. J.F.K. was supported by grants from the Simons Foundation (SFI-MPS-SRM-00005221) and from the National Science Foundation's Directorate for Geosciences (2151093). T.M. was partially supported by the Packard Fellowship for Science and Engineering award. The scope of these awards addresses submarine volcanism and the radiative effects of aerosols. We also thank the SAGE-III/ISS, LibRadtran, GEISA, and NASA teams for providing access to their data and model archives.

## Author contributions

A.G. contributed to the conceptual development of this work, conducted the analysis, and produced all figures and results. R.B. conducted the FaIR model simulation. A.G. drafted the original manuscript, which was reviewed by T.M., J.F.K., R.B., and K.E.F.

## Competing interests

The authors declare no competing interests.
