## [Transparent Peer Review file · Communications Earth & Environment]

The January 2022 Hunga eruption cooled the southern hemisphere in 2022 and 2023

Corresponding Author: Dr Ashok Gupta

Version 0:

Decision Letter:

Dear Dr Gupta,

Your manuscript titled "The January 2022 Hunga eruption cooled the southern hemisphere in 2022 and 2023" has now been seen by 3 reviewers, and we include their comments at the end of this message. They find your work of interest, but some important points are raised. We are interested in the possibility of publishing your study in Communications Earth & Environment, but would like to consider your responses to these concerns and assess a revised manuscript before we make a final decision on publication. Specifically, we ask you to:

1. Compellingly demonstrate that the sulfate aerosols produced a large cooling following the 2022 Hunga submarine eruption.
2. Compellingly demonstrate the robustness of your ozone calculations and the deviations in ozone following the Hunga eruption compared to previous observations and provide estimates of uncertainties.
3. Fully explain and justify the use of SAGE over OMPS-LP aerosols for SAOD and MLS for ozone, temperature, and water vapor.

We therefore invite you to revise and resubmit your manuscript, along with a point-by-point response that takes into account the points raised. Please highlight all changes in the manuscript text file.

Please use the following link to submit your revised manuscript, point-by-point response to the referees' comments (which should be in a separate document to any cover letter), a tracked-changes version of the manuscript (as a PDF file) and the completed checklist:

Link Redacted

We hope to receive your revised paper within six weeks; please let us know if you aren't able to submit it within this time so that we can discuss how best to proceed. If we don't hear from you, and the revision process takes significantly longer, we may close your file. In this event, we will still be happy to reconsider your paper at a later date, as long as nothing similar has been accepted for publication at Communications Earth & Environment or published elsewhere in the meantime.

Please do not hesitate to contact us if you have any questions or would like to discuss these revisions further. We look forward to seeing the revised manuscript and thank you for the opportunity to review your work.

Best regards,

Dr Alireza Bahadori
Associate Editor
Communications Earth & Environment

EDITORIAL POLICIES AND FORMATTING

Editorial Policy: [Policy requirements](https://www.nature.com/documents/nr-editorial-policy-checklist.pdf) (Download the link to your computer as a PDF.)

- Behavioural and social science
- Ecological, evolutionary & environmental sciences
- Life sciences

<https://www.nature.com/documents/nr-reporting-summary.zip>

Furthermore, please align your manuscript with our format requirements, which are summarized on the following checklist: [Communications Earth & Environment formatting checklist](https://www.nature.com/documents/commsj-phys-style-formatting-checklist-article.pdf)

and also in our style and formatting guide [Communications Earth & Environment formatting guide](https://www.nature.com/documents/commsj-phys-style-formatting-guide-accept.pdf) .

*** DATA: Communications Earth & Environment endorses the principles of the Enabling FAIR data project (<http://www.copdess.org/enabling-fair-data-project/>). We ask authors to make the data that support their conclusions available in permanent, publically accessible data repositories. (Please contact the editor if you are unable to make your data available).

All Communications Earth & Environment manuscripts must include a section titled "Data Availability" at the end of the Methods section or main text (if no Methods). More information on this policy, is available at <http://www.nature.com/authors/policies/data/data-availability-statements-data-citations.pdf>.

If a community resource is unavailable, data can be submitted to generalist repositories such as [figshare](https://figshare.com/) or [Dryad Digital Repository](http://datadryad.org/). Please provide a unique identifier for the data (for example a DOI or a permanent URL) in the data availability statement, if possible. If the repository does not provide identifiers, we encourage authors to supply the search terms that will return the data. For data that have been obtained from publically available sources, please provide a URL and the specific data product name in the data availability statement. Data with a DOI should be further cited in the methods reference section.

REVIEWER COMMENTS:

Reviewer #1 (Remarks to the Author):

General comments:

The paper tackles an important scientific question about the climate impact of the 2022 Hunga eruption. The methodological approach is sound. The authors use SAGE observations to calculate the radiative forcing of volcanic aerosols, stratospheric water vapor, and ozone, coming to a plausible, robust conclusion about net cooling caused by the Hunga eruption. The magnitude of the forcing sounds reasonable, although, at least for water vapor, it is overestimated as, in the way it is calculated, it includes climate temperature variability that is not estimated in the paper. The authors analyze for two post-eruption years. This is not the first attempt. There are at least two similar in a pipeline for publication (<https://essopenarchive.org/users/523044/articles/741323-evolution-of-the-climate-forcing-during-the-two-years-after-the-hunga-tonga-hunga-ha-apai-eruption> and <https://essopenarchive.org/doi/full/10.22541/essoar.169091894.48592907>)

Among the results, I'm afraid I have to disagree with the explanation of why the Hunga eruption is more effective in generating SAOD. With recent estimates, which push the initial emission of SO₂ to 1.5 Mt, it may not be that dramatic. The paper is well-written and logically organized. However, it uses unconventional terminology in some places. The authors must better explain their choice of meteorological profiles in control and perturbed cases. They say about this in supplemental, but very briefly. SAGE itself uses meteorological profiles from a reanalysis for its retrievals. Combining vertical profiles from different pieces may be unnecessary.

Specific comments:

L119: As I mentioned above, some similar studies are in the pipeline for publication.

L123-126: This should go to conclusions.

L135: It is quite a short climatology. In addition, the post-eruption anomalies will include variability associated with the 2022-2024 period. The impact on a long-wave (LW) forcing might be significant.

L169: Not by zonal wind.

L209: It is mixing ratio, not concentration.

L229: You calculate not instantaneous but adjusted radiative forcing as you account for temperature change between the climatology and the 2022-2024 period. You also have to mention that all forcings are clear-sky.

Fig2: The figure caption is mixed up. Please correct.

L271: It isn't very clear how to compare this with the background in this way.

L298: A weak forcing cannot produce a strong interhemispheric asymmetry.

L309: Wet in the stratosphere?

L313: "underestimated"

L365-366: MEE is per aerosol mass. Figure 6 in supplemental shows MEE for 521 nm. It will be much flatter if you average over the entire solar spectrum. Half the solar energy comes in the near IR band between 750 and 4000 nm. Also, your effective radius for the Hunga eruption is relatively tiny compared to other estimates. Equation (1) in the supplement is miswritten. Please correct.

L589: MEE of SO₂ sounds confusing.

L595: "SO₂ aerosol density"?

L599: Remove "the"

L606: It is 75% of H₂SO₄ solution.

L632: I doubt the ground-based AERONET station could see the stratospheric effects well, as they sense through the denser tropospheric aerosol layer.

L681: Discuss the meteorological profiles here.

L688-694: What was the spectral resolution?

L714: SAGE also takes meteorological profiles from a reanalysis

L722: I believe you calculate adjusted (not instantaneous) radiative forcing. Please make it clear.

Reviewer #2 (Remarks to the Author):

The Hunga volcanic eruption was unprecedented in its impact on the stratosphere, and studies on this topic represent important and appropriate content for Communications Earth & Environment. This paper focuses on the radiative impact of changes to water vapor, aerosol, and ozone in the two years following the Hunga volcanic eruption and the associated changes in temperature. I would support the publication of this paper upon consideration of the suggested changes below.

*Major Comments:

1) The reading of the paper could be substantially improved by having the figure ordering match the order in which the discussion proceeds in the text. The paper jumps around referencing panels from Figs 1, 2, 3, and 4 often out of order and requiring the reader to view individual panels from 2 or 3 different figures simultaneously. Why not regroup the figure panels, so that they match the order in which the paper is written? It would substantially improve the readability of the paper. On a related note, I would suggest saving the discussion of Figure 4 until after the discussion of Figures 1-3 is complete rather than calling Fig 4 from the earlier SWV and aerosol sections without sufficient explanation.

2) The authors need to decide if their primary conclusion is that Hunga cooled the SH or cooled the planet. The title and abstract are limited to the SH. But the main paper, e.g., Lines 124-125 and Line 470 take a stronger position and say that the planet was cooled.

3) The discussion of ozone in the manuscript needs revision. The statements on lines 149-152, 245, 389-391, and 403-404 (as well as Supplemental lines 61-62) are misleading. MLS data unequivocally show that for certain latitudes and altitudes, there were deviations in ozone following the Hunga eruption outside the range of any previous observations in the satellite data record.

See for example Figure 6 and Supplemental Figure S10 in Wilmouth et al. (your reference 28), Figure 8 in Wang et al. (Stratospheric climate anomalies and ozone loss caused by the Hunga Tonga-Hunga Ha'apai volcanic eruption. *J. Geophys. Res.: Atmos.*, 128, e2023JD039480, 2023), and Figure 1 in Zhang et al. (Chemistry contribution to stratospheric ozone depletion after the unprecedented water-rich Hunga Tonga eruption. *Geophys. Res. Lett.*, 51, e2023GL105762, 2024). As shown in these papers, the observed ozone losses at certain SH latitudes and altitudes in the second half of 2022 were unprecedented and well outside the range of normal variability, contrary to what is currently stated in the Gupta et al. manuscript.

I would speculate that perhaps the reason Figure 2 in this paper doesn't see the unprecedented ozone loss more clearly is because such a broad latitude range (0-60 S) is being averaged, so the strong midlatitude loss in the second half of 2022 is diluted by averaging in the tropics. Or perhaps there is some other issue with the analysis in this manuscript. Regardless of the reason, this manuscript should be careful not to make overly dismissive statements regarding the observed ozone reductions following Hunga, because the ozone losses were unprecedented in the 20-year MLS record for certain latitudes, altitudes, and times.

Also, please double check the manuscript and supplemental to clean up the confusion as to which cited references studied ozone depletion in the immediate days/weeks after the eruption and which studied ozone depletion in the months/year that followed the eruption.

4) Lines 312-330 and Lines 355-386. I believe it is already well established that the excess water vapor from Hunga is what led to significantly enhanced stratospheric aerosols relative to what might be expected given the small amount of SO₂ injected. Several papers have discussed this. Just a couple off the top of my head:

Y. Zhu et al., Perturbations in stratospheric aerosol evolution due to the water-rich plume of the 2022 Hunga-Tonga eruption. *Commun. Earth Environ.* 3, 248 (2022).

Asher, E., et al., Unexpectedly rapid aerosol formation in the Hunga Tonga plume. *Proc. Nat. Acad. Sci.*, 120, e2219547120 (2023).

5) Lines 780-784. The choice of a 20 km cutoff is fine for water and ozone, but it almost certainly leads to loss of data for the aerosols, which are known to have descended relative to the water plume. Other papers have shown this, as does Figure 1e and 1f in this manuscript. When the radiative forcing results are presented, the manuscript should acknowledge that the results of this study will be biased because of the 20-km altitude cutoff and attempt to estimate the uncertainty introduced.

*Additional comments:

Line 42. Add "volcanic" after "submarine"

Line 43. Delete ", which warms the climate,". It interrupts the flow of the sentence and is unnecessary because this point is

repeated in the next sentence.

Lines 101-103. The sentence beginning with “If water vapor’s warming effects...” should be revised. It is an overstatement as written. The climate impact of any future submarine eruptions will depend on the relative amount of water and SO₂. Not all submarine eruptions are the same.

Line 108. Don’t start paragraph with “However”.

Lines 135-136: Why define CLIM as “7 June 2017 – 9 Dec 2021”? The dates seem completely random. Please explain this choice in the paper. Most studies I’ve seen go back much farther than 2017 to establish the background.

Line 161. Change HTHH-2022 to HTHH-2023.

Line 183. Change December 2022 to January 2022.

Line 191. Reference 28 (Wilmouth et al., 2023) could be added here, as it shows the importance of BDC on water vapor distribution from Hunga.

Line 209 and other places. Don’t say “concentration” when the units are ppmv or ppbv. Change to “mixing ratio”.

Figure 2c. Units are missing on y axis.

Figure 2e. Please change the x axis to make it less cluttered and easier to read.

Line 305. known to produce

Line 428. all two?

Lines 505-796. The Methods section is extraordinarily long. I would recommend moving most of this content to the Supplemental. I’m guessing the journal will require this.

*A few comments on the Supplemental:

Line 64. “However, there is no clear signal afterwards.” This statement is false. The largest ozone losses were observed in the June – December 2022 timeframe. See the papers highlighted in major comment #3 above.

Line 155. NO₂ decreases, not increases.

Figure S2 is essentially unreadable on a print copy. The panels are too small. Also, some of the panels are never listed in the caption.

Reviewer #3 (Remarks to the Author):

Review of “The January 2022 Hunga eruption cooled the southern hemisphere in 2022 and 2023” by Gupta et al.

Major Points

The authors argue that they can neglect ozone changes which their own results suggest is a non-negligible component of the climate forcing. Another recent study indicates that ozone change is an important component of the total flux change – and the authors own table indicate this. The authors need to clarify their discussion of the ozone change and they may have not considered the ozone response correctly.

The FAIR parameterized climate model results are presented without any uncertainty temperature response. I suspect if uncertainty is included the surface temperature impact would be close to zero.

There are a number of important papers missing in the reference lists including Santee et al. (2023).

The author should justify not using OMPS-LP aerosols for SAOD and not using MLS for ozone, temperature and water vapor.

The long discussion trying to reconcile SO₂ and SAOD from previous eruptions is not convincing and distracts from the main focus of the paper.

Specific points.

Line 53 “mass extinction” ... just extinction

Line 87 also Dessler et al. (2013) www.pnas.org/cgi/doi/10.1073/pnas.1310344110

Line 95 you should include Fujiwara et al (2020) <https://doi.org/10.5194/acp-20-345-2020>.

Line 98 estimates of actual ozone depletion are shown in Santee et al. (2023) <https://doi.org/10.1029/2023JD039169> - the others are models.

Line 116 Schoeberl et al. (2024) has updated the calculations and extended them through 2023. This paper is a preprint available at <https://doi.org/10.22541/essoar.171288896.63010190/v1>

This paper provides a similar but more comprehensive trace gas calculation than this work and computed the tropopause flux relative to 10 years of MLS observations.

Line 130 The MLS instrument is on the Aura satellite not Aqua.

Line 135 Why CLIM = 5 years and not 10 or 15 years of observations?

Line 138 Specifically which species? I presume H₂O, aerosols and O₃. Please state for clarity. Note that N₂O is also radiatively active, but MLS observations show a very small N₂O perturbation. You might mention that your TOA flux implicitly includes temperature variations. Because of the stratospheric cooling, colder temperatures will reduce the tropopause flux and presumably the TOA flux as well.

Line 144 Again refer to the Santee et al (2023) paper which is more comprehensive.

Line 148 A substantial change in southern extra-tropical stratospheric ozone in the July-Aug period occurs in mid 2022 due to the secondary circulation set up by the water vapor cooling. Wang et al. (2023) <https://doi.org/10.1029/2023JD039480> Schoeberl et al. (2024) shows that this mid 2022 ozone decrease increases the short wave flux at the tropopause. This component cannot be neglected as the authors have claim to have done here. On the other hand, some of the results include ozone changes and they appear to be non-negligible.

Line 177 "The widespread vertical distribution of SWV in 2023, in contrast to 2022, is attributed to strong changes in stratospheric circulation patterns and responses from localized heating effects." Based on what evidence? Compared to the normal Brewer Dobson circulation, the Hunga induced circulation appears to be small (Fleming et al., 2023 and others). The westerly phase QBO descended in a normal fashion.

Line 182 "seasonal-mean value between 2017 and 2021" you mean the seasonal 5 year climatology starting in 2017?

Line 183-193 I am curious on why the authors use SAGE rather than the more abundant and high quality V5 MLS data. SAGE is low spatial resolution and can't measure in polar night. The MLS data set is not large nor difficult to work with ...

I also don't agree with the comment in the caption that ozone anomaly is negligible. You can clearly see in this (low resolution) diagram that 2022 SON ozone is below normal. Averaging 0-60S reduces the amplitude of the change. The ozone anomaly would be more evident if the authors used MLS data and looked at Santee et al. (op. cit.) or Wang et al. (op. cit.)

Line 190-191 The authors should include more recent references which actually analyze Hunga data rather than Foster et al. and Solomon et al.

Line 201 The use of SW for short wave and SWV for stratospheric water vapor is confusing.

Line 205 Schoeberl et al. (2022) <https://agupubs.onlinelibrary.wiley.com/doi/10.1029/2022GL100248> was the first to note that the local temperature decrease after the eruption was consistent with H₂O radiative cooling.

Line 227 Reword, confusing.

Line 228 Is this TOA flux?

Line 230 I think reporting the surface temperature anomaly due to water vapor without giving an uncertainty is a little misleading. I suspect the uncertainty (usually reported by climate models) will likely be larger than the surface warming value (+0.02 K). Also, I think that the extra SWV emitted photons are going to be absorbed at the cold tropopause and not make it to the surface. This comment also applies to Line 2023 where an even more ridiculously small warming for 2023 is estimated.

Line 258 Recent paper by Sellitto suggest 0.7 Tg as the authors note in Line 314

Line 272 SAGE 5.3 provides an SAOD value which is integrated from the cloud top. Why did the authors choose to generate their own SAOD?

Line 273 Units?

Fig 3e. Silletto et al. (2022) computed the net aerosol heating and found that the long wave and short wave roughly cancelled. I don't believe the authors have included the long wave cooling of the aerosols in the net. Or if they have, it wasn't clear to me. Sato et al, J. Geophys. Res. 98 (1993) 22,987–22,994 show the importance of the LW and SW components of aerosol forcing.

Line 309 Wet scavenging in the stratosphere? Unlikely. More likely is coagulation and settling of the aerosols.

Line 322 The difference in reduction rate due between water and aerosols is due to aerosols settling. As shown in multiple studies, water vapor moves upward on the BD circulation and aerosols settle out.

Line 328 I am confused – did you use SAGE SAOD or did you compute your own SAOD (Line 272). This is a good point about the aerosol SAOD and SO₂ required, but there is some disagreement on SAOD. Another point is that the Aubery simulator uses ambient stratospheric water vapor, whereas Hunga had very elevated water vapor. This may have led to additional larger aerosols and higher SAOD. Finally, USask OMPS SAOD is about 0.01. I would drop this part since you are not on solid ground here.

Line 346-386 Again, this is an interesting, but not all that relevant to the paper's focus. Linking the aerosol forcing to the SO₂ is dicey since SO₂ is uncertain in previous eruptions. Why do you focus on SO₂ when you have the measured SAOD? Also, you have neglected OMPS-LP SAOD. I think the case for higher SAOD/SO₂ distracts from the climate focus of the paper and the arguments more speculative than quantitative. I think you should write a separate paper on this material and delete it here.

Line 390 Totally false as shown in Zhu et al. and Santee et al.

Line 401-409. This result disagrees with Schoeberl et al. (2024) who found ozone a contributor to SW forcing at the tropopause especially in mid to late 2022. You can see the ozone change in the highly averaged figures presented by the author. Even in Table 1 the ozone forcing is as large as the water vapor forcing – with ozone -0.49, without ozone -0.37 (ozone ~30%), unless I misread the table (which is poorly labeled).

I am not sure that the ozone calculation is done correctly. The decrease in ozone should increase the shortwave flux at TOA. In fact this is how satellites measure ozone changes, the atmosphere outgoing flux is absorbed by ozone so if it increase there is an ozone decrease. But Table 1 seems to indicate the opposite there a larger decrease in outgoing flux when ozone is included. This might occur if you only considered the LW component of ozone.

While including ozone does not change the conclusion that Hunga cooled the climate according to the above paper, I believe the authors should look again at ozone changes.

Line 425 Please include uncertainty in surface cooling estimate.

Line 436 -0.37 a global instantaneous value or the peak following the eruption.

Line 464 I assume you are arguing that submarine eruptions are different from subaerial eruptions based on the SO₂-> aerosol conversion. Now Pinatubo had plenty of water (50 Tg) but the eruption was low enough that all the water froze out. The difference between Hunga and Pinatubo is that the eruption went high enough that the water did not freeze out and was available for aerosols. So the difference is more likely the altitude of the eruption rather than submarine versus subaerial.

Line 474 Even though most of Hunga water remains in the stratosphere and mesosphere, the Hunga injection was only about 13% of the total stratospheric water.

Most of this water is above 30 km so looking at the weighting function in Solomon et al. (2010) the impact will be tiny.

Communications Earth & Environment is committed to improving transparency in authorship. As part of our efforts in this direction, we are now requesting that all authors identified as 'corresponding author' create and link their Open Researcher and Contributor Identifier (ORCID) with their account on the Manuscript Tracking System prior to acceptance. ORCID helps the scientific community achieve unambiguous attribution of all scholarly contributions. You can create and link your ORCID from the home page of the Manuscript Tracking System by clicking on 'Modify my Springer Nature account' and following the instructions in the link below. Please also inform all co-authors that they can add their ORCIDs to their accounts and that they must do so prior to acceptance.

Version 1:

Decision Letter:

Dear Dr Gupta,

Your revised manuscript titled "The January 2022 Hunga eruption cooled the southern hemisphere in 2022 and 2023" has now been seen by the 3 original reviewers, whose comments are appended below. You will see that although they find your work of some potential interest, they continue to raise concerns that we feel must be addressed before we can make a final decision on your article.

Specifically, for publication in Communications Earth & Environment to be appropriate, we ask you to double-check and correct, where necessary, all calculations in light of the points the reviewers have raised, and thereby ensure the main conclusion of the paper - that sulfate aerosols produced a large cooling following the 2022 Hunga eruption - is robustly supported. We therefore invite you to revise and resubmit your manuscript, along with a point-by-point response that takes into account the points raised. Please highlight all changes in the manuscript text file.

Please submit your point-by-point responses as a separate file, distinct from your cover letter where you can add responses to the Editors' comments that you do not want to be made available to the reviewers. Word files are preferred.

Important: The response to reviewers must not include any figures, tables or graphs. If you wish to respond to the reviewer reports with additional data in one of these formats, please add them to the main article or Supplementary Information, and refer to them in the rebuttal. Due to current technical limitations, any figures, tables, or graphs embedded in your rebuttal will not be included in the peer review file, if published.

Please use the following link to submit your revised manuscript, point-by-point response to the referees' comments (which should be in a separate document to any cover letter), a tracked-changes version of the manuscript (as a PDF file) and the completed checklist:

Link Redacted

We hope to receive your revised paper within six weeks; please let us know if you aren't able to submit it within this time so that we can discuss how best to proceed. If we don't hear from you, and the revision process takes significantly longer, we may close your file. In this event, we will still be happy to reconsider your paper at a later date, as long as nothing similar has been accepted for publication at Communications Earth & Environment or published elsewhere in the meantime.

Please do not hesitate to contact us if you have any questions or would like to discuss these revisions further. We look forward to seeing the revised manuscript and thank you for the opportunity to review your work.

Best regards,

Dr Alireza Bahadori
Associate Editor
Communications Earth & Environment

EDITORIAL POLICIES AND FORMATTING

Editorial Policy: [Policy requirements](https://www.nature.com/documents/nr-editorial-policy-checklist.pdf) (Download the link to your computer as a PDF.)

- Behavioural and social science
- Ecological, evolutionary & environmental sciences

• Life sciences

<https://www.nature.com/documents/nr-reporting-summary.zip>

Furthermore, please align your manuscript with our format requirements, which are summarized on the following checklist:

<https://www.nature.com/documents/commsj-phys-style-formatting-checklist-article.pdf> > Communications Earth & Environment formatting checklist

and also in our style and formatting guide <https://www.nature.com/documents/commsj-phys-style-formatting-guide-accept.pdf> > Communications Earth & Environment formatting guide .

*** DATA: Communications Earth & Environment endorses the principles of the Enabling FAIR data project (<http://www.copdess.org/enabling-fair-data-project/>). We ask authors to make the data that support their conclusions available in permanent, publically accessible data repositories. (Please contact the editor if you are unable to make your data available).

All Communications Earth & Environment manuscripts must include a section titled "Data Availability" at the end of the Methods section or main text (if no Methods). More information on this policy, is available at a

<http://www.nature.com/authors/policies/data/data-availability-statements-data-citations.pdf> > <http://www.nature.com/authors/policies/data/data-availability-statements-data-citations.pdf> .

DATA SOURCES: All new data associated with the paper should be placed in a persistent repository where they can be freely and enduringly accessed. We recommend submitting the data to discipline-specific, community-recognized repositories, where possible and a list of recommended repositories is provided at a

<http://www.nature.com/sdata/policies/repositories> > <http://www.nature.com/sdata/policies/repositories> .

If a community resource is unavailable, data can be submitted to generalist repositories such as <https://figshare.com/> > figshare or <http://datadryad.org/> > Dryad Digital Repository . Please provide a unique identifier for the data (for example a DOI or a permanent URL) in the data availability statement, if possible. If the repository does not provide identifiers, we encourage authors to supply the search terms that will return the data. For data that have been obtained from publically available sources, please provide a URL and the specific data product name in the data availability statement. Data with a DOI should be further cited in the methods reference section.

Please refer to our data policies at a

<http://www.nature.com/authors/policies/availability.html> > <http://www.nature.com/authors/policies/availability.html> .

REVIEWER COMMENTS:

Reviewer #1 (Remarks to the Author):

I thank the authors for the thorough explanation of their approach. From their answers, it is clear that the paper presents the instantaneous radiative forcing of three major optically active constituents: stratospheric Water Vapor (WV), Sulfate Aerosols, and Ozone. I still believe it is a valuable and relevant study, but I am afraid the paper in its current form contains a technical error. The longwave (LW) instantaneous radiative forcing (IRF) of WV at the top of the atmosphere (TOA) can not be positive. Adjustment of the stratospheric temperature changes the sign of the forcing, so an adjusted Radiative forcing (ARF) of WV at TOA is positive. At the same time, IRF at tropopause is positive. The two-level energy balance model, used to evaluate a climate response, requires the forcing at the tropopause. I suggest the authors reconsider their calculation of WV LW IRF and spare a paragraph in the paper to discuss IRF at the tropopause and the surface. This will significantly increase the value of their paper.

Reviewer #2 (Remarks to the Author):

I find the manuscript to be improved from the previous draft. I would support the publication of this paper after implementing the changes below.

1) The focus of this paper is on radiative effects. As such, all that's needed for ozone is to quantify the ozone changes, not explain why ozone changed. No reader of this paper will expect the authors to explain why ozone changed in 2023. The authors themselves state that explaining the ozone loss is out of scope for the present paper on lines 440–444. And yet, despite this, the authors somewhat inexplicably spend significant text speculating about why and how ozone changed: lines

426–438 of the manuscript and lines 132–174 of the Supplemental. I strongly encourage the authors to delete all of this. There are many mistakes in what the authors are saying, particularly in the Hunga-Ozone interaction section of the Supplemental and the interpretation of Figure S19. It will not compromise the story of the paper in any way to delete these parts and, it will significantly improve the paper by removing the inaccurate and speculative statements.

2) Figure 5d and Lines 486–503 need additional clarification. It is stated that the radiative forcing of -0.38 Wm^{-2} caused a surface cooling of -0.05 K in 2022. How is it then that a radiative forcing of only -0.01 Wm^{-2} caused the exact same amount of surface cooling of -0.05 K in 2023. Why didn't the delta T become less negative in 2023 in Figure 5d when the radiative forcing was smaller? This should be discussed in the manuscript.

3) I wonder about the ozone filtering for 2023. Lines 651–679 describe that an ozone change was only linked to Hunga if water was simultaneously greater than 2 sigma from the mean, based on the method of Wilmouth et al. But Wilmouth et al. only used that method for 2022, when most of the sulfate and water vapor was still in the Southern Hemisphere. This approach may not work as well for 2023, as done in the Gupta et al. manuscript. Even for 2022, Wilmouth et al. were concerned about the separation of the aerosol layer and water vapor and thus tested their results over different pressure levels; and it is important to note that they were just looking for the peak anomalies, which is a different analysis than what is being done in the present manuscript. In short, the authors of Gupta et al. should check for 2023 that they aren't filtering out ozone data where there were perturbations due to Hunga from elevated sulfate in the absence of substantially elevated water vapor.

4) I continue to contend that the Methods section is far too long. Much of this detail can be moved to the Supplemental, with only the most essential information retained in the main manuscript.

Additional comments:

Line 43. The statement "... with less sulfur dioxide than past eruptions" is incorrect as written. Do you mean less sulfur dioxide than past eruptions of comparable size?

Lines 45-46. This sentence is very unclear: "However, considering the cooling impact of sulfate aerosols and ozone loss, and the warming effect of ozone gain is crucial to understand the eruption's effects." Consider changing this to: "However, considering the cooling impact of sulfate aerosols and the potential for ozone loss (gain) to cause cooling (warming), it is crucial to understand the eruption's effects."

Line 48. Delete "their"

Line 54. Why "respectively"? There's only one number listed.

Line 115. Reference 15 is from a Preprint and should not be cited unless it is actually published.

Line 122. TOA has already been defined on line 89.

Line 124. Change "the comprehensive" to "a comprehensive"

Line 135 and many other places. Aura is a word, not an acronym. It should be written as Aura, not AURA.

Lines 281-284 and Line 299. Clarify why the text says that Hunga was 1.5 Tg and Raikoke was 1.5 Tg but then says that the quantity of SO₂ injected from Hunga was much smaller.

Line 372. "net TOA" should say "net TOA radiative flux"

Line 386. "with estimate" doesn't make sense. Do you mean "than the estimate"?

Line 395. Change "factor 3 differences between" to "factor of 3 between"

Line 424. It would be better to reorder this from "ozone chemistry and stratospheric dynamics" to "stratospheric dynamics and ozone chemistry". List the dynamics first because it is more important.

Figure 5. Use a larger font on the axis labels.

Line 452. Delete "We show that"

Lines 472-473. Change "contributing to stratospheric cooling" to "contributing to stratospheric warming"

Table 1. Bold the delta NRF for O₃, as was done for SAOD and H₂O. And I suggest making all the numbers bold in all 3 delta NRF rows as well, so that the numbers being added stand out. Move the delta H₂O total row up, so that the delta NRF row for water is last in the dark orange section.

Line 530. H₂O mass is Tg, not Pg.

Supplemental Lines 224-225 and Figure captions S16-S19. The MLS background is listed 3 different ways. It's definitely wrong in the Supplement text lines 224-225. Captions S16 and S17 say it's 2005-2021, but then state it's 15 years, so that's inconsistent. 2005-2021 would be 17 years. The stated background years change again in Figures S18 and S19, where the background is listed as 2007-2021. All of these dates need to be checked and corrected.

Reviewer #3 (Remarks to the Author):

Review of Gupta et al.

Overview: The paper is an improvement over the previous version – the authors included ozone changes in their estimate of radiative flux changes and addressed previous reviewer comments. The overall results are in agreement with Schoeberl et al. (2024, JGR in press) and Stenchikov et al. (submitted to JGR). While the paper has a good analysis using only the SAGE data, the overall discussion is still hard to follow with a lot of speculation and unclear paragraphs. I would recommend that the paper go back to the authors for extensive 'clean up' removing speculation and consolidating some points that are repeated in different paragraphs.

Specific Comments.

The paper by Schoeberl et al. (2024) has been accepted in JGR. This paper covers the same material and comes to the same conclusion. Schoeberl et al. (2024) used OMPS-LP and MLS measurements in their computations and since the spatial resolution of those instruments is higher, they were able to better detail the evolution of the aerosols and trace gases.

Ln 119 – previous work – please be clear. Do you mean Jenkins or Schoeberl? Jenkins did not include direct radiative forcing reduction by aerosols. This sentence is also a little misleading. While Schoeberl et al. (2024) did not include TOA nor the estimate the impact on climate warmth, it is obvious that a net decrease in forcing would not increase surface temperatures. I suggest the authors drop sentences Ln 119 to 122 since they are somewhat misleading.

Ln 124 'a' not 'the'

Ln 138 I am puzzled why the authors use SAGE water vapor data for the RT calculation instead of MLS. SAGE has better vertical resolution, but is noisier at HT altitudes and doesn't extend as far vertically. The authors should explain their choice in a few sentences here. Ln 610 has an explanation, but the argument about self-consistency isn't quite correct. The authors should consult the SAGE ATDB. Water vapor is retrieved using the weak absorption band at 940nm. All of the gas absorptions and Rayleigh scatter components are estimated first including ozone which interferes with the water vapor retrieval. Extinction that still can't be explained by gas absorbing regions is assigned to aerosols – the residual. Examination of individual SAGE aerosol profiles show that they are quite noisy at higher altitudes. I would not call this 'self consistent.'

Also, because the H₂O absorption band is so weak, the signal is difficult to detect above ~ 40 km. This noise is apparent in Fig. 1a above 32 km.

Ln 144-149 I don't understand this part of the paragraph. Is this about determining the background aerosol levels? The authors are excluding SAGE II measurements. Why aren't the authors using OMPS-LP aerosol measurements. That record extends from 2014 and Hunga papers using OMPS-LP data are in the literature (e.g. Taha et al., 2023). SAGE also missed the initial eruption due to its orbit inclination.

Ln 228 'A narrower band...' not sure what you are referring to here.

Ln 276 'Produced a surface warming...' This is unclear wording. What is happening is that there is an increase in downward IR flux due to stratospheric water vapor. This radiation is most likely absorbed at the cold tropopause where there is plenty of water vapor below the temperature of the high-altitude emission layer. Thus, I doubt the radiation will penetrate to the surface. I suggest rewording as 'tropospheric warming.'

Figure 2. Please define NRF in the caption.

Ln 310 I think that SAOD should be computed from the tropopause upward rather than from 20-42 km. Outside the tropics the tropopause is much lower than 20 km and OMPS-LP observed aerosols moving southward along isentropes into the region below 20 km especially in late spring. Aerosols below 20 km will enhance the SAOD so the solar forcing reduction estimated here will be low biased.

Ln 281/282 Sellitto actual estimate is 0.7 Tg ± 0.5 Tg. I think 1.5 Tg is a little high and having spoken to Sellitto, the 1 Tg number is more reasonable. Carn et al. (2022, <https://doi.org/10.3389/feart.2022.976962>) estimates total as high as 0.7 Tg. These estimates are different from those mentioned in the paragraph starting on line 349.

Ln 299 Missing sentence

Ln 314/315 I don't see how Fig. 3e supports the statement that there is no strong longitudinal pattern. Do you mean Fig. 3c?

Ln 347 This is just a laundry list of explanations for aerosol changes. Please be more specific. Given that transport of water vapor is quite different from aerosols, it is doubtful that dynamics is the explanation for the different distribution. That leaves gravitational settling as the most likely hypothesis as noted in a number of papers such as Legras et al. (2022). Dry deposition is a surface process which is unlikely to occur in the stratosphere. Aerosols can also evaporate, but the lower stratosphere is too cold for evaporation to remove sulfate aerosols.

Ln 349-363 I can't understand the point the author is trying to make here. That the SAOD is inconsistent with sulfur emissions. Recall that this eruption occurs with a significant amount of water and the rapid growth of aerosol concentration may be due to the presence of so much water. SO₂ is oxidized by OH to form sulfate aerosols, with all the water present, there will be an abundance of OH. This point is better made on Ln 394-397. Perhaps these paragraphs could be combined.

Ln 360 This speculation about the Brewer Dobson circulation makes no sense.

Ln 367 All this points to is the failure of the EVA_H model for this type of eruption. Is that your point?

Ln 426-438 Lots of speculation here – no real information.

Ln 539 The authors used the 5 previous pre-eruption years to create a baseline. The slight warming in the NH could simply be year-to-year variability and not associated with Hunga. They should look at individual years within the 5 year average to estimate the year-to-year variability.

Ln 615 ... higher vertical resolution... But given the low spatial sampling of SAGE relative to MLS and OMPS-LP I am not sure your point about better spatio-temporal sampling from SAGE is correct. I am not seeing anywhere that the slightly lower vertical resolution from MLS makes a difference in your calculation (also see Ln 583). Also see comment above about SAGE data consistency. Note that MLS data extends to much higher altitude than SAGE, covers the region between 60-82° unlike SAGE and provides global coverage every day unlike SAGE.

Ln 711 The uncertainty in SAGE is also given in Wrana et al. (2021); <https://doi.org/10.5194/amt-14-2345-2021>

Ln 851 Dawn-dusk stratospheric profiles.

Ln 866 Schoeberl et al. (2024) included temperature anomalies and found that, although not as significant as water vapor or aerosols, should be included.

Ln 883 a simple comparison to OMPS-LP will tell you how much SAOD you underestimate by only integrating to 20 km. This might be important.

Communications Earth & Environment is committed to improving transparency in authorship. As part of our efforts in this direction, we are now requesting that all authors identified as 'corresponding author' create and link their Open Researcher and Contributor Identifier (ORCID) with their account on the Manuscript Tracking System prior to acceptance. ORCID helps the scientific community achieve unambiguous attribution of all scholarly contributions. You can create and link your ORCID from the home page of the Manuscript Tracking System by clicking on 'Modify my Springer Nature account' and following the instructions in the link below. Please also inform all co-authors that they can add their ORCIDs to their accounts and that they must do so prior to acceptance.

Version 2:

Decision Letter:

Dear Dr Gupta,

Your manuscript titled "The January 2022 Hunga eruption cooled the southern hemisphere in 2022 and 2023" has now been seen by our reviewers, whose comments appear below. In light of their advice we are delighted to say that we are happy, in principle, to publish a suitably revised version in Communications Earth & Environment, provided you include a robust discussion of the limitations of the methodology as well as the associated uncertainties regarding your interpretation of a large cooling following the 2022 Hunga eruption, along the lines recommended by our reviewers.

Given that some of our referee 3's concerns have been raised before and not fully addressed, we are not certain whether you are able to add such a robust discussion; if not, unfortunately we cannot consider your manuscript further, and would therefore recommend you seek publication elsewhere.

We also ask you to edit your manuscript to comply with our format requirements and to maximise the accessibility and therefore the impact of your work.

EDITORIAL REQUESTS:

****Please take care to match our formatting and policy requirements. We will check revised manuscript and return manuscripts that do not comply. Such requests will lead to delays. ****

SUBMISSION INFORMATION:

OPEN ACCESS:

Communications Earth & Environment is a fully open access journal. Articles are made freely accessible on publication. For further information about article processing charges, open access funding, and advice and support from Nature Research, please visit <https://www.nature.com/commsenv/open-access>

Link Redacted

Best regards,

Alireza Bahadori, PhD
Associate Editor
Communications Earth & Environment

REVIEWERS' COMMENTS:

Reviewer #1 (Remarks to the Author):

As was said in the first-round review, the paper tackles an important scientific question, uses a relevant scientific approach, is logically organized, and is well-written. It characterizes the impact of the Hunga eruption, combining observations available since 2022 and radiative transfer calculations to estimate radiative forcing and climate impact of sulfate aerosols, stratospheric water vapor, and perturbations of ozone forced by volcanic injection. I think it is a relevant and important study. Unfortunately, it suffers from minor errors and misinterpretations. In the revision, the authors corrected their calculation of

water vapor instantaneous radiative forcing and added an analysis of the effect of stratospheric temperature adjustment in the supplement. This improved the paper, but several minor things must be dealt with before the paper is ready for publication.

Specific comments:

L103: Shallow submarine eruptions are infrequent. Most submarine eruptions happen in deep water, bringing little volcanic material to the surface.

L116: Wang&Huang published recently a paper presenting calculations of the Hunga radiative forcing conducted offline and within a climate model. Please cite this paper.

Wang, Y., & Huang, Y. (2024). Compensating atmospheric adjustments reduce the volcanic forcing from Hunga stratospheric water vapor enhancement. *Science Advances*, 10(32), eadl2842.

L126: I believe that you use the 1D radiative transfer model that you apply in each grid cell of the global domain that makes a 3D output. Please correct me if I am wrong.

L137: Remove "longitudinal"

L167: Why not at 100 hPa? The 16-km level will not work for middle and high latitudes.

L223: I do not think you can make this conclusion. You did not calculate the stratospheric temperature responses. Moreover, below, you report the -0.5 K/day ozone cooling rate, three times the water vapor effect.

L225-228: This is either an incorrect interpretation, or it is spelled so that it sounds wrong. The lifting is not induced by volcanic impact. It is the background BCD. Water vapor diabatic cooling delayed lifting.

L236: It is not because of trapping but because of the downward thermal emission of the water vapor layer.

L265-267: You do not account for clouds and any temperature adjustments. Clear-sky radiative forcing could be twice as minor compared with all-sky forcing. So, your uncertainty may be larger than you report.

L311-314: What is the median radius of the aerosol size distribution? In Mie calculations, we use a geometric radius but not an effective one.

L365: Boichu et al. and Khaykin et al. reported much larger particles.

L370: This is misleading. In literature, it is assumed that the Hunga aerosol particles are large compared to the small subaerial eruptions and small compared to large eruptions like Pinatubo.

l392-416: Two factors define MEE dependence on particle radius. The first factor is non-monotonic extinction efficiency. Its first maximum is about 25% larger than the second maximum. And the second factor is the sum of particle geometric crosssections. For a given mass of material, this factor is inversely proportional to the particle radius. If Hunga's particles have a radius of 0.2 microns and Pinatubo's particles have a radius of 0.5, then Hunga's aerosol should be 2.5 times more optically effective per unit mass than Pinatubo's aerosols. Thus, for comparison of Hunga and Pinatubo aerosols, the major role belongs to this second factor, but not extinction efficiency, as the authors claim. This factor works in the opposite direction if we compare Hunga's aerosols with small eruptions like Raikoke, Calbuco, or Nabro.

L468-476: The ozone radiative forcing sounds reasonable, but ozone heating rates are larger than expected. In Table 1 (column for SH, 2022), energy loss in the stratosphere (TOA forcing - 16km forcing) due to water vapor is 0.182 W/m², while due to ozone -0.011, almost 20 times less. However, ozone stratospheric heating rates are 2-3 times bigger than water vapor's. Could you please clarify this?

Reviewer #2 (Remarks to the Author):

The authors appear to have addressed my concerns from the previous version. I have just a few new comments.

Line 43. Why "submarine"? What prior submarine eruptions are being referred to? I'd delete the word submarine here.

There is a discrepancy between Lines 53-54 and Lines 554 and 564. Is it -0.07 K in 2022 and 2023 or -0.07 K in 2022 and -0.06 K in 2023? Based on Fig 5d, 2023 should be more negative than 2022, so depending on how the values are rounded, the text is wrong in both places mentioned above. This is very important to clean up, especially since it's a key result appearing in the Abstract. If cooling does round to -0.07 K for both years, other than correcting Line 564, I'd also suggest being clearer in the Abstract about what is meant, i.e., change "at the end of 2022 and 2023." to "at the end of each year 2022 and 2023."

Figure 2. Consider moving the legends outside the righthand panels, so the lines within the plots on the right side can be

rescaled to use all the space. Otherwise, the lines are very congested and can't be distinguished in some cases.

Line 81. Lowercase volcano
Line 99. concentrations
Line 134. employs a solar
Line 142. altitudes
Line 152. No dash after Aura

There are many inconsistencies in the formatting of the references. The authors should use the same format for every reference per the journal guidelines. And make sure the use of 'et al.' is being done consistent with how the journal requires.

Supplemental:
Lines 305-306. The MLS dates of Jun 2017 to Jan 2023 are not correct.

Lines 308-310. Delete this sentence.

There are a lot of figures in the Supplemental, and many have been renumbered and new figures have been added since the last draft. I strongly encourage the authors to verify that every time a Supplemental figure is mentioned in the manuscript text or Supplemental text, it is listed using the correct Supplemental figure number. I don't believe that this is currently the case.

Reviewer #3 (Remarks to the Author):

The authors have made significant improvements to this paper and done a lot of work to respond the reviews. However, there are still nagging issues with this calculation. It is no surprise, however, that their results agree with already published works. There are also lot of typos and unclear statements in the current text. This paper is still pretty rough.

1) I still find the argument about using only SAGE data weird. First of all, the SAGE data is not very good for water vapor retrievals at high altitude and it also has problems below 20 km. Second, SAGE is also the major component of GloSSAC which is why they agree. USask OMPS and NASA OMPS disagree and this is due to the size distribution assumptions which (in the NASA case) are not consistent with better distribution width used by Duchamp et al. & Kneff et al. Prior to the Hunga eruption, the three retrievals schemes agreed well.

2) Because the authors are stuck on using SAGE, their climatology is only 5 years which might be a little short for this kind of calculation. Other authors have used 10 years. I wonder if the QBO will create a bias in their results since the QBO cycle is 18 months and may not be cleanly averaged using just 5 years. They may want to compare to an MLS climatology of 10 or 15 years.

3) The justification for the filtering scheme is a little odd. The scheme co-locates changes in ozone with anomalies of a certain size and eliminates anomalies that are outside requirements. This assumes that ozone anomalies are coincident water vapor anomalies – as I understand it. However, ozone changes generated by the anomalies' secondary circulation will be outside of the anomaly location and thus eliminated by the 2D filter. The authors need to comment on this possibility.

4) The concept of instantaneous forcing – not including the temperature response – is quite relevant for climate models where ocean temperature changes will take place on a longer time scale. But here, that argument falls flat. Stratospheric temperatures respond in days-weeks so it is appropriate to include the temperature response in the flux calculation. The cooler Hunga stratosphere will reduce the downward IR flux.

5) Integrating the aerosol extinction from 20 km up underestimates the SAOD and water vapor impact. What is more confusing is they report radiative forcing at 16 km, 4 km lower. Outside the tropics, the tropopause is ~12 km and OMPS shows significant aerosol concentration well below 20km as the figure from Taha et al. (2024) (in attachment, I marked 20 km to show where your cutoff is). Solomon et al. (2010) vertical weighting function for water vapor shows the highest values close to the tropopause, thus I think the authors need to tell us the level of uncertainty introduced by limiting the integration height.

Specific comments

L 51 sentence is confusing
L142 My experience with SAGE aerosol data is that it is good down to the tropopause, it is noisier above 25 km. The water vapor is quite a bit noisier in the aerosol layer which the authors don't discuss. This likely explains the dry bias since high water and high aerosol are co-located. In any event, the statement about 'enhanced self-consistency' is silly especially since the data are 2D filtered. Also, the argument about vertical resolution should be demonstrated – does vertical resolution have any actual impact on the calculations? Finally, they undercut their own argument about using SAGE on line 156 where they note that the data sets all agree.
L176 The authors use HTHH and Hunga interchangeably. The community has decided on just 'Hunga' so the author may want to reconsider the labels
L184 The sentence 'more dilute water' what does that mean?
L 193 This sentence is awkward due to all the prepositional caveats.

L 215 This where I get confused. The forcing is computed at 16 km but the SAOD is computed above 20 km. This is inconsistent. Fig. 1g is misleading indicating the NRF at 16 km.

Ln 268 again mentions 16 km but the integration of the SAOD component is actually above 20 km: 4 km missing
Figure 1 a-d looks like there are no values below zero – so why have the scale range go negative. Also you should show the distribution down to 16 km since the forcing is computed at that level.

Figure 2 becomes a little misleading when the lines connect the dots. For example, 2a seems to indicate water vapor increased before the eruption. It might be better to stop the lines up to the Hunga boundary and start them after.

Figure 4 is really hard to read with the lines overlaid.

L 351 I don't understand how GloSSAC and SAGE come up with such different SAOD, is it the integration height range? GloSSAC is basically SAGE with a little CALIPSO in 2022 and part of 2023. You need to explain this.

L371 There are a number of papers on the settling time scale for volcanic aerosols (e.g. Hamill et al., 1997) that you should reference.

L400 This is an odd paragraph. After discussion on line 304 of ~ 1Tg of SO₂ the estimate used here is for 0.6 Tg and the authors are surprised by the low level of SAOD. The authors need to get this straight.

Figure 5 – 5d suggests the there is net (small) 2022 NH heating at 16 km. How did this happen? I wonder if this is a result of an ozone bias in CLIM. CLIM should be compared to MLS 10 year averages.

Table 1 – doing a visual comparison of these results with Schoeberl et al (2024) there is reasonable agreement, and differences can probably be ascribed to different SAOD and temperature changes not being included.

L579 the reference here is to Pinatubo which was ~ 20Tg of sulfur compared to < 1 Tg of sulfur for HTHH – Pinatubo created a climate change of (at most) -0.5 K so scaling just with just sulfur the impact would be -0.025K which is probably not detectable.

L593 Decade? More like 2.5 years – see Fleming et al. (2023)

made.

The authors have made significant improvements to this paper and done a lot of work to respond the reviews. However, there are still nagging issues with this calculation. It is no surprise, however, that their results agree with already published works. There are also lot of typos and unclear statements in the current text. This paper is still pretty rough.

- 1) I still find the argument about using only SAGE data weird. First of all, the SAGE data is not very good for water vapor retrievals at high altitude and it also has problems below 20 km. Second, SAGE is also the major component of GloSSAC which is why they agree. USask OMPS and NASA OMPS disagree and this is due to the size distribution assumptions which (in the NASA case) are not consistent with better distribution width used by Duchamp et al. & Kneff et al. Prior to the Hunga eruption, the three retrievals schemes agreed well.
- 2) Because the authors are stuck on using SAGE, their climatology is only 5 years which might be a little short for this kind of calculation. Other authors have used 10 years. I wonder if the QBO will create a bias in their results since the QBO cycle is 18 months and may not be cleanly averaged using just 5 years. They may want to compare to an MLS climatology of 10 or 15 years.
- 3) The justification for the filtering scheme is a little odd. The scheme co-locates changes in ozone with anomalies of a certain size and eliminates anomalies that are outside requirements. This assumes that ozone anomalies are coincident water vapor anomalies – as I understand it. However, ozone changes generated by the anomalies' secondary circulation will be outside of the anomaly location and thus eliminated by the 2D filter. The authors need to comment on this possibility.
- 4) The concept of instantaneous forcing – not including the temperature response – is quite relevant for climate models where ocean temperature changes will take place on a longer time scale. But here, that argument falls flat. Stratospheric temperatures respond in days-weeks so it is appropriate to include the temperature response in the flux calculation. The cooler Hunga stratosphere will reduce the downward IR flux.
- 5) Integrating the aerosol extinction from 20 km up underestimates the SAOD and water vapor impact. What is more confusing is they report radiative forcing at 16 km, 4 km lower.

Outside the tropics, the tropopause is ~12 km and OMPS shows significant aerosol concentration well below 20km as the figure from Taha et al. (2024) below shows (I marked 20 km to show where your cutoff is). Solomon et al. (2010) vertical weighting function for water vapor shows the highest values close to the tropopause, thus I think the authors need to tell us the level of uncertainty introduced by limiting the integration height.

Specific comments

L 51 sentence is confusi

L142 My experience with SAGE aerosol data is that it is good down to the tropopause, it is noisier above 25 km. The water vapor is quite a bit noisier in the aerosol layer which the authors don't discuss. This likely explains the dry bias since high water and high aerosol are co-located. In any event, the statement about 'enhanced self-consistency' is silly especially since the data are 2D filtered. Also, the argument about vertical resolution should be demonstrated – does vertical resolution have any actual impact on the calculations? Finally, they undercut their own argument about using SAGE on line 156 where they note that the data sets all agree.

L176 The authors use HTHH and Hunga interchangeably. The community has decided on just 'Hunga' so the author may want to reconsider the labels

L184 The sentence 'more dilute water' what does that mean?

L 193 This sentence is awkward due to all the prepositional caveats.

L 215 This where I get confused. The forcing is computed at 16 km but the SAOD is computed above 20 km. This is inconsistent. Fig. 1g is misleading indicating the NRF at 16 km.

Ln 268 again mentions 16 km but the integration of the SAOD component is actually above 20 km: 4 km missing

Figure 1 a-d looks like there are no values below zero – so why have the scale range go negative. Also you should show the distribution down to 16 km since the forcing is computed at that level.

Figure 2 becomes a little misleading when the lines connect the dots. For example, 2a seems to indicate water vapor increased before the eruption. It might be better to stop the lines up to the Hunga boundary and start them after.

Figure 4 is really hard to read with the lines overlaid.

L 351 I don't understand how GloSSAC and SAGE come up with such different SAOD, is it the integration height range? GloSSAC is basically SAGE with a little CALIPSO in 2022 and part of 2023. You need to explain this.

L371 There are a number of papers on the settling time scale for volcanic aerosols (e.g. Hamill et al., 1997) that you should reference.

L400 This is an odd paragraph. After discussion on line 304 of ~ 1Tg of SO₂ the estimate used here is for 0.6 Tg and the authors are surprised by the low level of SAOD. The authors need to get this straight.

Figure 5 – 5d suggests there is net (small) 2022 NH heating at 16 km. How did this happen? I wonder if this is a result of an ozone bias in CLIM. CLIM should be compared to MLS 10 year averages.

Table 1 – doing a visual comparison of these results with Schoeberl et al (2024) there is reasonable agreement, and differences can probably be ascribed to different SAOD and temperature changes not being included.

L579 the reference here is to Pinatubo which was ~ 20Tg of sulfur compared to < 1 Tg of sulfur for HTHH – Pinatubo created a climate change of (at most) -0.5 K so scaling just with just sulfur the impact would be -0.025K which is probably not detectable.

L593 Decade? More like 2.5 years – see Fleming et al. (2023)

Reply to Reviewers

We would like to thank the editor and all three reviewers for carefully reviewing our manuscript and providing valuable feedback. Their insightful comments have greatly improved the quality of our work. In the revised version of our manuscript, we present a more comprehensive analysis that reveals clear changes in ozone levels associated with the Hunga eruption in 2022 and 2023. We acknowledge that our statement in the original submission that there was no significant hemispherically averaged ozone change following the Hunga eruption was misleading – we didn't mean to suggest that there were no ozone changes following the eruption, just that they are not globally or hemispherically coherent, unlike the changes observed in stratospheric water vapor and SAOD, and thus may not be important from a global radiative budget perspective.

To rectify this, we have included a new analysis that follows previous work on the Hunga eruption related first-year ozone anomalies (e.g., Wilmouth et al., 2023 and Santee et al., 2023) to identify the ozone changes due to the Hunga eruption for both 2022 and 2023. Specifically, by utilizing zonally filtered data (termed as 2D-Filtered technique; see revised Methods section), we are able to statistically determine ozone anomalies and accurately calculate the corresponding instantaneous radiative effect. Our new and detailed analysis removes our previous assumptions/approximations, enabling us to statistically determine the ozone anomalies and calculate the corresponding instantaneous radiative effect.

We do note that even with this significant additional analysis, our primary conclusions of the manuscript remain unchanged since the dominant radiative effect is because of SAOD and stratospheric water vapor, though the exact amplitude of the top-of-the-atmosphere radiative forcing due to the Hunga eruption does change slightly compared to the original submission.

The response to reviewer #1 is provided in blue text, reviewer #2 in red, and reviewer #3 in purple. We have also added various new references related to ozone and other content. We have revised the manuscript according to the suggestions, including restructuring the figures to improve the paper's readability.

Reviewer #1 (Remarks to the Author)

General comments:

Q1. The paper tackles an important scientific question about the climate impact of the 2022 Hunga eruption. The methodological approach is sound. The authors use SAGE observations to calculate the radiative forcing of volcanic aerosols, stratospheric water vapor, and ozone, coming to a plausible, robust conclusion about net cooling caused by the Hunga eruption. The authors analyze for two post-eruption years. This is not the first attempt. There are at least two similar in a pipeline for publication (<https://essopenarchive.org/users/523044/articles/741323-evolution-of-the-climate-forcing-during-the-two-years-after-the-hunga-tonga-hunga-ha-apai-eruption> and <https://essopenarchive.org/doi/full/10.22541/essoar.169091894.48592907>)

Reply 1: We appreciate your critical evaluation of our manuscript and your positive comments regarding its consideration for publication in this journal.

We had transferred the manuscript titled "The January 2022 Hunga eruption cooled the southern hemisphere in 2022 and 2023" from Nature Climate Change to Nature Geosciences Journal in November 2023 and posted that original paper on Research Square (<https://www.researchsquare.com/article/rs-3493146/v1>). The papers that the reviewers mention were submitted while this paper was in the review process.

We initially referred to it as our first submission, we have since removed the word "first" from **Line #124** since other related studies maybe in pipeline for publication.

We recognize that Schoeberl et al. (2023) demonstrated the cooling effect of the eruption by estimating the radiative effect of sulfate aerosols using a scaling relationship between stratospheric optical depth (SAOD) and radiative forcing based on observational and modeling results for previous subaerial eruptions. While this provides a good first order approximation, our study provides rigorous calculations of radiative forcing by doing forward radiative transfer models and accounting for spatially varying aerosol size distribution.

We conducted a detailed analysis of aerosol extinction profiles and particle size evolution using multiple wavelengths in the SAGE-III data. We also assessed the energy loss caused by sulfate aerosols, which was found to be higher than the energy loss from Stratospheric water vapor (SH₂O). Furthermore, we have provided evidence through role of particle size in producing higher MEE to explain why the Hunga eruption led to a net cooling despite a relatively small volume of volcanic SO₂ (compared to large eruptions like 1991 Pinatubo) Therefore, we believe that our study contributes valuable information to the comprehensive assessment of the impact of the Hunga eruption in 2022 and 2023 due to three species—stratospheric water vapor, sulfate aerosol and **ozone (in the revised manuscript; Lines 406-439)**.

Q2. The magnitude of the forcing sounds reasonable, although, at least for water vapor, it is overestimated as, in the way it is calculated, it includes climate temperature variability that is not estimated in the paper.

Reply 2: Please note that we conducted the idealized instantaneous radiative forcing analysis for all three species. We did not change the climate temperature variability between HTHH and CLIM values (see our revised texts in the **Lines #261-266** in the Main Manuscript and **Lines #772-777**).

Q3. Among the results, I'm afraid I have to disagree with the explanation of why the Hunga eruption is more effective in generating SAOD. With recent estimates, which push the initial emission of SO₂ to 1.5 Mt, it may not be that dramatic.

Reply 3: We thank the reviewer for the comment. With regards to the Hunga eruption's SO₂ estimate, multiple studies (Duchamp et al., 2023; Boichu et al., 2024; Sellitto et al., 2023) have found that the Hunga eruption released between 0.5 and 0.7 Tg of SO₂. To our knowledge, the only reference that mentions a much higher SO₂ budget of 1.5 Tg is Sellitto et al. (2023), but this study actually refers to 1.5 Tg as the total amount of SO₂ in both the troposphere and stratosphere (see **Lines 281-282**). The total SA mass burden was estimated at 1.6 ± 0.1 Tg in total column, with possibly ~45% in the stratosphere (~0.7 Tg) and the remaining ~55% in the troposphere (~0.9 Tg). However, only the stratospheric SO₂ (~ 0.5 Tg) has a sufficiently long atmospheric lifetime to cause substantial radiative cooling on Earth.

Q4. The paper is well-written and logically organized. However, it uses unconventional terminology in some places. The authors must better explain their choice of meteorological profiles in control and perturbed cases. They say about this in supplemental, but very briefly. SAGE itself uses meteorological

profiles from a reanalysis for its retrievals. Combining vertical profiles from different pieces may be unnecessary.

Reply 4: We appreciate your assessment that our manuscript is well-written and logically organized. In response to concerns, we have replaced SWV with SH₂O to avoid confusion with shortwave SW.

It is important to clarify that different perturbed temperature profiles were not employed for the HTHH and CLIM calculations. Instead, we maintained consistent temperature profiles for both periods. This method involves subtracting identical background temperature profiles from both HTHH and CLIM, unrelated to adjusted temperature profile calculations. Therefore, it's crucial to note that this method does not lead to an overestimation in the instantaneous calculation of radiative forcing (RF).

Retrieving 3D information (latitude, longitude, altitude) for three perturbed species (H₂O, O₃, and aerosol extinction) throughout the lower troposphere (e.g., below 9 km in the tropics) is hindered by limitations of the SAGE-III instruments, based on solar occultation techniques. To maintain consistency with perturbed species, we used meteorological profiles above 9 km from SAGE-III and below 9 km from ERA5 reanalysis data. Identical 3D seasonal climatological mean meteorological profiles were applied for both HTHH and CLIM, ensuring that our instantaneous RF calculations are not overestimated (Line 261-266).

Please also note that the SAGE team retrieved the temperature and pressure profiles based on the Oxygen band absorption spectrum (<https://eosps0.gsfc.nasa.gov/sites/default/files/atbd/atbd-sage-solar-lunar.pdf>; see Page 32 and Eq. 3.2.16). They do use and compared the retrieved temperature and pressure profiles from the Oxygen band absorption spectrum with global NCEP data (see Fig. 3.2.1 and Fig. 3.2.7 in <https://eosps0.gsfc.nasa.gov/sites/default/files/atbd/atbd-sage-solar-lunar.pdf>).

Specific comments:

L119: As I mentioned above, some similar studies are in the pipeline for publication.

Reply: We removed “first”. See L124.

L123-126: This should go to conclusions.

Reply: We removed it. We already have similar statement in the conclusion section.

L135: It is quite a short climatology. In addition, the post-eruption anomalies will include variability associated with the 2022-2024 period. The impact on a long-wave (LW) forcing might be significant.

Reply: Although we used all of the available SAGE-III dataset starting from June 2017, we agree that a period longer than five years for the climatology would have been preferable. Therefore, we utilized the same radio-occultation (SAGE-III) instrument to assess the pre- and post-eruption response of the Hunga eruption, in order to avoid any biases that might arise from merging climatology datasets from different instruments such as MLS (see revised **Lines #144-150** and **Lines #131-138** in Main Manuscript).

We compared the 5-year climatology observed by SAGE-III with the 15-year climatology observed by MLS. On average, the background value of water vapor of 5 ppmv matches well with the MLS-based 15 Climatology. Also on average, the background values of ozone anomalies align well with the corresponding 15-year anomalies based on MLS data. (For ozone, see MLS-based data in Supplementary

Figures 16 and SAGE-III-based data in Supplementary Figure 10; for water vapor, see MLS-based data in Supplementary Figures 16 and SAGE-based data in Figure 1).

Additionally, we chose to use the same instrument, SAGE-III, because it provides comprehensive information on H₂O, aerosol extinction at nine wavelengths, and ozone mixing ratio. This is another reason why using the available 5-year dataset makes sense from SAGE-III (**Lines 144-150**).

Note that aerosol extinction at nine wavelengths is unique for assessing aerosol size distribution, which is not possible with other instruments such as OMPS, which focuses on observing the 756 nm ozone region.

The impact on long-wave (LW) forcing will be significant if we were estimating the adjusted RF. Instead, we are only assessing the instantaneous RF due to SH₂O only, SAOD only, and O₃ only (**L261-266**).

L169: Not by zonal wind.

Reply: We simplified it as “stratospheric wind” (**Line 171**).

L209: It is mixing ratio, not concentration.

Reply: Thank you for pointing that out. Corrected.

L229: You calculate not instantaneous but adjusted radiative forcing as you account for temperature change between the climatology and the 2022-2024 period. You also have to mention that all forcings are clear-sky.

Reply: We thank the reviewer for the comment. We use the climatology of temperature profiles for both HTHH-2022 and HTHH-2023 and thus do NOT account for stratospheric temperature changes (in particular the stratospheric cooling following the eruption). In the atmospheric background profiles for these years, we only change the H₂O mixing ratio, O₃ mixing ratio, and aerosol levels. We do not change the temperature profiles; they remain consistent with the climatological mean values for both HTHH-2022 and HTHH-2023. Thus, we estimate the instantaneous radiative forcing, not the adjusted radiative forcing.

We made this clearer in the methods section by specifying that “only” the change in stratospheric water vapor (H₂O) is considered in the caption. We further clarifies this point as “This increased net instantaneous radiative flux is calculated based solely on changes in SWV, while the background climatology of the temperature profile remains the same for both pre-eruption and post-eruption periods ” (see **Lines 262-263**).

Fig2: The figure caption is mixed up. Please correct.

Reply: We corrected it.

L271: It isn't very clear how to compare this with the background in this way.

Reply: We have revised this text to clarify that this is a ~400% change compared to the background climatology (See revised Supplementary Figure 2).

L298: A weak forcing cannot produce a strong interhemispheric asymmetry.

Reply: We agree that a weak forcing cannot produce a strong interhemispheric asymmetry. We have removed the word strong in the revised text (see L327).

L309: Wet in the stratosphere?

Reply: Thanks. We removed it. We added gravitational settling processes, dry deposition, stratospheric dynamics, and chemistry^{22,40} (Line 348).

L313: "underestimated"

Reply: Thanks. Corrected (Line 343).

L365-366: MEE is per aerosol mass. Figure 6 in supplemental shows MEE for 521 nm. It will be much flatter if you average over the entire solar spectrum. Half the solar energy comes in the near IR band between 750 and 4000 nm. Also, your effective radius for the Hunga eruption is relatively tiny compared to other estimates. Equation (1) in the supplement is miswritten. Please correct.

Reply. We agree that averaging MEE over the entire shortwave (SW) spectrum will result in a flatter curve. However, MEE at 521 nm is significant because the small size of the sulfate aerosols means that they interact much more efficiently with light in this part of the spectrum than with light in the IR band. These interactions play a crucial role in understanding radiative forcing and the Earth's energy balance. Thus, we show the MEE at 521 nm to allow for better interpretation of results and to facilitate useful comparisons of MEE per aerosol mass.

We thank you for pointing that the Equation (1) in the supplement is miswritten. We corrected it (see L227 in the supplementary Information).

L589: MEE of SO2 sounds confusing.

Reply: Thank you. We removed it (L593).

L595: "SO2 aerosol density"?

Reply: Thank you. We removed "aerosol" and called it as "SO2 density" (L599).

L599: Remove "the"

Reply: Done (revised tracked change manuscript).

L606: It is 75% of H2SO4 solution.

Reply: Thank you. We corrected it (L699).

L632: I doubt the ground-based AERONET station could see the stratospheric effects well, as they sense through the denser tropospheric aerosol layer.

Reply. We corrected the citation (L75-727). We agree that ground-based AERONET stations might not adequately capture stratospheric effects due to their sensing through the denser tropospheric aerosol layer. But a strong perturbation in the stratospheric aerosol concentration can well indicate in the total AOD values.

The total AOD values were substantially higher following the eruption compared to background levels, particularly in the Southern Hemisphere near the eruption sites as shown by **Boichu et. al. (2023; Line 991-995)**. They utilized around 600 ground-based AERONET stations to assess changes in the cloud-screened total Aerosol Optical Depth (AOD). Additionally, they derived size distribution information for both the coarse and fine modes of AOD values.

We have now accurately cited the study by Boichu (2023) with the first year post eruption stratospheric aerosol size distribution analysis [<https://agupubs.onlinelibrary.wiley.com/doi/10.1029/2023JD039010>].

L681: Discuss the meteorological profiles here.

Reply. Thank you for your suggestion. To enhance clarity regarding the changes in meteorological profiles, we have included the following sentences in the revised version (**L772-777**): "For both the HTHH-2022 and HTHH-2023 eruption periods, the basic background 5-year mean meteorological profiles, such as temperature profiles, remain unchanged. Consequently, the radiative forcing calculations primarily involve instantaneous radiative forcing calculations. The only perturbations made are to the three radiatively significant stratospheric species: stratospheric water vapor, stratospheric sulfate aerosol, and ozone."

L688-694: What was the spectral resolution?

Reply. We used REPTRAN Coarse mode at a spectral resolution width of 15 cm^{-1} . We also compared the results using REPTRAN Fine mode at 1 cm^{-1} . The magnitude of spectrally-integrated values for net radiative forcing at LW (longwave) and SW (shortwave) remain the same. The spectral resolution parameterizations for coarse, medium, and fine modes are provided in Gasteiger et al., (2014). <https://www.sciencedirect.com/science/article/pii/S0022407314002842>.

Gasteiger, J., Emde, C., Mayer, B., Buras, R., Buehler, S., and Lemke, O.: Representative wavelengths absorption parameterization applied to satellite channels and spectral bands, *J. Quant. Spectrosc. Radiat. Transfer*, 148, 99–115, doi:10.1016/j.jqsrt.2014.06.024, 2014.

L714: SAGE also takes meteorological profiles from a reanalysis

Reply. The SAGE team retrieved the temperature and pressure profiles based on the Oxygen band absorption spectrum (<https://eospsa.gsfc.nasa.gov/sites/default/files/atbd/atbd-sage-solar-lunar.pdf>; see Page 32 and Eq. 3.2.16). They do use and compared the retrieved temperature and pressure profiles from the Oxygen band absorption spectrum with global NCEP data (see Fig. 3.2.1 and Fig. 3.2.7 in <https://eospsa.gsfc.nasa.gov/sites/default/files/atbd/atbd-sage-solar-lunar.pdf>).

L722: I believe you calculate adjusted (not instantaneous) radiative forcing. Please make it clear.

Reply. We addressed this comment above. See our reply to your comment under L681.

References:

16. Duchamp, C., Wrana, F., Legras, B., Sellitto, P., Belhadji, R. and von Savigny, C. Observation of the aerosol plume from the 2022 Hunga Tonga—Hunga Ha'apai eruption with SAGE III/ISS. *Geophysical Research Letters*, 50(18), p.e2023GL105076 (2023).
17. Boichu, M., Grandin, R., Blarel, L., Torres, B., Derimian, Y., Goloub, P., Brogniez, C., Chiapello, I., Dubovik, O., Mathurin, T. and Pascal, N., 2023. Growth and global persistence of stratospheric sulfate aerosols from the 2022 Hunga Tonga—Hunga Ha'apai volcanic eruption. *Journal of Geophysical Research: Atmospheres*, 128(23), p.e2023JD039010.
15. Schoeberl, M., Wang, Y., Taha, G., Zawada, D.J., Ueyama, R. and Dessler, A., 2024. Evolution of the Climate Forcing During the Two Years after the Hunga Tonga-Hunga Ha'apai Eruption. *Authorea Preprints*.
47. Sellitto, P., Siddans, R., Belhadji, R., Carboni, E., Legras, B., Podglajen, A., Duchamp, C. and Kerridge, B., 2023. Observing the SO₂ and Sulphate Aerosol Plumes from the 2022 Hunga Tonga-Hunga Ha'apai Eruption with IASI. *Authorea Preprints*.

Reviewer #2 (Remarks to the Author):

The Hunga volcanic eruption was unprecedented in its impact on the stratosphere, and studies on this topic represent important and appropriate content for Communications Earth & Environment. This paper focuses on the radiative impact of changes to water vapor, aerosol, and ozone in the two years following the Hunga volcanic eruption and the associated changes in temperature. I would support the publication of this paper upon consideration of the suggested changes below.

Reply: We appreciate your thorough evaluation of our manuscript and your conclusion that it merits consideration for publication in this journal after addressing your comments. We have addressed all your comments point by point below.

*Major Comments:

1) The reading of the paper could be substantially improved by having the figure ordering match the order in which the discussion proceeds in the text. The paper jumps around referencing panels from Figs 1, 2, 3, and 4 often out of order and requiring the reader to view individual panels from 2 or 3 different figures simultaneously. Why not regroup the figure panels, so that they match the order in which the paper is written? It would substantially improve the readability of the paper. On a related note, I would suggest saving the discussion of Figure 4 until after the discussion of Figures 1-3 is complete rather than calling Fig 4 from the earlier SWV and aerosol sections without sufficient explanation.

We thank you for above suggestion. We rearranged the Figures 1-3 based on species types and write-up as per the suggestions to improve the text readability.

2) The authors need to decide if their primary conclusion is that Hunga cooled the SH or cooled the planet. The title and abstract are limited to the SH. But the main paper, e.g., Lines 124-125 and Line 470 take a stronger position and say that the planet was cooled.

Reply: We appreciate your comment. We have made revisions and now indicate that Hunga cooled the "SH" in 2022 and 2023. As suggested by Reviewer #1, we have moved lines 124-125 (Overall, we conclude that the Hunga eruption did not warm^{5,12}) from the end of the introduction into the conclusion. We have also corrected lines L539-544: "In conclusion, we show that the Hunga eruption did not cause warming in the SH or globally between 2022 and 2023. Instead, it had a cooling effect in the SH. The efficient conversion of SO₂ into sulfate aerosol in a water-rich stratosphere in the SH likely caused the Hunga eruption to produce a net cooling effect at the Earth's TOA in 2022 and 2023. However, in 2023, we find a slight warming in the NH due to the increased SH₂O and ozone, with minimal perturbation of SAOD".

3) The discussion of ozone in the manuscript needs revision. The statements on lines 149-152, 245, 389-391, and 403-404 (as well as Supplemental lines 61-62) are misleading. MLS data unequivocally show that for certain latitudes and altitudes, there were deviations in ozone following the Hunga eruption outside the range of any previous observations in the satellite data record.

See for example Figure 6 and Supplemental Figure S10 in Wilmouth et al. (your reference 28), Figure 8 in Wang et al. (Stratospheric climate anomalies and ozone loss caused by the Hunga Tonga-Hunga Ha'apai volcanic eruption. J. Geophys. Res.: Atmos., 128, e2023JD039480, 2023), and Figure 1 in Zhang

et al. (Chemistry contribution to stratospheric ozone depletion after the unprecedented water-rich Hunga Tonga eruption. *Geophys. Res. Lett.*, 51, e2023GL105762, 2024). As shown in these papers, the observed ozone losses at certain SH latitudes and altitudes in the second half of 2022 were unprecedented and well outside the range of normal variability, contrary to what is currently stated in the Gupta et al. manuscript.

I would speculate that perhaps the reason Figure 2 in this paper doesn't see the unprecedented ozone loss more clearly is because such a broad latitude range (0-60 S) is being averaged, so the strong midlatitude loss in the second half of 2022 is diluted by averaging in the tropics. Or perhaps there is some other issue with the analysis in this manuscript. Regardless of the reason, this manuscript should be careful not to make overly dismissive statements regarding the observed ozone reductions following Hunga, because the ozone losses were unprecedented in the 20-year MLS record for certain latitudes, altitudes, and times.

Also, please double check the manuscript and supplemental to clean up the confusion as to which cited references studied ozone depletion in the immediate days/weeks after the eruption and which studied ozone depletion in the months/year that followed the eruption.

Reply: We appreciate your suggestion regarding the potential ambiguity in the ozone loss signal from averaging data between the equator and mid-latitude regions. Following your advice, we have significantly revised the text and incorporated extensive new analysis for ozone anomaly calculations, following the methodologies of Wilmouth et al. (2023) and Santee et al. (2023). Our results show an ozone loss in the first half of 2022, followed by an ozone gain in 2023 (e.g., Zhang et al., 2024; Raymond et al., 2024), which can be qualitatively attributed to a combination of meteorological and chemical factors (Wang et al., 2023; Santee et al., 2023).

We have added new Figure 4 and associated texts (Lines 134-139; Lines 331-338; Lines 406-444), including new supplementary Figures 14-19 (Lines 181-217), related to ozone changes after Hunga eruption. Using the 2D-Filtered technique, we show that our anomaly analysis with SAGE-III data (Supplementary Figure 4) qualitatively aligns with the results obtained using AURA-MLS (Supplementary Figure 18) data from 2007 to December 2023 (Lines 134-138).

In summary, we conducted radiative forcing calculations for all three perturbed species. Although the magnitude of radiative forcing has changed slightly, the main conclusion of the paper remains unchanged from the previous revision. It is important to note that our primary focus was on quantifying the vertically-latitude-perturbed ozone following the Hunga eruption. We successfully determined that the eruption had an impact on ozone loss in 2022 and subsequent recovery in 2023. This quantitative analysis was carried out using a similar approach as in the study by Wilmouth et al. (2023) and Santee et al. (2023). We would like to emphasize here that we are not assessing the exact physical processes (e.g., stratospheric dynamics, chemistry, secondary circulation from radiative feedback) related to ozone gain in 2023, as this can only be answered through observational analysis combined with detailed stratospheric-chemistry equipped climate modeling. This manuscript is primarily dealt with answering the combined instantaneous radiative forcing of three perturbed species associated with Hunga eruption.

4) Lines 312-330 and Lines 355-386. I believe it is already well established that the excess water vapor from Hunga is what led to significantly enhanced stratospheric aerosols relative to what might be expected given the small amount of SO₂ injected. Several papers have discussed this. Just a couple off the top of my head:

Y. Zhu et al., Perturbations in stratospheric aerosol evolution due to the water-rich plume of the 2022 Hunga-Tonga eruption. *Commun. Earth Environ.* 3, 248 (2022).

Asher, E., et al., Unexpectedly rapid aerosol formation in the Hunga Tonga plume. *Proc. Nat. Acad. Sci.*, 120, e2219547120 (2023).

Reply: We thank you for your suggestion. We have also cited these papers and revised the text to cite them at the mentioned lines. We also include a citation to a recent study that analyzed the long-term aerosol growth using a sectional aerosol model and found results very much consistent with our conclusions (Chenwei et. al., 2024).

Li, Chenwei, et al. "Microphysical simulation of the 2022 Hunga volcano eruption using a sectional aerosol model." *Geophysical Research Letters* 51, no. 11 (2024): e2024GL108522.

5) Lines 780-784. The choice of a 20 km cutoff is fine for water and ozone, but it almost certainly leads to loss of data for the aerosols, which are known to have descended relative to the water plume. Other papers have shown this, as does Figure 1e and 1f in this manuscript. When the radiative forcing results are presented, the manuscript should acknowledge that the results of this study will be biased because of the 20-km altitude cutoff and attempt to estimate the uncertainty introduced.

Reply: In the limitation section, we acknowledge that the 20 km altitude cutoff may limit the aerosol data, which could slightly underestimate the cooling calculations of radiative forcing. However, we chose the 20 km cutoff to ensure that all three species are at the same vertical height for combined radiative calculations (L882-885).

*Additional comments:

Line 42. Add “volcanic” after “submarine”

Reply: Added.

Line 43. Delete “,which warms the climate,”. It interrupts the flow of the sentence and is unnecessary because this point is repeated in the next sentence.

Reply: Deleted.

Lines 101-103. The sentence beginning with “If water vapor’s warming effects...” should be revised. It is an overstatement as written. The climate impact of any future submarine eruptions will depend on the relative amount of water and SO₂. Not all submarine eruptions are the same.

Reply: Done. See revised L(103-105).

Line 108. Don’t start paragraph with “However”.

Reply: Done.

Lines 135-136: Why define CLIM as “7 June 2017 – 9 Dec 2021”? The dates seem completely random. Please explain this choice in the paper. Most studies I’ve seen go back much farther than 2017 to establish the background.

Reply: The SAGE-III observational data has been available since June 2017. We are only using SAGE-III data and not including any data from previous SAGE-I or II instruments. This decision is based on SAGE-III's ability to observe across multiple wavelengths. There was a gap in the data from December 9

to 19, 2021. Also, starting from December 19, 2021, we are excluding data due to the initial eruption of Hunga on that date. Thus, background calculations are taken from June 7, 2017, to December 9, 2021 (Line 144-150; Lines 609-616).

Line 161. Change HTHH-2022 to HTHH-2023.

Reply. Done.

Line 183. Change December 2022 to January 2022.

Reply. Done.

Line 191. Reference 28 (Wilmouth et al., 2023) could be added here, as it shows the importance of BDC on water vapor distribution from Hunga.

Reply. Done.

Line 209 and other places. Don't say "concentration" when the units are ppmv or ppbv. Change to "mixing ratio".

Reply. Done.

Figure 2c. Units are missing on y axis.

Reply. SAOD does not have any units. It is new Revised Figure 5, where we added the term "unitless"

Figure 2e. Please change the x axis to make it less cluttered and easier to read.

Done. (see Revised Figure 2)

Line 305. known to produce

Reply. Done.

Line 428. all two?

Reply. Done.

Lines 505-796. The Methods section is extraordinarily long. I would recommend moving most of this content to the Supplemental. I'm guessing the journal will require this.

Reply. We thank you for the suggestion. We have slightly trimmed the method sections and also provided minimum information in the Main Methods section for the readers with more details in the supplement, *A few comments on the Supplemental:

Line 64. "However, there is no clear signal afterwards." This statement is false. The largest ozone losses were observed in the June – December 2022 timeframe. See the papers highlighted in major comment #3 above.

Reply. Corrected everywhere.

Line 155. NO₂ decreases, not increases.

Reply. We thank you for pointing this out. Corrected.

Figure S2 is essentially unreadable on a print copy. The panels are too small. Also, some of the panels are never listed in the caption.

Reply. We have revised Figure S2 accordingly. We are only showing the relative changes plots. We hope that it is more clear now.

References in Main manuscript

6. Zhu, Y. et. al. Perturbations in stratospheric aerosol evolution due to the water-rich plume of the 2022 Hunga-Tonga eruption. *Communications Earth & Environment*, 3(1), 248 (2022).
31. Wilmouth, David M., Freja F. Østerstrøm, Jessica B. Smith, James G. Anderson, and Ross J. Salawitch. "Impact of the Hunga Tonga volcanic eruption on stratospheric composition." *Proceedings of the National Academy of Sciences* 120, no. 46 (2023): e2301994120.
32. Wang, X. et. al. 2023. Stratospheric Climate Anomalies and Ozone Loss Caused by the Hunga Tonga-Hunga Ha'apai Volcanic Eruption. *Journal of Geophysical Research: Atmospheres*, 128(22), p.e2023JD039480.
33. Zhang, Jun, Douglas Kinnison, Yunqian Zhu, Xinyue Wang, Simone Tilmes, Kimberlee Dube, and William Randel. "Chemistry contribution to stratospheric ozone depletion after the unprecedented water-rich Hunga Tonga eruption." *Geophysical Research Letters* 51, no. 7 (2024): e2023GL105762.
34. Santee, M.L., Lambert, A., Froidevaux, L., Manney, G.L., Schwartz, M.J., Millán, L.F., Livesey, N.J., Read, W.G., Werner, F. and Fuller, R.A., 2023. Strong Evidence of Heterogeneous Processing on Stratospheric Sulfate Aerosol in the Extrapolar Southern Hemisphere Following the 2022 Hunga Tonga-Hunga Ha'apai Eruption. *Journal of Geophysical Research: Atmospheres*, 128(16), p.e2023JD039169.
35. Asher, E., Todt, M., Rosenlof, K., Thornberry, T., Gao, R.S., Taha, G., Walter, P., Alvarez, S., Flynn, J., Davis, S.M. and Evan, S., 2023. Unexpectedly rapid aerosol formation in the Hunga Tonga plume. *Proceedings of the National Academy of Sciences*, 120(46), p.e2219547120
36. Legras, B., Duchamp, C., Sellitto, P., Podglajen, A., Carboni, E., Siddans, R., Grooß, J.U., Khaykin, S. and Ploeger, F., 2022. The evolution and dynamics of the Hunga Tonga–Hunga Ha'apai sulfate aerosol plume in the stratosphere. *Atmospheric Chemistry and Physics*, 22(22), pp.14957-14970.
37. Fleming, E.L., Newman, P.A., Liang, Q. and Oman, L.D., 2024. Stratospheric Temperature and Ozone Impacts of the Hunga Tonga-Hunga Ha'apai Water Vapor Injection. *Journal of Geophysical Research: Atmospheres*, 129(1), p.e2023JD039298.

57. Raymond, Neil, Peter Bernath, and Chris Boone. "Atmospheric effects of the Tonga volcanic sulfate aerosols." *Journal of Quantitative Spectroscopy and Radiative Transfer* (2024): 109056.

58. Zhang, J., Kinnison, D., Zhu, Y., Wang, X., Tilmes, S., Dube, K. and Randel, W., 2024. Chemistry contribution to stratospheric ozone depletion after the unprecedented water-rich Hunga Tonga eruption. *Geophysical Research Letters*, 51(7), p.e2023GL105762.

Cited in Supplementary Information

Davis, S.M., Damadeo, R., Flittner, D., Rosenlof, K.H., Park, M., Randel, W.J., Hall, E.G., Huber, D., Hurst, D.F., Jordan, A.F. and Kizer, S., 2021. Validation of SAGE III/ISS solar water vapor data with correlative satellite and balloon-borne measurements. *Journal of Geophysical Research: Atmospheres*, 126(2), p.e2020JD033803.

Wang, H.R., Damadeo, R., Flittner, D., Kramarova, N., Taha, G., Davis, S., Thompson, A.M., Strahan, S., Wang, Y., Froidevaux, L. and Degenstein, D., 2020. Validation of SAGE III/ISS solar occultation ozone products with correlative satellite and ground-based measurements. *Journal of Geophysical Research: Atmospheres*, 125(11), p.e2020JD032430.

Reviewer #3 (Remarks to the Author):

Review of “The January 2022 Hunga eruption cooled the southern hemisphere in 2022 and 2023” by Gupta et al.

Major Points

1. The authors argue that they can neglect ozone changes which their own results suggest is a non-negligible component of the climate forcing. Another recent study indicates that ozone change is an important component of the total flux change – and the authors own table indicate this. The authors need to clarify their discussion of the ozone change and they may have not considered the ozone response correctly.

Reply: We appreciate your feedback regarding potential ambiguity in interpreting ozone variability due to wide-band latitude averaging. Following your suggestion, we integrated ozone anomaly calculations following the approach of Wilmouth et al. (2023) and Santee et al. (2023) (see **methods section; L650-679; Lines 406-444**). Our findings confirm a period of ozone depletion during the first half of 2022, followed by subsequent ozone recovery in 2023, likely influenced by meteorological factors.

Our investigation identified ozone depletion in 2022, succeeded by ozone recovery (or gain) in 2023, supported by both MLS and SAGE-III observational data (**revised Figure 4, Supplementary Figure 10,18-19**). Notably, SAGE-III data provides better vertical resolution at 0.5 km but with more limited global coverage compared to MLS, which offers comprehensive global insights into various ozone-related species.

See our response to a similar question raised by Reviewer #2 under comments #3.

2. The FAIR parameterized climate model results are presented without any uncertainty temperature response. I suspect if uncertainty is included the surface temperature impact would be close to zero.

Reply: We thank you for the above comment. We have now added the uncertainty related to surface temperature change (Line 56; Lines 841-846). Note that this standard deviation (or uncertainty) associated with changes in surface temperature, using the FaIR model, is estimated based on the propagated standard deviation for the combined instantaneous radiative effects of three stratospheric species. But we do agree that this idealized surface temperature change does not include the uncertainty related to dynamic changes related to radiatively-induced feedback effect and thus, it only addresses the simple uncertainty associated with radiative forcing calculations.

3. There are a number of important papers missing in the reference lists including Santee et al. (2023).

Reply: We have now added a reference to Santee et al. (2023) as well as several other additional papers, especially regarding the effect of the eruption on ozone (see Lines 1038-1081).

4. The author should justify not using OMPS-LP aerosols for SAOD and not using MLS for ozone, temperature and water vapor.

We now confirm the H₂O and O₃ perturbations following the Hunga eruption using both MLS (Supplementary Figures 16 and 17) and SAGE-III (Figs. 1-3 and Supplementary Figure 3) observations. The primary reason for utilizing SAGE-III data is to ensure consistency by using all three observations (sulfate aerosols, H₂O, and O₃) from a single instrument, thereby avoiding biases related to the number of measurements and spatio-temporal sampling (e.g., SAGE-III has a higher depth resolution compared to MLS). Additionally, SAGE-III offers observations across nine wavelengths, which is advantageous for retrieving the particle size of sulfate aerosols and assessing the associated radiative forcing (see also L609-616). We also show that our particle size estimations are well matching with different studies based on multi-instruments, such as MISR, CALIOP, OMPS-LP and ground-based observations (Knepp et al., 2023; Kahn et al., 2024; Duchamp et al., 2023; Boichu et al., 2024; Wrana et al., 2023 and others).

5. The long discussion trying to reconcile SO₂ and SAOD from previous eruptions is not convincing and distracts from the main focus of the paper.

Reply: We have reduced the text related to SO₂ and SAOD from previous eruptions.

Specific points.

Line 53 “mass extinction” ... just extinction

Reply: We have removed that part because the abstract is limited to only 200 words. However, we have retained the radius range of sulfate aerosol (Line 48).

But we meant to convey here that the extinction produced per unit mass is maximized for aerosols in the size range found for the Hunga eruption. We therefore think that “mass extinction efficiency” is the correct term here.

Line 87 also Dessler et al. (2013) www.pnas.org/cgi/doi/10.1073/pnas.1310344110

Reply. Cited

Line 95 you should include Fujiwara et al (2020) <https://doi.org/10.5194/acp-20-345-2020>.

Reply. Cited

Line 98 estimates of actual ozone depletion are shown in Santee et al. (2023) <https://doi.org/10.1029/2023JD039169> - the others are models.

Reply. Cited

Line 116 Schoeberl et al. (2024) has updated the calculations and extended them through 2023. This paper is a preprint available at <https://doi.org/10.22541/essoar.171288896.63010190/v1>

Reply. Cited. (Line 983-985)

This paper provides a similar but more comprehensive trace gas calculation than this work and computed the tropopause flux relative to 10 years of MLS observations.

Reply. We now added an analysis using 17 years of MLS observations (also see our reply to Reviewer #1 for L135).

Line 130 The MLS instrument is on the Aura satellite not Aqua.

Reply. Thank you for pointing that out. Corrected.

Line 135 Why CLIM = 5 years and not 10 or 15 years of observations?

Reply: SAGE-III provides data from June 2017 onwards and that is why we only took five years of background calculations. Because of this, we checked the CLIM values for 15-years using MLS observations against the 5-year SAGE III climatology and the results are consistent (see Supplementary Figs. 3, 10, 16 and 17). Please see the comparative plots between 5th and 6th rows anomaly plots of O₃ from SAGE-III in Supplementary Figure 10 with in Supplementary Figure 16 within 60N/S. Similarly, H₂O anomaly from SAGE-III and AURA MLS is compared (also see our response to your earlier query about ozone).

Line 138 Specifically which species? I presume H₂O, aerosols and O₃. Please state for clarity. Note that N₂O is also radiatively active, but MLS observations show a very small N₂O perturbation. You might mention that your TOA flux implicitly includes temperature variations. Because of the stratospheric cooling, colder temperatures will reduce the tropopause flux and presumably the TOA flux as well.

Reply: Yes, these three species are H₂O, aerosols, and O₃. No, this does not implicitly include temperature variations, as we are using an idealized radiative transfer model to calculate instantaneous radiative forcing, not the adjusted radiative forcing. This is clarified more in the revised methods section (L261-266). Also see our response to general comment section from reviewer #1.

Line 144 Again refer to the Santee et al (2023) paper which is more comprehensive.

Reply: Done.

Line 148 A substantial change in southern extra-tropical stratospheric ozone in the July-Aug period occurs in mid 2022 due to the secondary circulation set up by the water vapor cooling. Wang et al.

(2023) <https://doi.org/10.1029/2023JD039480>

Schoeberl et al. (2024) shows that this mid 2022 ozone decrease increases the short wave flux at the tropopause. This component cannot be neglected as the authors have claim to have done here. On the other hand, some of the results include ozone changes and they appear to be non-negligible.

Reply: We agree with the above suggestion and have incorporated all the radiative forcing analysis related to ozone changes in the revised manuscript. Please see our reply to your major comments.

Line 177 “The widespread vertical distribution of SWV in 2023, in contrast to 2022, is attributed to strong changes in stratospheric circulation patterns and responses from localized heating effects.” Based on what evidence? Compared to the normal Brewer Dobson circulation, the Hunga induced circulation appears to be small (Fleming et al., 2023 and others). The westerly phase QBO descended in a normal fashion.

Reply. We thank the reviewer for pointing out this out – we did not mean to imply that the vertical distribution of SWV was primarily because of Hunga induced circulation. We have revised the text as

“The widespread vertical distribution of SH₂O in 2023, in contrast to 2022, is attributed to the stratospheric Brewer-Dobson circulation in combination with some perturbations in stratospheric circulation patterns due to stratospheric temperature perturbations following the Hunga eruption^{32,37} (L179-182)”.

32. Wang, X. et. al. 2023. Stratospheric Climate Anomalies and Ozone Loss Caused by the Hunga Tonga-Hunga Ha'apai Volcanic Eruption. *Journal of Geophysical Research: Atmospheres*, 128(22), p.e2023JD039480.

37. Fleming, E.L., Newman, P.A., Liang, Q. and Oman, L.D., 2024. Stratospheric Temperature and Ozone Impacts of the Hunga Tonga-Hunga Ha'apai Water Vapor Injection. *Journal of Geophysical Research: Atmospheres*, 129(1), p.e2023JD039298.

Line 182 “seasonal-mean value between 2017 and 2021” you mean the seasonal 5 year climatology starting in 2017?

Reply. Corrected.

Line 183-193 I am curious on why the authors use SAGE rather than the more abundant and high quality V5 MLS data. SAGE is low spatial resolution and can't measure in polar night. The MLS data set is not large nor difficult to work with ...

Reply: We confirm the H₂O and O₃ perturbations following the Hunga eruption using both MLS and SAGE-III observations. The primary reason for utilizing SAGE-III data is to ensure consistency by using all three observations from a single instrument, thereby avoiding biases related to the number of measurements. Additionally, SAGE-III offers observations across nine wavelengths, which are suitable for retrieving the particle size of sulfate aerosols and assessing the associated radiative forcing (Line 609-616).

I also don't agree with the comment in the caption that ozone anomaly is negligible. You can clearly see in this (low resolution) diagram that 2022 SON ozone is below normal. Averaging 0-60S reduces the amplitude of the change. The ozone anomaly would be more evident if the authors used MLS data and looked at Santee et al. (op. cit.) or Wang et al. (op. cit.)

Reply: Corrected. See our reply to your Major comment #1.

Line 190-191 The authors should include more recent references which actually analyze Hunga data rather than Foster et al. and Solomon et al.

Reply: We cited these new references, as suggested.

Line 201 The use of SW for short wave and SWV for stratospheric water vapor is confusing.

Reply: That's a good point. We have changed it from SWV to SH₂O.

Line 205 Schoeberl et al. (2022) <https://agupubs.onlinelibrary.wiley.com/doi/10.1029/2022GL100248> was the first to note that the local temperature decrease after the eruption was consistent with H₂O radiative cooling.

Reply. Cited (see L83).

Line 227 Reword, confusing.

Reply. Done.

Line 228 Is this TOA flux?

Reply. Yes. TOA radiative flux. Added.

Line 230 I think reporting the surface temperature anomaly due to water vapor without giving an uncertainty is a little mis-leading. I suspect the uncertainty (usually reported by climate models) will likely be larger than the surface warming value (+0.02 K). Also, I think that the extra SWV emitted photons are going to be absorbed at the cold tropopause and not make it to the surface. This comment also applies to Line 203 where an even more ridiculously small warming for 2023 is estimated.

Reply: See our response to your comment #2 in major points.

Line 258 Recent paper by Sellitto suggest 0.7 Tg as the authors note in Line 314

Reply: See our response to Reviewer #1 for Q3.

Line 272 SAGE 5.3 provides an SAOD value which is integrated from the cloud top. Why did the authors choose to generate their own SAOD?

Reply: We aimed to estimate the radiative forcing calculations between 20 and 40 km, which is why we performed our own SAOD calculations. Similarly, H₂O and ozone measurements were taken within this altitude range.

Line 273 Units?

Reply: SAOD has no unit – hence this is correct.

Fig 3e. Silletto et al. (2022) computed the net aerosol heating and found that the long wave and short wave roughly cancelled. I don't believe the authors have included the long wave cooling of the aerosols in the net. Or if they have, it wasn't clear to me. Sato et al, J. Geophys. Res. 98 (1993) 22,987–22,994 show the importance of the LW and SW components of aerosol forcing.

Reply: Please do see the aerosol LW and SW calculations in Supplementary Figures 4 (for shortwave SW calculations) and 5 (for LW calculations) and then the sum is shown in the main manuscript. We sum the results of LW + SW for the net calculations (Figure 1, 2, 3, 4 and other NRHR/NRF figures). We are assessing the instantaneous net radiative forcing for aerosols (L948-950).

Line 309 Wet scavenging in the stratosphere? Unlikely. More likely is coagulation and settling of the aerosols.

Reply. We corrected it as “gravitational settling processes”.

Line 322 The difference in reduction rate due between water and aerosols is due to aerosols settling. As shown in multiple studies, water vapor moves upward on the BD circulation and aerosols settle out.

Reply. Thank you for pointing that out. We added a sentence based on the above comment.

Line 328 I am confused – did you use SAGE SAOD or did you compute your own SAOD (Line 272). This is a good point about the aerosol SAOD and SO₂ required, but there is some disagreement on SAOD. Another point is that the Aubery simulator uses ambient stratospheric water vapor, whereas Hunga had very elevated water vapor. This may have led to additional larger aerosols and higher SAOD. Finally, USask OMPS SAOD is about 0.01. I would drop this part since you are not on solid ground here.

Reply. We used our own SAOD calculation from SAGE-III data. We removed that statement (revised L348).

Line 346-386 Again, this is an interesting, but not all that relevant to the paper's focus. Linking the aerosol forcing to the SO₂ is dicey since SO₂ is uncertain in previous eruptions. Why do you focus on SO₂ when you have the measured SAOD? Also, you have neglected OMPS-LP SAOD. I think the case for higher SAOD/SO₂ distracts from the climate focus of the paper and the arguments more speculative than quantitative. I think you should write a separate paper on this material and delete it here.

Reply. We have removed these paragraphs from Line 346-386.

Line 390 Totally false as shown in Zhu et al. and Santee et al.

Reply. We agree. It was due to the wide-latitudinal band averaging that the ozone signal was lost. We revised the ozone analysis using 2D-filtered technique (see Lines 406-444; Fig. 4; Table 1; Lines 650-679) and show a loss in the 2022 and slight gain in 2023.

Line 401-409. This result disagrees with Schoeberl et al. (2024) who found ozone a contributor to SW forcing at the tropopause especially in mid to late 2022. You can see the ozone change in the highly averaged figures presented by the author. Even in Table 1 the ozone forcing is as large as the water vapor forcing – with ozone -0.49, without ozone -0.37 (ozone ~30%), unless I misread the table (which is poorly labeled).

Reply: To address this and similar comments, we have recalculated the ozone radiative forcing using the 2D-filtered technique, and our results are now consistent with Schoeberl et al., 2024. They show a global decrease in the downward radiative flux at the tropopause over a two-year period due to three species.

I am not sure that the ozone calculation is done correctly. The decrease in ozone should increase the shortwave flux at TOA. In fact this is how satellites measure ozone changes, the atmosphere outgoing flux is absorbed by ozone so if it increase there is an ozone decrease. But Table 1 seems to indicate the opposite there a larger decrease in outgoing flux when ozone is included. This might occur if you only considered the LW component of ozone.

Reply. We thank the reviewer for the comment but respectfully disagree with them. As we note in the supplementary (L120-125): “When the ozone layer in the stratosphere is depleted, it causes less absorption of solar radiation (shortwave), resulting in more radiation being upwelled to the Top of the Atmosphere. This leads to a net cooling effect at the Earth's TOA, contrasting with positive shortwave radiative forcing at the tropopause. It is also important to note that the longwave radiative forcing from ozone is negative at the Earth's TOA and tropopause, further contributing to the cooling effect. Thus, to first order, ozone depletion in the stratosphere leads to a negative radiative forcing at the TOA. This instantaneous radiative response of ozone depletion agrees with radiative transfer calculations in Shine et al., 2022 (see Table 5; therein) as well as Maycock et al. 2021 (see Figure 2).” See also, **L461-465 in the Main text.**

Shine, K.P., Byrom, R.E. and Checa-Garcia, R. Separating the shortwave and longwave components of greenhouse gas radiative forcing. *Atmos. Science Letters*, 23(10), e1116 (2022).

Maycock, A. C., Smith, C. J., Rap, A., & Rutherford, O. (2021). On the structure of instantaneous radiative forcing kernels for greenhouse gases. *Journal of the Atmospheric Sciences*, 78(3), 949-965 (<https://journals.ametsoc.org/view/journals/atsc/78/3/JAS-D-19-0267.1.xml>)

While including ozone does not change the conclusion that Hunga cooled the climate according to the above paper, I believe the authors should look again at ozone changes.

Reply. The RF calculations related ozone changes are indeed now incorporated in the revised version (as discussed above).

Line 425 Please include uncertainty in surface cooling estimate.

Reply. We now added this, based on the RF calculations.

Line 436 -0.37 a global instantaneous value or the peak following the eruption.

Reply. It (-0.38 with ozone) shows a global instantaneous value (added in L613).

Line 464 I assume you are arguing that submarine eruptions are different from subaerial eruptions based on the SO₂-> aerosol conversion. Now Pinatubo had plenty of water (50 Tg) but the eruption was low enough that all the water froze out. The difference between Hunga and Pinatubo is that the eruption went high enough that the water did not freeze out and was available for aerosols. So the difference is more likely the altitude of the eruption rather than submarine versus subaerial.

Reply. We thank the reviewer for the comment – we agree with the reviewer that the effect of the Hunga eruption differs from Pinatubo in part due to the plume height that in turn affected the radiative impact through amount of available water. But, this difference in the height of the eruption and the water content is likely because of the submarine nature of the eruption (e.g., see Mastin, L. G., Van Eaton, A. R., & Cronin, S. J. (2024). Did steam boost the height and growth rate of the giant Hunga eruption plume?. *Bulletin of Volcanology*, 86(7), 64.) – furthermore, the nature of the volcanic plume for a submarine eruption was different and this had a key role in determining the amount of SO₂ and halogens that made it into the stratosphere (Carn et. al., 2022). We have modified the text to clarify this and tweak the language accordingly (L534-546).

Line 474 Even though most of Hunga water remains in the stratosphere and mesosphere, the Hunga injection was only about 13% of the total stratospheric water. Most of this water is above 30 km so looking at the weighting function in Solomon et al. (2010) the impact will be tiny.

Reply. We do agree with the above statement. But injected water vapor at higher height could have perturbed the ozone concentrations and hence it can also indirectly influence the surface temperature warming.

REVIEWER COMMENTS:

Reviewer #1 (Remarks to the Author):

I thank the authors for the thorough explanation of their approach. From their answers, it is clear that the paper presents the instantaneous radiative forcing of three major optically active constituents: stratospheric Water Vapor (WV), Sulfate Aerosols, and Ozone. I still believe it is a valuable and relevant study, but I am afraid the paper in its current form contains a technical error. The longwave (LW) instantaneous radiative forcing (IRF) of WV at the top of the atmosphere (TOA) can not be positive. Adjustment of the stratospheric temperature changes the sign of the forcing, so an adjusted Radiative forcing (ARF) of WV at TOA is positive. At the same time, IRF at tropopause is positive. The two-level energy balance model, used to evaluate a climate response, requires the forcing at the tropopause. I suggest the authors reconsider their calculation of WV LW IRF and spare a paragraph in the paper to discuss IRF at the tropopause and the surface. This will significantly increase the value of their paper.

We sincerely thank you for raising these important concerns. After conducting a rigorous analysis and reproducing some of the results from previously published works (e.g., radiative kernel and TOA and Tropospheric forcings for stratospheric water vapor), we have identified and resolved the issue in the revised manuscript. We acknowledge that there was a technical error in our original calculations and sincerely apologize for this oversight in the initial submission. Following these changes, we have updated the results for both the TOA and 16 km levels for water vapor, stratospheric aerosols, and ozone.

In the revised analysis, we now clearly observe a negative IRF at the TOA and a positive IRF at 16 km and 11 km for stratospheric water vapor in the upper/middle stratosphere (> 20 km height). After rigorously rechecking our calculations, we identified and corrected a critical technical error: the incorrect assignment of a zero H₂O mixing ratio at unperturbed locations (i.e., during CLIM conditions). This error had previously caused an inaccurate positive IRF at the TOA. Essentially, this mistake led to a systematic underestimation of IRF values under background conditions, which produced the erroneous positive IRF at the TOA. With this correction, our results are now fully aligned with the expected physical behavior of negative IRF of perturbed stratospheric water vapor at the TOA and a positive IRF at 16 km (near equatorial tropopause; see Figure 1 and Supplementary Figure 3 and Supplementary Figures 26-30).

We have rechecked and verified our revised results with the findings in Solomon et al. (2010), Maycock et al. (2021), and Wang & Huang (2024). Our corrected results are now consistent with their findings, as demonstrated in Supplementary Figures 27-30 and the updated text in the Supplementary Note. We have also checked and updated our ozone calculations to ensure that a similar error doesn't affect our results.

In the revised text, we have comprehensively addressed this issue throughout the manuscript. The revised figures (Fig. 1-5, Table 1) and Supplementary Figures 26-30 now reflect the corrected analysis. We also included net radiative forcing kernels at three levels: TOA, 16 km, and 11 km in Supplementary Figures 26-30. With the updated results, we find that our main

conclusion—regarding the combined negative net radiative forcing of Hunga activity in both 2002 and 2023—remains unchanged, though the IRF magnitudes do change.

Once again, we greatly appreciate your valuable feedback, which has helped improve the accuracy of our findings and identifying the error in the original results.

References:

Solomon, S. et. al. Contributions of stratospheric water vapor to decadal changes in the rate of global warming. *Science*, 327(5970), 1219-1223, (2010).

Wang, Y. and Huang, Y., 2024. Compensating atmospheric adjustments reduce the volcanic forcing from Hunga stratospheric water vapor enhancement. *Science Advances*, 10(32), p.ead12842.

Maycock, A. C., Smith, C. J., Rap, A., & Rutherford, O. (2021). On the structure of instantaneous radiative forcing kernels for greenhouse gases. *Journal of the Atmospheric Sciences*, 78(3), 949-965.

Reviewer #2 (Remarks to the Author):

I find the manuscript to be improved from the previous draft. I would support the publication of this paper after implementing the changes below.

1) The focus of this paper is on radiative effects. As such, all that's needed for ozone is to quantify the ozone changes, not explain why ozone changed. No reader of this paper will expect the authors to explain why ozone changed in 2023. The authors themselves state that explaining the ozone loss is out of scope for the present paper on lines 440–444. And yet, despite this, the authors somewhat inexplicably spend significant text speculating about why and how ozone changed: lines 426–438 of the manuscript and lines 132–174 of the Supplemental. I strongly encourage the authors to delete all of this. There are many mistakes in what the authors are saying, particularly in the Hunga-Ozone interaction section of the Supplemental and the interpretation of Figure S19. It will not compromise the story of the paper in any way to delete these parts and, it will significantly improve the paper by removing the inaccurate and speculative statements.

We appreciate your feedback and agree that the speculation regarding the ozone changes goes beyond the scope of the paper. We removed the sections discussing ozone change speculation on lines 426–438 of the manuscript and lines 132–174 of the Supplementary materials. We also removed any other speculative statements as advised by Reviewer #3.

2) Figure 5d and Lines 486–503 need additional clarification. It is stated that the radiative forcing of -0.38 Wm^{-2} caused a surface cooling of -0.05 K in 2022. How is it then that a radiative forcing of only -0.01 Wm^{-2} caused the exact same amount of surface cooling of -0.05 K in 2023. Why didn't the delta T become less negative in 2023 in Figure 5d when the radiative forcing was smaller? This should be discussed in the manuscript.

We thank you for your insightful question. The apparent discrepancy between the radiative forcing values and the corresponding surface temperature changes in 2022 and 2023 is indeed worth clarifying.

While a radiative forcing of -0.51 W/m^2 at TOA led to a surface cooling of $\sim -0.07 \pm 0.01 \text{ K}$ in 2022, the same surface temperature response in 2023 ($\sim -0.06 \pm 0.01 \text{ K}$) with a much smaller radiative forcing (-0.17 W/m^2) at TOA is very likely because of the inertia in the climate system. That is, the temperature change depends partially on the cumulative radiative forcing over time. This means that even though the radiative forcing in 2023 is small, the temperature change in 2023 also reflects the residual effects of the larger forcing from 2022, resulting in a similar temperature change (see Lines 570-573 in the revised main text).

3) I wonder about the ozone filtering for 2023. Lines 651–679 describe that an ozone change was only linked to Hunga if water was simultaneously greater than 2 sigma from the mean, based on

the method of Wilmouth et al. But Wilmouth et al. only used that method for 2022, when most of the sulfate and water vapor was still in the Southern Hemisphere. This approach may not work as well for 2023, as done in the Gupta et al. manuscript. Even for 2022, Wilmouth et al. were concerned about the separation of the aerosol layer and water vapor and thus tested their results over different pressure levels; and it is important to note that they were just looking for the peak anomalies, which is a different analysis than what is being done in the present manuscript. In short, the authors of Gupta et al. should check for 2023 that they aren't filtering out ozone data where there were perturbations due to Hunga from elevated sulfate in the absence of substantially elevated water vapor.

We appreciate your concern regarding the applicability of the Wilmouth et al. (2022) methodology for 2023. You are correct that Wilmouth et al. (2022) focused on 2022, when most of the sulfate and water vapor were confined to the Southern Hemisphere.

To ensure that we are not inadvertently filtering out ozone data associated with the Hunga eruption, particularly in cases where elevated sulfate aerosols were present without significantly elevated water vapor, we applied the ozone filtering technique both with and without the water vapor criterion for 2023. Our analysis showed that there was no significant difference in the ozone gain and loss linked to the Hunga eruption between these two approaches (see Supplementary Figure 20). The largest difference was observed in the Southern Hemisphere, where the deviation was just 0.25-0.5 DU during DJF and MAM of 2023. Importantly, no changes were observed in the Northern Hemisphere or near-global domains (see Supplementary Figure 20).

4) I continue to contend that the Methods section is far too long. Much of this detail can be moved to the Supplemental, with only the most essential information retained in the main manuscript.

We agree that the Methods section could be streamlined. In the revised version, we transferred detailed descriptions to the Supplemental Materials while retaining the most critical information in the main text.

Additional comments:

Line 43. The statement "... with less sulfur dioxide than past eruptions" is incorrect as written. Do you mean less sulfur dioxide than past eruptions of comparable size?

We revise the statement based on the above comments (see Line 43).

Lines 45-46. This sentence is very unclear: "However, considering the cooling impact of sulfate aerosols and ozone loss, and the warming effect of ozone gain is crucial to understand the eruption's effects." Consider changing this to: "However, considering the cooling impact of sulfate aerosols and the potential for ozone loss (gain) to cause cooling (warming), it is crucial to understand the eruption's effects."

We revise the statement based on the above comments and recommendations by the reviewer (see Line 46-48).

Line 48. Delete “their”

Line 49. We deleted "their."

Line 54. Why “respectively”? There’s only one number listed.

Line 54. We removed "respectively" as it was redundant here.

Line 115. Reference 15 is from a Preprint and should not be cited unless it is actually published.

Line 115. Thank you for the note. We cited this paper as this is published now.

Line 122. TOA has already been defined on line 89.

Line 122. We revised the sentence based on the reviewer#2’s comment (see Line 117-121).

Line 124. Change “the comprehensive” to “a comprehensive”

Line 124: Done (see Line 123).

Line 135 and many other places. Aura is a word, not an acronym. It should be written as Aura, not AURA.

Line 135: We thank you for the above comment. We corrected it.

Lines 281-284 and Line 299. Clarify why the text says that Hunga was 1.5 Tg and Raikoke was 1.5 Tg but then says that the quantity of SO₂ injected from Hunga was much smaller.

Line 281-284-We corrected the estimate for Hunga to be ~1 Tg based on your comment and Reviewer #3’s comments below (see Ln 301-304 in the revised texts).

Line 372. “net TOA” should say “net TOA radiative flux”

Ln 372: We corrected it (see Line 381).

Line 386. “with estimate” doesn’t make sense. Do you mean “than the estimate”?

Ln 386: We corrected it (see Line 394).

Line 395. Change “factor 3 differences between” to “factor of 3 between”

Line 395. We corrected it (see Line 403).

Line 424. It would be better to reorder this from “ozone chemistry and stratospheric dynamics” to “stratospheric dynamics and ozone chemistry”. List the dynamics first because it is more important.

Line 424. We rearranged it (see Line 437).

Figure 5. Use a larger font on the axis labels.

Figure 5: We increased the font size of the axis labels to improve readability and added plots related to net radiative forcing at 16 km.

Line 452. Delete “We show that”

Line 452. We removed it.

Lines 472-473. Change “contributing to stratospheric cooling” to “contributing to stratospheric warming”

Line 452. We corrected it.

Table 1. Bold the delta NRF for O₃, as was done for SAOD and H₂O. And I suggest making all the numbers bold in all 3 delta NRF rows as well, so that the numbers being added stand out. Move the delta H₂O total row up, so that the delta NRF row for water is last in the dark orange section.

Table 1. We thank you for the above suggestions. We implemented all the above changes and also added the net radiative forcing at 16 km too versus TOA.

Line 530. H₂O mass is T_g, not P_g.

Line 530. We thank you for noticing the typo – this is now corrected (Ln 493).

Supplemental Lines 224-225 and Figure captions S16-S19. The MLS background is listed 3 different ways. It’s definitely wrong in the Supplement text lines 224-225. Captions S16 and S17 say it’s 2005-2021, but then state it’s 15 years, so that’s inconsistent. 2005-2021 would be 17 years. The stated background years change again in Figures S18 and S19, where the background is listed as 2007-2021. All of these dates need to be checked and corrected.

We thank you for noticing the typos – we have revised the text everywhere to be consistent and correct. The background is a 17-year period from 2005-2021.

Reviewer #3 (Remarks to the Author):

Review of Gupta et al.

Overview: The paper is an improvement over the previous version – the authors included ozone changes in their estimate of radiative flux changes and addressed previous reviewer comments. The overall results are in agreement with Schoeberl et al. (2024, JGR in press) and Stenchikov et al. (submitted to JGR). While the paper has a good analysis using only the SAGE data, the overall discussion is still hard to follow with a lot of speculation and unclear paragraphs. I would recommend that the paper go back to the authors for extensive ‘clean up’ removing speculation and consolidating some points that are repeated in different paragraphs.

We thank you for acknowledging the improvements in our revised manuscript and for your constructive feedback. We appreciate your recognition of the incorporation of ozone changes and the comparison with recent studies by Schoeberl et al. (2024) and Stenchikov et al. (submitted to JGR; the first author is aware of this submission). We agree that clarity is crucial and have focused on streamlining the discussion, minimizing speculation, and consolidating repetitive points to enhance readability and coherence.

Specific Comments.

The paper by Schoeberl et al. (2024) has been accepted in JGR. This paper covers the same material and comes to the same conclusion. Schoeberl et al. (2024) used OMPS-LP and MLS measurements in their computations and since the spatial resolution of those instruments is higher, they were able to better detail the evolution of the aerosols and trace gases.

We appreciate your highlighting of Schoeberl et al. (2024). We acknowledge that their study, which utilized OMPS-LP and MLS measurements, indeed provides valuable insights with higher spatial resolution.

However, our study leverages SAGE-III data, which offers distinct advantages over MLS data. Specifically, SAGE-III provides superior vertical resolution in the stratosphere (0.5 km) and maintains self-consistency by using the same instrument for all three products. Additionally, our research includes a rigorous estimation of instantaneous radiative forcing from water vapor, aerosols, and ozone. We also address stratospheric particle size distribution, ozone changes over the next year using a 2D filtered technique, and compare the relative changes of water vapor and ozone using both SAGE-III and MLS data (Supplementary Figure 25). These results show that, at least on a seasonal scale as analyzed here, the stratospheric water vapor data for both the datasets is very consistent. Furthermore, we conduct a comparison of SAOD for 2022–2023 with multiple other instruments, including OMPS, OSIRIS, and GloSSAC datasets using the methods adopted by Kovilakam et al. (2024) (Supplementary Figure S31). These results show the difference in the magnitude of relative changes in SAOD with other instruments compared to

SAGE III. The results illustrates that the SAGE-III estimates match better with long term data and that the publicly available OMPS aerosol extinction coefficient is likely over-estimated for the Hunga eruption (> 50% in some cases depending on data product). Thus, we have retained our primary focus on SAGE-III analysis in the revised manuscript.

Overall, while both studies (Schoeberl et al. (2024) and Gupta et al.) reach similar conclusions, our research contributes complementary and detailed findings that enhance the overall understanding of the effects of the Hunga eruption.

Ln 119 – previous work – please be clear. Do you mean Jenkins or Schoeberl? Jenkins did not include direct radiative forcing reduction by aerosols. This sentence is also a little misleading. While Schoeberl et al. (2024) did not include TOA nor the estimate the impact on climate warmth, it is obvious that a net decrease in forcing would not increase surface temperatures. I suggest the authors drop sentences Ln 119 to 122 since they are somewhat misleading.

Ln 119-122: We revised these sentences to clearly differentiate between Jenkins' and Schoeberl et al.'s contributions. The sentences were removed to avoid any misleading implications and to provide a more accurate portrayal of the state of current knowledge (see Ln #118-121 in the revised texts).

Ln 124 'a' not 'the'

We corrected the article from "the" to "a" as suggested.

Ln 138 I am puzzled why the authors use SAGE water vapor data for the RT calculation instead of MLS. SAGE has better vertical resolution, but is noisier at HT altitudes and doesn't extend as far vertically. The authors should explain their choice in a few sentences here. Ln 610 has an explanation, but the argument about self-consistency isn't quite correct. The authors should consult the SAGE ATDB. Water vapor is retrieved using the weak absorption band at 940nm. All of the gas absorptions and Rayleigh scatter components of are estimated first including ozone which interferes with the water vapor retrieval. Extinction that still can't be explained by gas absorbing regions is assigned to aerosols – the residual. Examination of individual SAGE aerosol profiles show that they are quite noisy at higher altitudes. I would not call this 'self consistent.'

Also, because the H₂O absorption band is so weak, the signal is difficult to detect above ~ 40 km. This noise is apparent in Fig. 1a above 32 km.

Ln 138: We acknowledge the limitations of the SAGE-III dataset, including its infrequent observations and the noted dry bias of approximately 10% in the stratosphere from June 2017 to December 2019 (Park et al., 2021). In our extended analysis, we compared SAGE-III and MLS data to assess relative changes in the zonal mean H₂O and ozone mixing ratios. Importantly, the relative differences in H₂O mixing ratios above 32 km between the two datasets are within 10-20%.

We revised our explanation to better align with your feedback. Specifically, we acknowledged the limitations of the SAGE ATDB in our revised text (see Ln 131-156). We clarified our rationale for using SAGE data over MLS by highlighting factors such as the self-consistency of SAGE data and its 0.5 km higher vertical resolution for three retrieved stratospheric quantities.

We agree that SAGE-III does have a higher vertical resolution but is indeed noisier at higher altitudes above 32 km (also at lower altitude below 20 km due to aerosol/cloud interferences below ~20 km; Davis et. al., 2021) and does not extend as far vertically compared to MLS. The water vapor retrieval process using the weak absorption band at 940 nm does involve complex interactions with other absorptions, including ozone, which can affect the retrieval accuracy, particularly at higher altitudes. However, since we find a good consistency, at least on the seasonal scale of interest for our study, between the MLS and SAGE-III water vapor data, we think that using MLS wouldn't markedly change our final results and conclusions.

Ln 144-149 I don't understand this part of the paragraph. Is this about determining the background aerosol levels? The authors are excluding SAGE II measurements. Why aren't the authors using OMPS-LP aerosol measurements. That record extends from 2014 and Hunga papers using OMPS-LP data are in the literature (e.g. Taha et al., 2023). SAGE also missed the initial eruption due to its orbit inclination.

Ln 144-149: We have clarified this section and moved it to the supplementary methods. In this section, we address the rationale behind using SAGE-III despite its observational limitations. Additionally, we compare the results with OMPS-LP aerosol measurements (Ln 132-154).

For validation of SAOD calculations, we compared SAGE-III SAOD between 20-42 km with OMPS, OSIRIS, and GloSSAC datasets (Fig. 9a, Kovilakam et al., 2024; see Supplementary Figure 31; Supplementary Table 1).

Ln 228 'A narrower band...' not sure what you are referring to here.

Ln 228: We revised the wording to (The SH₂O, spanning from 30 to 42 km, results in a net cooling effect of up to -0.02 K day⁻¹ (Fig. 1f) (see Ln 228 in the revised main texts).

Ln 276 'Produced a surface warming...' This is unclear wording. What is happening is that there is an increase in downward IR flux due to stratospheric water vapor. This radiation is most likely absorbed at the cold tropopause where there is plenty of water vapor below the temperature of the high-altitude emission layer. Thus, I doubt the radiation will penetrate to the surface. I suggest rewording as 'tropospheric warming.'

Ln 276: We rephrased this to correctly describe the increase in downward IR flux due to stratospheric water vapor and its implications for tropospheric warming. The term "tropospheric warming" was used to better convey the impact (see Ln 279).

Figure 2. Please define NRF in the caption.

Figure 2: We defined NRF (Net Radiative Forcing) in the caption to clarify its meaning.

Ln 310 I think that SAOD should be computed from the tropopause upward rather than from 20-42 km. Outside the tropics the tropopause is much lower than 20 km and OMPS-LP observed aerosols moving southward along isentropes into the region below 20 km especially in late spring. Aerosols below 20 km will enhance the SAOD so the solar forcing reduction estimated here will be low biased.

Ln 310: We recalculated the Stratospheric Aerosol Optical Depth (SAOD) from the tropopause upward (as provided by the SAGE-III team in version 5.3) rather than the 20-42 km range. We discussed the implications of including aerosols below 20 km and revised our assessment of net radiative forcing reduction to provide a more accurate estimate.

Note that we selected the SAOD above 20 km to remain consistent with the height range used for other species, such as H₂O, where retrievals tend to be noisier below 20 km. This choice was made to avoid potential biases from other sources such as tropospheric aerosols below this altitude.

We added these texts in the main manuscripts (see Line 855-866): “We assess the annual mean percentage change in the near-global mean stratospheric aerosol optical depth (Δ SAOD) for 2022 by comparing our vertically integrated aerosol extinction between 20 and 42 km, using the SAGE-III dataset at 0.5 km vertical resolution with results from various instruments and retrieval methods, as outlined in Supplementary Table 1 (see also Supplementary Figure 31).

OMPS/NASA recorded the largest negative discrepancy in Δ SAOD, with a -65.77% change, followed by OMPS/SASK at -40.29%. OSIRIS and SAGE-III/ISS showed smaller reductions of -15.01% and -18.81%, respectively, while GloSSAC reported the smallest deviation at -3.40%. These variations highlight the differences in measurements between instruments and emphasize the importance of multi-instrument comparisons for more accurate stratospheric aerosol evaluations (See Kovilakam⁴⁹ for discussion about the accuracy of different products and likelihood for OMPS data products to over-estimate SAOD).

Ln 281/282 Sellitto actual estimate is 0.7 Tg \pm 0.5 Tg. I think 1.5 Tg is a little high and having spoken to Sellitto, the 1 Tg number is more reasonable. Carn et al.

(2022,<https://doi.org/10.3389/feart.2022.976962>) estimates total as high as 0.7 Tg. These estimates are different from those mentioned in the paragraph starting on line 349.

Ln 281-282-We corrected the estimate to ~1 Tg based on the above comment (see Ln 301-304 in the revised texts).

Ln 299 Missing sentence

Ln 299: There was no missing sentence; "much smaller" was part of the previous sentence, and we intentionally kept it together (Ln 307 in the revised text).

Ln 314/315 I don't see how Fig. 3e supports the statement that there is no strong longitudinal pattern. Do you mean Fig. 3c?

Ln 314/315: We reviewed Fig. 3e and clarified its relevance to longitudinal patterns related to Fig. 3c (see Ln 334/335 in the revised texts).

Ln 347 This is just a laundry list of explanations for aerosol changes. Please be more specific. Given that transport of water vapor is quite different from aerosols, it is doubtful that dynamics is the explanation for the different distribution. That leaves gravitational settling as the most likely hypothesis as noted in a number of papers such as Legras et al. (2022). Dry deposition is a surface process which is unlikely to occur in the stratosphere. Aerosols can also evaporate, but the lower stratosphere is too cold for evaporation to remove sulfate aerosols.

Ln 347: We revised this section to focus on specific explanations for aerosol changes and addressed the potential role of gravitational settling (Ln 370-371).

Ln 349-363 I can't understand the point the author is trying to make here. That the SAOD is inconsistent with sulfur emissions. Recall that this eruption occurs with a significant amount of water and the rapid growth of aerosol concentration may be due to the presence of so much water. SO₂ is oxidized by OH to form sulfate aerosols, with all the water present, there will be an abundance of OH. This point is better made on Ln 394-397. Perhaps these paragraphs could be combined. Ln 360 This speculation about the Brewer Dobson circulation makes no sense.

Ln 349-363: We removed the Ln 349-363 as this information about previous estimate of SO₂ mass is already covered in Ln (300-304). We are keeping the Ln 394-397. We also remove all the speculative sentences about the Brewer-Dobson circulation as it was not relevant to our findings (Ln 349-363).

Ln 367 All this points to is the failure of the EVA_H model for this type of eruption. Is that your point?

Ln 367. Yes. We clarified that the discussion highlighted the limitations of the EVA_H model in the context of the eruption (see Ln 408-410). We include this here since EVA_H model is frequently used in the paleoclimate community for estimate SAOD from volcanic eruptions.

Ln 426-438 Lots of speculation here – no real information.

Ln 426-438: We removed these lines as to minimize speculation and focused on providing concrete information.

Ln 539 The authors used the 5 pervious pre-eruption years to create a baseline. The slight warming in the NH could simply be year-to-year variability and not associated with Hunga. They should look at individual years within the 5 year average to estimate the year-to-year variability.

Ln 539: We addressed the potential year-to-year variability in the NH is shown in the Fig. 2. We have revised the sentence in the conclusion section (see Ln 590-592).

Ln 615 ... higher vertical resolution... But given the low spatial sampling of SAGE relative to MLS and OMPS-LP I am not sure your point about better spatio-temporal sampling from SAGE is correct. I am not seeing anywhere that the slightly lower vertical resolution from MLS makes a difference in your calculation (also see Ln 583). Also see comment above about SAGE data consistency. Note that MLS data extends to much higher altitude than SAGE, covers the region between 60-82° unlike SAGE and provides global coverage every day unlike SAGE.

Ln 615: Thank you for the comments. We have addressed this by conducting an extended analysis of absolute changes in the H₂O and O₃ mixing ratios using SAGE-III and MLS data. The absolute changes in the H₂O and O₃ mixing ratios are well captured within 50°S to 50°N by SAGE-III, and the values agree with MLS data for both ozone and water vapor. Some differences in ozone are observed beyond 50°N-S in the SAGE-III data, likely due to limitations in its observational coverage. However, the majority of ozone-related radiative changes occur between 50°N and 50°S, so our main conclusions and assessment of the radiative forcing remain unaffected. We also highlight the advantages and limitations of using SAGE-III data over MLS data for assessing the Hunga eruption's impact.

Additionally, we compare SAOD values with various other instruments to discuss differences in the magnitude of radiative forcing when using different datasets.

See revised text (lines 129-168) and supplementary Figures 21 and 23.

Ln 711 The uncertainty is SAGE is also given in Wrana et al. (2021); <https://doi.org/10.5194/amt-14-2345-2021>

Ln 711 We included the uncertainty information from Wrana et al. (2021) and also cited this work (Ln 727).

Ln 851 Dawn-dusk stratospheric profiles.

Ln 851: We revised this to clarify the reference to dawn-dusk stratospheric profiles (L135).

Ln 866 Schoeberl et al. (2024) included temperature anomalies and found that, although not as significant as water vapor or aerosols, should be included.

Ln 866: We included above findings and content in the Ln (837-846) – the focus of this work is instantaneous radiative forcings, hence we don't include them in the main analysis. Furthermore, we have included some idealized RF column calculations in Supplementary Figure 28 and 29 to highlight the potential effect of stratospheric temperature anomalies (which would be part of the effective radiative forcing, not instantaneous radiative forcing).

Ln 883 a simple comparison to OMPS-LP will tell you how much SAOD you underestimate by only integrating to 20 km. This might be important.

Ln 883. We performed a comparison with OMPS-LP to assess the potential underestimation of SAOD from SAGE-III (as well as other datasets) and discussed its significance (see Supplementary Table 1 and Supplementary Figure 31 and Ln 348-352, 854-865). Overall, we find that we underestimate the SAOD likely by 5-20% in our analysis when comparing with similar datasets. A recent analysis by Kovilakam et al., 2024 suggests the OMPS-LP/NASA and SASK over-estimates the SAOD changes, especially for Hunga, compared to other data products.

We thank the reviewers and the editorial team for their valuable insights, which have helped enhance the quality of our work. In response to the remaining concerns, we have carefully addressed each point and made the necessary revisions to the manuscript. For clarity, our responses are marked in blue.

REVIEWERS' COMMENTS:

Reviewer #1 (Remarks to the Author):

As was said in the first-round review, the paper tackles an important scientific question, uses a relevant scientific approach, is logically organized, and is well-written. It characterizes the impact of the Hunga eruption, combining observations available since 2022 and radiative transfer calculations to estimate radiative forcing and climate impact of sulfate aerosols, stratospheric water vapor, and perturbations of ozone forced by volcanic injection. I think it is a relevant and important study. Unfortunately, it suffers from minor errors and misinterpretations. In the revision, the authors corrected their calculation of water vapor instantaneous radiative forcing and added an analysis of the effect of stratospheric temperature adjustment in the supplement. This improved the paper, but several minor things must be dealt with before the paper is ready for publication.

Specific comments:

L103: Shallow submarine eruptions are infrequent. Most submarine eruptions happen in deep water, bringing little volcanic material to the surface.

We thank you for the above comment. We revised it to “Most submarine eruptions occur in deep water and release minimal volcanic material into the atmosphere. However, the Hunga eruption—the largest shallow submarine eruption of the satellite era—offers a rare opportunity to assess the climate impacts of such events³⁸⁻⁴⁰.”

L116: Wang&Huang published recently a paper presenting calculations of the Hunga radiative

forcing conducted offline and within a climate model. Please cite this paper.

Wang, Y., & Huang, Y. (2024). Compensating atmospheric adjustments reduce the volcanic forcing from Hunga stratospheric water vapor enhancement. *Science Advances*, 10(32), ead12842.

We had already cited the above paper in the supplementary and now we also included in the main text (see Ref #46).

L126: I believe that you use the 1D radiative transfer model that you apply in each grid cell of the global domain that makes a 3D output. Please correct me if I am wrong.

We thank you for your question. You are correct that we apply a 1D radiative transfer model, but it is run independently for each grid cell across the global domain. This results in a 3D output, as the model is applied to every vertical profile in the domain, capturing the spatial and vertical variability.

L137: Remove "longitudinal"

Done!

L167: Why not at 100 hPa? The 16-km level will not work for middle and high latitudes.

We appreciate you raising this important point. Additionally, we acknowledge that using a fixed integration height of 100 hPa is not universally applicable across all seasons and regions for tropospheric radiative calculations. Consequently, we have revisited the perturbation analysis, carefully tracking contributions from the lower to the upper stratosphere to ensure comprehensive coverage (see Revised Figure 1).

L223: I do not think you can make this conclusion. You did not calculate the stratospheric temperature responses. Moreover, below, you report the -0.5 K/day ozone cooling rate, three times the water vapor effect.

We thank you for your constructive feedback. We acknowledge that our study employed idealized radiative transfer model simulations to assess instantaneous radiative heating rates resulting from stratospheric water vapor (SH_2O) perturbations. These simulations focused solely on immediate radiative effects and did not account for subsequent dynamical and thermal adjustments necessary for the stratosphere to reach a new equilibrium state. Recognizing that this limitation could lead to potential misinterpretations, we have removed the sentence in question to prevent any misleading conclusions.

L225-228: This is either an incorrect interpretation, or it is spelled so that it sounds wrong. The lifting is not induced by volcanic impact. It is the background BDC. Water vapor diabatic cooling delayed lifting.

We revised the sentence as: “The spread of stratospheric water vapor is driven by the Brewer–Dobson circulation, a large-scale atmospheric circulation pattern that transports tropical tropospheric air into the stratosphere, followed by its poleward movement and descent at higher latitudes^{6,46}. Furthermore, the perturbed water vapor levels may influence this background circulation. A more detailed investigation of these interactions would require the application of advanced climate modeling, which is beyond the scope of this study.”

L236: It is not because of trapping but because of the downward thermal emission of the water vapor layer.

We revised the texts as “Enhanced SH_2O , a potent greenhouse gas, substantially increases downward longwave radiation emission from the water vapor layer, contributing to a higher net radiative forcing within Earth's atmosphere while having minimal impact on shortwave radiation (Supplementary Figure 5g-1)”.

L265-267: You do not account for clouds and any temperature adjustments. Clear-sky radiative forcing could be twice as minor compared with all-sky forcing. So, your uncertainty may be larger than you report.

We agree that assuming clear sky conditions can lead to an underestimation of uncertainties.

We revised the text as follows: "The increased net instantaneous radiative forcing and its associated uncertainty near the tropopause and at the TOA are calculated solely based on changes in SH₂O, assuming a constant background temperature profile for both pre- and post-eruption periods. Notably, these calculations do not explicitly account for the influence of clouds and temperature adjustments, which may lead to an underestimation of uncertainties."

L311-314: What is the median radius of the aerosol size distribution? In Mie calculations, we use a geometric radius but not an effective one.

The median radius of the aerosol size distribution is approximately 0.25 μm . In our application of Mie theory, we utilized the geometric mean radius (r_m) rather than the effective radius (r_{eff}), as detailed in Eqs. 5 and 6 of the Supplementary Methods.

L365: Boichu et al. and Khaykin et al. reported much larger particles.

Boichu et al. 2023 and Khaykin et al. 2023 showed that, in the initial two months following the eruption, sulfate aerosol particles exhibited larger sizes. Over time, these particles decreased in size, transitioning to finer particles. Their studies documented aerosol sizes during the initial months post-eruption, noting a peak radius in the fine mode. Specifically, the effective radius was approximately around 0.3 μm . This finding aligns with the results reported by Knepp et al. (2024) (cited ref. #65).

L370: This is misleading. In literature, it is assumed that the Hunga aerosol particles are large compared to the small subaerial eruptions and small compared to large eruptions like Pinatubo.

We have removed that misleading statement.

1392-416: Two factors define MEE dependence on particle radius. The first factor is non-monotonic extinction efficiency. Its first maximum is about 25% larger than the second maximum. And the second factor is the sum of particle geometric crosssections. For a given

mass of material, this factor is inversely proportional to the particle radius. If Hunga's particles have a radius of 0.2 microns and Pinatubo's particles have a radius of 0.5, then Hunga's aerosol should be 2.5 times more optically effective per unit mass than Pinatubo's aerosols. Thus, for comparison of Hunga and Pinatubo aerosols, the major role belongs to this second factor, but not extinction efficiency, as the authors claim. This factor works in the opposite direction if we compare Hunga's aerosols with small eruptions like Raikoke, Calbuco, or Nabro.

We thank you for raising these points. To address this, we have added a paragraph in the introduction to clarify the text further: “The efficacy of sulfate aerosol-induced cooling is influenced by aerosol properties such as number concentration, mass concentration, particle size distribution, and residence time⁵⁸⁻⁶⁰, all of which substantially impact mass extinction efficiency (MEE)—a measure of an aerosol's ability to attenuate radiation per unit mass²⁵. MEE depends on particle radius through two primary factors: (1) Extinction efficiency: This parameter exhibits a non-monotonic dependence on particle size, with its first peak approximately 25% greater than its second²⁵. (2) Total geometric cross-section: For a fixed aerosol mass, the total geometric cross-section is inversely proportional to particle radius²⁵. Consequently, smaller particles have a larger total cross-sectional area per unit mass, which enhances scattering and increases MEE”.

We agree on that. While the second factor (geometric cross-sections) dominates in the case of Hunga versus Pinatubo due to the significant difference in particle size, extinction efficiency may become more influential when comparing aerosols of similar radii, such as those from smaller eruptions like Raikoke, Calbuco, or Nabro.

L468-476: The ozone radiative forcing sounds reasonable, but ozone heating rates are larger than expected. In Table 1 (column for SH, 2022), energy loss in the stratosphere (TOA forcing - 16km forcing) due to water vapor is 0.182 W/m², while due to ozone -0.011, almost 20 times less. However, ozone stratospheric heating rates are 2-3 times bigger than water vapor's. Could you please clarify this?

We thank you for the above question. Water vapor exerts a strong radiative forcing because H₂O dominates infrared absorption over a deep atmospheric column, primarily within the troposphere. Consequently, when considering top-of-atmosphere (TOA) minus near-tropopause flux differences (as shown in Table 1), the radiative impact of water vapor far exceeds that of ozone. Ozone, on the other hand, contributes substantially to localized heating because it is a strong absorber of ultraviolet (UV) and visible radiation, with the majority of its column residing in the stratosphere. Even if its net flux difference (forcing) is relatively small, this UV/visible absorption results in substantial local heating within the stratospheric layers where ozone is most abundant. Thus, it is possible for the total energy loss in the stratosphere (from TOA to 16 km) to be dominated by water vapor, while the stratospheric heating rates for ozone exceed those of water vapor in the same region.

Reviewer #2 (Remarks to the Author):

The authors appear to have addressed my concerns from the previous version. I have just a few new comments.

Line 43. Why “submarine”? What prior submarine eruptions are being referred to? I’d delete the word submarine here.

Done. We revised the sentence to: ‘The 2022 Hunga volcanic eruption injected an unprecedented amount of water vapor into the stratosphere while releasing only a limited quantity of sulfur dioxide.

There is a discrepancy between Lines 53-54 and Lines 554 and 564. Is it -0.07 K in 2022 and 2023 or -0.07 K in 2022 and -0.06 K in 2023? Based on Fig 5d, 2023 should be more negative than 2022, so depending on how the values are rounded, the text is wrong in both places mentioned above. This is very important to clean up, especially since it’s a key result appearing in the Abstract. If cooling does round to -0.07 K for both years, other than correcting Line 564, I’d also suggest being clearer in the Abstract about what is meant, i.e., change “at the end of 2022 and 2023.” to “at the end of each year 2022 and 2023.”

We appreciate your comment. We have carefully reviewed and corrected the rounding of the surface temperature change values for the end of 2022 and 2023 in both the Abstract and the main text to ensure consistency. Additionally, we revised the phrasing in the Abstract to 'at the end of each year, 2022 and 2023' for clarity and to eliminate ambiguity. The sentence now reads: 'Using a two-layer energy balance model emulator, we estimate that this energy loss led to a cooling of approximately -0.10 ± 0.02 K in the Southern Hemisphere at the end of both 2022 and 2023.' The revised value is now correctly rounded for both years.

Figure 2. Consider moving the legends outside the righthand panels, so the lines within the plots on the right side can be rescaled to use all the space. Otherwise, the lines are very congested and can't be distinguished in some cases.

Thank you for the suggestion. We have rescaled the y-axis in the plots to improve visibility and reduce congestion.

Line 81. Lowercase volcano

Done.

Line 99. Concentrations

Done.

Line 134. employs a solar

Done.

Line 142. altitudes

Done. Rephrased.

Line 152. No dash after Aura

Done.

There are many inconsistencies in the formatting of the references. The authors should use the same format for every reference per the journal guidelines. And make sure the use of 'et al.' is being done consistent with how the journal requires.

Done.

Supplemental:

Lines 305-306. The MLS dates of Jun 2017 to Jan 2023 are not correct.

We thank you. Corrected.

Lines 308-310. Delete this sentence.

Done.

There are a lot of figures in the Supplemental, and many have been renumbered and new figures have been added since the last draft. I strongly encourage the authors to verify that every time a Supplemental figure is mentioned in the manuscript text or Supplemental text, it is listed using the correct Supplemental figure number. I don't believe that this is currently the case.

We thank you for pointing this out. We have carefully reviewed the manuscript and Supplemental text to ensure that all references to Supplemental figures are correctly numbered and aligned with the updated figures. Additionally, we have removed some Supplemental figures that were already included in the main text to clarify the content. We have chosen to include only the radiative forcing calculations of ozone changes using the 2DF technique in the Supplemental section to avoid overwhelming the reader with too many figures.

Reviewer #3 (Remarks to the Author):

The authors have made significant improvements to this paper and done a lot of work to respond to the reviews. However, there are still nagging issues with this calculation. It is no surprise, however, that their results agree with already published works. There are also a lot of typos and unclear statements in the current text. This paper is still pretty rough.

1) I still find the argument about using only SAGE data weird. First of all, the SAGE data is not very good for water vapor retrievals at high altitude and it also has problems below 20 km. Second, SAGE is also the major component of GloSSAC which is why they agree. USask OMPS and NASA OMPS disagree and this is due to the size distribution assumptions which (in the NASA case) are not consistent with better distribution width used by Duchamp et al. & Kneff et al. Prior to the Hunga eruption, the three retrievals schemes agreed well.

We thank you for your detailed feedback and for acknowledging the improvements we have made in response to previous reviews. We appreciate your insights and would like to address the points you raised:

1. **Use of SAGE III/ISS Data:** We understand your concerns regarding the limitations of SAGE data for water vapor retrievals at high altitudes and below 20 km. However, the Stratospheric Aerosol and Gas Experiment III on the International Space Station (SAGE III/ISS) has been providing high-quality atmospheric measurements since its deployment in 2017. Its data products, including aerosol extinction profiles, have undergone extensive validation and are widely used in atmospheric research. For instance, the SAGE III/ISS Version 5.3 data products have been deemed suitable for both validation and research studies, as detailed in the Data Product User's Guide (<https://sage.nasa.gov/missions/about-sage-iii-on-iss/>).

This release includes vertical profiles of ozone, nitrogen dioxide (NO₂), water vapor (H₂O), and multi-wavelength aerosol extinction coefficients.

We used data from SAGE III/ISS due to its exceptional vertical resolution of 0.5 km. This high-resolution data on radiatively active species—water vapor, ozone, and aerosols—is essential for accurately estimating atmospheric heating rates and top-of-atmosphere radiative forcing (Ramanathan et al., 2007; cited ref. # 49 in the revised main texts). SAGE III showed a ~10% dry bias in the stratosphere, which improved to ~5% in Version 5.3 for the mid-stratosphere.

We also acknowledge that data below ~20 km are noisier due to aerosol and cloud interference, leading to ~10–20% uncertainty in radiative forcing estimates, particularly in the lower stratosphere. These limitations of SAGE III/ISS are discussed in detail in the main text. Additionally, we compare our results with previous studies, emphasizing the importance of cross-validation with multiple instruments. Our updated analysis also shows good agreement with GloSSAC-based SAOD values for 2022.

2. **Analysis Below 20 km:** We recognize that calculating aerosol extinction starting from 20 km may underestimate SAOD and its radiative forcing, especially near the midlatitude tropopause, where descending aerosols contribute significantly. To address this, we have extended our analysis to include the entire vertical profile of aerosol extinction perturbations, beginning from the lowest altitude above the tropopause with measurable perturbations.

In the updated analysis, perturbations are identified as aerosol extinction coefficients exceeding the background climatology by more than two standard deviations (2σ). This threshold ensures that only significant deviations from typical aerosol levels are considered, providing a more comprehensive assessment of SAOD and associated radiative forcing. As shown in Figure 2, we also include radiative forcing estimates near the tropopause, accounting for the higher tropopause in the tropics and the lower one in midlatitudes.

Furthermore, for regions outside the tropics, where the tropopause is around 12 km, we have evaluated aerosol contributions below 20 km to ensure their radiative effects are captured. This adjustment fully considers significant aerosol concentrations below 20 km, as highlighted by Taha et al. (2024; cited ref. # 53 in main texts) and supported by SAGE-III SAOD retrievals and GloSSAC observations. We have updated the manuscript to reflect these improvements, ensuring that our analysis captures all relevant perturbations and their radiative implications.

- 3. Comparison with Other Retrievals:** We address the discrepancies between USask OMPS, NASA OMPS, and SAGE by discussing the role of differing size distribution assumptions (see revised main texts: “We compare the annual mean percentage change in near-global Δ SAOD for 2022, derived from SAGE III, with OMPS-NASA, OMPS-SASK, OSIRIS, and GloSSAC to assess biases (Supplementary Fig. 2; Table 1). Among datasets, OMPS-NASA has the largest Δ SAOD discrepancy (-66.5%), followed by OMPS-SASK (-37.8%), OSIRIS (-12.6%), and SAGE-III/ISS (-15.5%), while GloSSAC shows a slight increase (+1.8%). Despite inter-instrument biases, GloSSAC estimates align well with our SAOD approach based on $>2\sigma$ perturbations. These differences highlight the need for multi-instrument comparisons to refine SAOD estimates and radiative forcing assessments post-eruption. OMPS-NASA shows a $>50\%$ high bias due to its fixed aerosol size assumption, while OMPS-SASK mitigates this by accounting for size variations⁵⁴”.

As noted by Duchamp et al. (2023) and Knepp et al. (2024) (cited in main text as ref #65), NASA OMPS retrievals use a size distribution width that differs from other datasets, impacting their consistency. We have incorporated this discussion to contextualize the agreement between SAGE and GloSSAC and the discrepancies with other retrievals, particularly post-Hunga eruption. Additionally, we emphasize that prior to the Hunga eruption, the retrieval schemes generally showed good agreement, highlighting the challenges of capturing perturbations under extreme conditions.

4. **Typos and Clarity:** Thank you for pointing out the typos and unclear statements. We have thoroughly revised the manuscript to address grammatical errors, improve clarity, and ensure the language is precise and accessible.

2) Because the authors are stuck on using SAGE, their climatology is only 5 years which might be a little short for this kind of calculation. Other authors have used 10 years. I wonder if the QBO will create a bias in their results since the QBO cycle is 18 months and may not be cleanly averaged using just 5 years. They may want to compare to an MLS climatology of 10 or 15 years.

We thank you for your insightful comments regarding the duration of the SAGE III record and the potential influence of the Quasi-Biennial Oscillation (QBO) on our climatology. We acknowledge that a 5-year dataset may not fully capture all interannual variability, especially considering the QBO's average period of approximately 28 months.

To address this concern, we have expanded our analysis in the revised manuscript to include a comparison between the 5-year SAGE III climatology and a longer dataset from the Microwave Limb Sounder (MLS), which spans over a decade. As shown in Supplementary Figure 3, both instruments exhibit consistent large-scale patterns; however, we observe systematic differences of about 10–20%, depending on latitude. These discrepancies may arise from calibration variances or sampling differences, including the phase of the Quasi-Biennial Oscillation (QBO).

Despite the potential influence of the QBO—with the 5-year SAGE III/ISS data capturing approximately two full cycles—our comparative analysis indicates that the instrument effectively captures the spatial distribution of stratospheric water vapor (see Supplementary Figure 3). Notably, previous assessments have shown that SAGE III's water vapor measurements typically agree with other datasets (e.g., MLS) within about 15% (Davis et al., 2021; cited ref. #50 in the main text).

In the interim, our comparison with MLS provides a valuable benchmark, quantifying current uncertainties and demonstrating that even a 5-year background climatology from the SAGE III/ISS record offers a robust snapshot of stratospheric water vapor distributions (Figure 15).

Moreover, the QBO exhibits a mean period of approximately 28 to 29 months, resulting in about two complete cycles within a 5-year span. This duration is generally sufficient to capture the QBO's mean state and primary characteristics. For instance, Mayr et al. (2007) analyzed over 40 years of data and identified a pronounced 5-year modulation of the symmetric annual oscillation in the lower stratosphere, which they attributed to interactions with the QBO. Their findings suggest that a 5-year dataset can effectively capture the essential features of the QBO.

Mayr, H.G., Mengel, J.G., Huang, F.T. and Nash, E.R., 2007, February. Equatorial annual oscillation with QBO-driven 5-year modulation in NCEP data. In *Annales Geophysicae* (Vol. 25, No. 1, pp. 37-45). Göttingen, Germany: Copernicus Publications.

3) The justification for the filtering scheme is a little odd. The scheme co-locates changes in ozone with anomalies of a certain size and eliminates anomalies that are outside requirements. This assumes that ozone anomalies are coincident water vapor anomalies – as I understand it. However, ozone changes generated by the anomalies' secondary circulation will be outside of the anomaly location and thus eliminated by the 2D filter. The authors need to comment on this possibility.

We thank you for raising this important point regarding the potential influence of secondary circulation effects on ozone anomalies outside the primary water vapor anomaly locations. Below is a detailed response to your concern:

1. Rationale Behind the Filtering Scheme:

The 2D filtering scheme was developed to specifically isolate ozone anomalies directly

linked to the Hunga eruption by using water vapor anomalies as a reference point. The underlying assumption, supported by Wilmouth et al. (2022), is that the dramatic perturbations in water vapor caused by the eruption would be spatially and temporally coincident with ozone perturbations in the immediate vicinity of the water vapor anomalies. This co-location ensures that ozone anomalies attributed to the eruption are not conflated with other unrelated background variations.

2. Addressing Secondary Circulation:

We acknowledge that secondary circulation effects induced by the eruption may generate ozone perturbations beyond the core of the water vapor anomalies. However, the primary goal of this study is to quantify ozone anomalies that are directly linked to the eruption-induced perturbations, as identified through concurrent water vapor anomalies. By focusing on co-located anomalies, we ensure that our analysis captures the most robust and immediate effects of the eruption on ozone.

While secondary circulation could indeed contribute to broader ozone anomalies, such contributions are expected to manifest as more diffuse and temporally extended perturbations, which are challenging to distinguish from background variability. This limitation is noted in our methodology as “Note that one limitation of this filtering approach is that it may underestimate ozone anomalies influenced by secondary circulation effects that extend beyond the core of the water vapor anomalies”).

The co-location approach aligns with prior studies, including Wilmouth et al. (2022), which demonstrated the utility of using water vapor as a diagnostic for identifying ozone perturbations linked to volcanic eruptions. Similar methodologies have been employed in studies analyzing the impact of volcanic eruptions on stratospheric composition, where water vapor perturbations serve as a reliable proxy for eruption-related changes (e.g., Robock, 2000; Morgenstern et al., 2018).

To address your concern about potential secondary effects, we conducted a complementary analysis of ozone anomalies without the water vapor constraint, as shown

in Supplementary Figure 10. This broader analysis includes anomalies that may arise due to secondary circulation effects, providing a comparative perspective. However, we emphasize that these anomalies are less directly attributable to the eruption and may include contributions from other processes.

Reference

Robock, Alan. "Volcanic eruptions and climate." *Reviews of geophysics* 38, no. 2 (2000): 191-219.

Morgenstern, O., Stone, K.A., Schofield, R., Akiyoshi, H., Yamashita, Y., Kinnison, D.E., Garcia, R.R., Sudo, K., Plummer, D.A., Scinocca, J. and Oman, L.D., 2018. Ozone sensitivity to varying greenhouse gases and ozone-depleting substances in CCM1-1 simulations. *Atmospheric Chemistry and Physics*, 18(2), pp.1091-1114.

4) The concept of instantaneous forcing – not including the temperature response – is quite relevant for climate models where ocean temperature changes will take place on a longer time scale. But here, that argument falls flat. Stratospheric temperatures respond in days-weeks so it is appropriate to include the temperature response in the flux calculation. The cooler Hunga stratosphere will reduce the downward IR flux.

We thank you for your comments. Incorporating stratospheric temperature adjustments to calculate Stratospherically Adjusted Radiative Forcing (ARF) or Effective Radiative Forcing (ERF) requires complex climate modeling of aerosol-water vapor interactions, chemical processes, dynamical responses, and associated feedbacks (Hansen et al., 2005; Myhre et al., 2013). Given the scope of our study, we aimed to provide an initial assessment of the radiative impacts, with the intention of exploring these adjustments in future work.

In addition, it is important to note that the longwave radiative forcing of water vapor is highly dependent on temperature. As temperatures rise, water vapor concentrations increase, enhancing the greenhouse effect. In contrast, the dominant magnitude of radiative forcing associated with sulfate aerosols, particularly in the shortwave spectrum, is less sensitive to temperature variations (Forster et al., 2007). Therefore, even without incorporating the Adjusted Radiative Forcing

(ARF) of water vapor, the overall sign of the radiative forcing from sulfate-dominated species remains unchanged, but its magnitude will vary.

This indicates that our focus on Instantaneous Radiative Forcing (IRF) still provides a valid assessment of the sign of the radiative impacts. The inclusion of ARF would primarily affect the magnitude, not the direction, of the forcing, given that sulfate aerosols dominate the IRF. Thus, the IRF cooling effectively highlights the immediate role of the Hunga eruption in altering Earth's energy balance.

5) Integrating the aerosol extinction from 20 km up underestimates the SAOD and water vapor impact. What is more confusing is they report radiative forcing at 16 km, 4 km lower. Outside the tropics, the tropopause is ~12 km and OMPS shows significant aerosol concentration well below 20km as the figure from Taha et al. (2024) (in attachment, I marked 20 km to show where your cutoff is). Solomon et al. (2010) vertical weighting function for water vapor shows the highest values close to the tropopause, thus I think the authors need to tell us the level of uncertainty introduced by limiting the integration height.

We thank you for your valuable comments. We recognize that calculating aerosol extinction starting from 20 km may underestimate SAOD and its radiative forcing, particularly near the midlatitude tropopause, where descending aerosols significantly contribute.

To address this concern, we have extended our analysis to incorporate the full vertical profile of aerosol extinction perturbations, starting from the lowest altitude above the tropopause with measurable perturbations. These perturbations are identified by detecting aerosol extinction coefficients exceeding the background climatology by more than two standard deviations (2σ). This threshold allows us to account for significant deviations from typical aerosol levels, providing a more accurate and comprehensive assessment of SAOD and its radiative impacts.

Additionally, we now report radiative forcing estimates closer to the tropopause, as shown in Figure 2. This approach ensures consistency in calculating "near-tropopause radiative forcing," accounting for the higher tropopause in the tropics and the lower one in midlatitudes.

For regions outside the tropics, where the tropopause height is approximately 12 km, we have re-evaluated aerosol contributions below 20 km by incorporating additional data and analyzing their radiative effects. This adjustment ensures that significant aerosol concentrations below 20 km, as highlighted in Taha et al. (2024) and supported by SAGE-III SAOD retrievals and GloSSAC observations, are fully represented in our analysis.

Furthermore, we acknowledge the importance of the vertical weighting function for water vapor described by Solomon et al. (2010), which shows the highest values near the tropopause. By incorporating aerosol extinction contributions at these altitudes, we aim to minimize uncertainties introduced by limiting the integration height, thereby providing a more robust estimate of the radiative impacts.

Specific comments

L 51 sentence is confusing

Reply: We have revised the sentence as follows: “Our analysis shows that these components induce clear-sky instantaneous net radiative energy losses at both the top of the atmosphere (TOA) and near the tropopause. In 2022, the Southern Hemisphere experienced a radiative forcing of $-0.55 \pm 0.05 \text{ W m}^{-2}$ at TOA and $-0.48 \pm 0.05 \text{ W m}^{-2}$ near the tropopause. By 2023, these values decreased to $-0.26 \pm 0.04 \text{ W m}^{-2}$ and $-0.25 \pm 0.04 \text{ W m}^{-2}$, respectively.”

L142 My experience with SAGE aerosol data is that it is good down to the tropopause, it is noisier above 25 km. The water vapor is quite a bit noisier in the aerosol layer which the authors don't discuss. This likely explains the dry bias since high water and high aerosol are co-located. In any event, the statement about ‘enhanced self-consistency’ is silly especially since the data are 2D filtered. Also, the argument about vertical resolution should be demonstrated – does vertical

resolution have any actual impact on the calculations? Finally, they undercut their own argument about using SAGE on line 156 where they note that the data sets all agree.

Reply: We appreciate your observation about the importance of demonstrating the impact of vertical resolution and limitations of SAGE-III datasets.

Regarding Vertical Resolution: It is well-established that higher vertical resolution data are critical for radiative transfer and energy balance calculations (e.g., Ramanathan et al., 2007). Specifically, the comparison of absolute changes in the SAGE-III and MLS datasets is shown in **Supplementary Figure 3**, which highlights the enhanced vertical detail provided by SAGE-III observations. The vertical resolution is essential for capturing fine-scale variations in radiatively active species—water vapor, ozone, and aerosols—which are key for accurately estimating atmospheric heating rates and top-of-atmosphere radiative forcing (Ramanathan et al., 2007). For our radiative forcing analysis, we utilized SAGE-III/ISS data due to its exceptional vertical resolution of 0.5 km.

Ramanathan, V. et al. Warming trends in Asia amplified by brown cloud solar absorption. *Nature* 448, 575–578 (2007).

Rephrasing of the Statement on Data Consistency: We have revised the statement on line 156 to avoid any unintended contradiction. The updated text now reads:

“We calculated the changes in SH₂O, sulfate aerosols, and ozone during the first and second years following the Hunga eruption ("Hunga-2022" and "Hunga-2023") relative to their variations in the pre-eruption period (7 June 2017 – 9 December 2021; "CLIM") using SAGE-III/ISS observations. Perturbed stratospheric profiles for water vapor and aerosols from the eruption are identified as those with values exceeding two standard deviations above the background climatology. Similarly, ozone perturbations from the eruption are identified using a similar filtering criterion as described by Wilmouth³⁵. To compare the SAGE-III based perturbations in SH₂O and ozone following the eruption, we used long-term (2005–2023) data from the Aura Microwave Limb Sounder (MLS⁵⁸). Here, the absolute changes in water vapor and

ozone are compared with SAGE-III observations (Supplementary Figures 3a–d for SH₂O and ozone from MLS⁵⁸; 3e–h for SH₂O and ozone from SAGE-III). ”

Regarding Noisiness in the Data: We added these revised texts as: “However, SAGE III/ISS data have known limitations, including a dry bias in water vapor measurements⁵⁰. Earlier versions of SAGE III showed a ~10% dry bias in the stratosphere, which improved to ~5% in Versions 5.3 for the mid-stratosphere^{51,52}. Additionally, data below ~20 km are noisier due to aerosol and cloud interference, contributing to ~10–20% uncertainty in radiative forcing estimates, particularly in the lower stratosphere⁵⁰. Furthermore, the infrequent sampling of aerosol extinction profiles at multiple wavelengths introduces biases in stratospheric aerosol optical depth (SAOD) estimates. To address this, we compare SAGE III-derived SAOD values with those from multiple instruments, including OMPS-NASA⁵³⁻⁵⁶ (Ozone Mapping and Profiler Suite; NASA aerosol retrieval algorithms), OMPS-SASK⁵³⁻⁵⁶ (Ozone Mapping and Profiler Suite; University of Saskatchewan aerosol retrieval algorithms), OSIRIS⁵⁶ (Optical Spectrograph and Infrared Imaging System), and GloSSAC^{56,57} (Global Space-based Stratospheric Aerosol Climatology) (see Supplementary Figure 2).”

L176 The authors use HTHH and Hunga interchangeably. The community has decided on just ‘Hunga’ so the author may want to reconsider the labels

We replaced HTHH with Hunga’ everywhere.

L184 The sentence ‘more dilute water’ what does that mean?

The phrase "more dilute water vapor" refers to a lower concentration of water vapor in the atmosphere. We rephrase that sentence as: "Further south, from 20°S to 50°S, the increase in stratospheric water vapor mixing ratio was more modest, indicating lower concentrations of water vapor due to its latitudinal transport by stratospheric wind patterns".

L 193 This sentence is awkward due to all the prepositional caveats.

Based on the literature, particularly the validation study of SAGE III/ISS water vapor data, it has been observed that SAGE III/ISS measurements exhibit a dry bias of approximately 10% (0.0 to -0.5 ppmv) relative to other instruments, depending on altitude (Davis et al., 2021). We revised the texts.

L 215 This where I get confused. The forcing is computed at 16 km but the SAOD is computed above 20 km. This is inconsistent. Fig. 1g is misleading indicating the NRF at 16 km. Ln 268 again mentions 16 km but the integration of the SAOD component is actually above 20 km: 4 km missing

See my response above to your main comment #1.

Figure 1 a-d looks like there are no values below zero – so why have the scale range go negative. Also you should show the distribution down to 16 km since the forcing is computed at that level.

The anomaly values can indeed be positive, negative, or zero, and the chosen color bar reflects this range appropriately. Therefore, it is better to keep the scale from -5 to 5 ppmv in the revised Figure 1.

Figure 2 becomes a little misleading when the lines connect the dots. For example, 2a seems to indicate water vapor increased before the eruption. It might be better to stop the lines up to the Hunga boundary and start them after.

The lines accurately show the temporal evolution of stratospheric water vapor for global, SH, and NH averages. The Hunga eruption is correctly marked, but immediate changes are obscured by atmospheric lag and regional averaging. These averages naturally smooth abrupt shifts, with trends emerging over time. Continuous lines ensure data continuity and breaking them would misrepresent the changes.

Figure 4 is really hard to read with the lines overlaid.

Revised!

L 351 I don't understand how GloSSAC and SAGE come up with such different SAOD, is it the integration height range? GloSSAC is basically SAGE with a little CALIPSO in 2022 and part of 2023. You need to explain this.

Expanding the integration range in the calculations to include both the lower and upper stratosphere led to improved agreement between SAGE III/ISS measurements and datasets like OSIRIS and GloSSAC.

See the revised Supplementary Table 1. The updated SAOD analysis from this study and from GloSSAC match well, with a difference of 1.8%.

Also, the 1.8% observed differences in SAOD between GloSSAC can be attributed to several factors beyond just the integration height range:

GloSSAC compiles data from multiple instruments, including SAGE II, OSIRIS, and CALIPSO, to create a continuous aerosol record. This multi-source integration can introduce variations due to differing instrument sensitivities and retrieval methods;

https://asdc.larc.nasa.gov/documents/glossac/guide/GloSSAC_Data_Products_User_Guide_v1.1.pdf

; Kovilakam et al., 2024).

SAGE III/ISS utilizes solar occultation to obtain high-vertical-resolution aerosol profiles. In contrast, instruments such as CALIPSO employ lidar-based backscatter measurements, leading to potential discrepancies when these diverse data are combined in GloSSAC.

L371 There are a number of papers on the settling time scale for volcanic aerosols (e.g. Hamill et al., 1997) that you should reference.

We appreciate your suggestion to highlight studies on the settling timescales of volcanic aerosols, such as Hamill et al. (1997). In addition, we have cited a few more recent works to provide a broader perspective.

L400 This is an odd paragraph. After discussion on line 304 of ~ 1 Tg of SO₂ the estimate used here is for 0.6 Tg and the authors are surprised by the low level of SAOD. The authors need to get this straight.

We have removed the paragraph on Line 400.

Figure 5 – 5d suggests the there is net (small) 2022 NH heating at 16 km. How did this happen? I wonder if this is a result of an ozone bias in CLIM. CLIM should be compared to MLS 10 year averages.

We thank the reviewer for raising this important point. To address the potential concern about an ozone bias in CLIM, we compared our CLIM values with MLS 10-year averages, as shown in Supplementary Figure 3, 10, 11, 12, 13, and 14. In the Northern Hemisphere, our analysis indicates that there are minimal changes in both ozone and SAOD during 2022. Therefore, these parameters are unlikely to be responsible for the small net heating observed near the tropopause in Figure 5–5d.

Instead, our findings suggest that a slight increase in water vapor (H₂O) is more likely contributing to the observed net heating. Increased H₂O can enhance the downward infrared (IR) radiation, leading to a minor warming effect at that altitude.

Table 1 – doing a visual comparison of these results with Schoeberl et al (2024) there is reasonable agreement, and differences can probably be ascribed to different SAOD and temperature changes not being included.

We thank the reviewer for this observation. As noted, our Table 1 shows a reasonable visual agreement with the results of Schoeberl et al. (2024). The remaining differences are most likely due to the different treatment of SAOD and the omission of temperature changes in our analysis.

L579 the reference here is to Pinatubo which was ~ 20Tg of sulfur compared to < 1 Tg of sulfur for HTHH – Pinatubo created a climate change of (at most) -0.5 K so scaling just with just sulfur the impact would be -0.025K which is probably not detectable.

We have revised the texts as: “Also, given that the Pinatubo eruption injected approximately 20 Tg of sulfur, resulting in a maximum cooling of about –0.5 K, a simple linear scaling suggests that the HTHH event, which injected less than 1 Tg of sulfur, would induce a cooling effect of approximately –0.025 K^{15, 22}. Such a small temperature change is likely to be masked by natural variability and may not be directly detectable. Given the modest magnitude of the expected cooling, further investigation using comprehensive climate model simulations is necessary to determine whether these gradients could trigger any measurable atmospheric responses.”

L593 Decade? More like 2.5 years – see Fleming et al. (2023)

Corrected.